# Local Signal Adaptivity: Provable Feature Learning in Neural Networks Beyond Kernels

**Stefani Karp**
Carnegie Mellon University and Google Research
shkarp@cs.cmu.edu

**Ezra Winston**
Carnegie Mellon University
ewinston@cs.cmu.edu

**Yuanzhi Li**
Carnegie Mellon University
yuanzhil@cs.cmu.edu

**Aarti Singh**
Carnegie Mellon University
aarti@cs.cmu.edu

## Abstract

Neural networks have been shown to outperform kernel methods in practice (including neural tangent kernels). Most theoretical explanations of this performance gap focus on learning a complex hypothesis class; in some cases, it is unclear whether this hypothesis class captures realistic data. In this work, we propose a related, but alternative, explanation for this performance gap in the *image classification* setting, based on finding a sparse signal in the presence of noise. Specifically, we prove that, for a simple data distribution with sparse signal amidst high-variance noise, a simple convolutional neural network trained using stochastic gradient descent simultaneously learns to threshold out the noise and find the signal. On the other hand, the corresponding neural tangent kernel, with a *fixed* set of predetermined features, is unable to *adapt* to the signal in this manner. We supplement our theoretical results by demonstrating this phenomenon empirically: in CIFAR-10 and MNIST images with various backgrounds, as the background noise increases in intensity, a CNN's performance stays relatively robust, whereas its corresponding neural tangent kernel sees a notable drop in performance. We therefore propose the *local signal adaptivity (LSA)* phenomenon as one explanation for the superiority of neural networks over kernel methods.

## 1 Introduction

Recently, deep learning (using multi-layer, *non-linear* neural networks) has demonstrated superior performance over traditional linear learners in many machine learning tasks. These achievements have bred much *theoretical* investigation into whether neural networks are, in fact, superior - and why. On the one hand, the Neural Tangent Kernel (NTK) and derivative works show that, under certain (limiting) conditions, a gradient-descent-trained neural network reduces to a kernel method with a specific architecture- and initialization-determined kernel [Jacot et al., 2018, Du et al., 2019]. However, this does *not* seem to be the full story, as it fails to capture the feature learning aspect of neural network training. This distinction between a *fixed* feature representation and a *data-adaptive* feature representation has been studied from a variety of perspectives, including the *lazy* vs. *active* regime framework [Chizat et al., 2019, Woodworth et al., 2020, Moroshko et al., 2020, Geiger et al., 2020, Wang et al., 2020]. Building on these insights, there has been increasing interest in now showing a *provable gap* between the performance of neural networks and kernel methods in various settings [Ghorbani et al., 2019, 2020, Allen-Zhu and Li, 2019, 2020, Li et al., 2020b, Malach et al., 2021, Kamath et al., 2020, Refinetti et al., 2021, Daniely and Malach, 2020, Chen et al., 2020, Domingo-Enrich et al., 2021].

35th Conference on Neural Information Processing Systems (NeurIPS 2021).

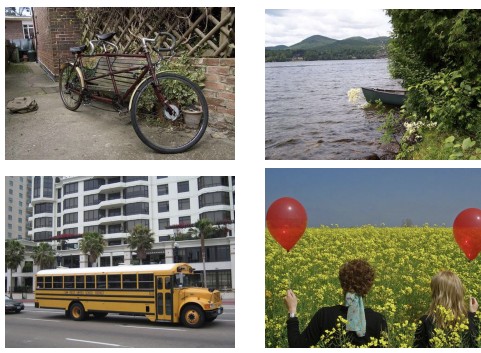

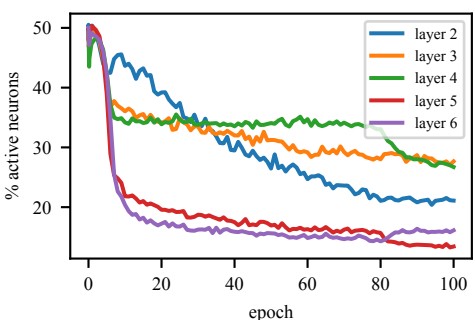

Figure 1: Examples from the IMAGENET2012 dataset, illustrating background noise in natural image classification tasks.

Figure 2: Sparsity of intermediate WRN layers during training on CIFAR-10.

In this work, we extend this theoretical investigation into the superiority of neural networks over linear learners and, inspired by practical settings, propose a new line of reasoning that we refer to as "Local Signal Adaptivity". In particular, we explore the power of convolutional neural networks (CNNs) in image classification, compared to linear functions over prescribed feature mappings.

**Our setting:** We study a simple data distribution that captures one key property of natural image classification tasks: a small set of localized "label-determining" features embedded within a "noisy" background. We formally prove that even when such background occupies a rather large fraction of an image, a CNN can be quite effective at locating the label-determining signal and thus ignoring background information that is mostly irrelevant to the true label, to achieve high accuracy.

**Our result:** In the formal setting of our simple data distribution (presented in Section 3), we ask whether a particular two-layer CNN trained via stochastic gradient descent can provably acquire this "signal-finding" ability and how this compares to its associated finite-width convolutional neural tangent kernel (CNTK). We answer with a separation result between our CNN and its CNTK: *We formally prove that a small CNN, trained using standard SGD from random initialization, can efficiently learn to find the "signal" and threshold out the noise*, whereas the corresponding CNTK requires *a comparatively larger model* (i.e., with more features) in order to accomplish this.

**Empirical justification:** While we pick a simple data distribution in our work to illustrate the main idea and obtain a formal proof, we point out that our setting is very natural in real images: in many image classification tasks, the label-determining feature only occupies a small fraction of the image, and most other parts are background noise (Figure 1). Furthermore, in neural networks trained on natural images, it is generally accepted that activation patterns become increasingly sparse throughout training, effectively zeroing out the activations of low-magnitude noise and locating the true signal (suggestive of the denoising/LSA phenomenon studied in this work). For completeness, we have included such an experiment in Figure 2 above, illustrating how the average percentage of active neurons per instance decreases throughout training (details are provided in Appendix C). Finally, to empirically study our theoretical results, we create new datasets by embedding CIFAR-10 and MNIST images within either random Gaussian or IMAGENET backgrounds. Our experiments show that, as the intensity of the background noise grows and thus the "denoising task" becomes harder, the performance of the neural network stays relatively stable, while the performance of the corresponding NTK does, in fact, degrade significantly (Section 6, Figure 4).

Based on our theorem and experiments, we therefore believe that this per-instance "signal finding within noisy backgrounds" ability of convolutional neural networks, which we dub "Local Signal Adaptivity" (LSA), is one key component of the superiority of SGD-trained convolutional neural networks over fixed feature mappings.

## 2 Related work

There is a long line of work proving that various concept classes can be learned efficiently by SGD-trained neural networks. Many of the theoretical works taking a "beyond-NTK" perspective and emphasizing the non-linearity of neural networks are only proved under Gaussian inputs [Kawaguchi, 2016, Soudry and Carmon, 2016, Xie et al., 2016, Ge et al., 2017, Soltanolkotabi et al., 2017, Tian, 2017, Brutzkus and Globerson, 2017, Zhong et al., 2017, Li and Yuan, 2017, Boob and Lan, 2017, Li et al., 2018, Vempala and Wilmes, 2018, Ge et al., 2018, Bakshi et al., 2018, Oymak and Soltanolkotabi, 2019, Yehudai and Shamir, 2019, Li and Yuan, 2017, Li and Liang, 2017, Li et al., 2016, Li and Dou, 2020, Li et al., 2020a]. In our work, we consider a more realistic setting in which there is a highly-structured signal hidden within random background noise. Along these lines, we recognize various aesthetic similarities to sparse blind deconvolution and related areas of the signal processing literature [Zhang et al., 2019, Kuo et al., 2019, Sun and Donoho, 2021, Qu et al., 2020]. However, there are key structural differences as well - e.g., rather than recovering the unknown sparse signal, we simply wish to extract the information relevant to the classification task; thus, further exploring possible connections between LSA and the signal processing literature presents an interesting avenue for future work.

We now give a more detailed comparison of our work with prior theoretical results on the separation between neural networks and kernels.

**Representation power of neural networks.** Many existing works focus on separating the representation power of neural networks from that of other models [Ghorbani et al., 2020, Refinetti et al., 2021, Gribonval et al., 2020, Suzuki, 2019, Suzuki and Nitanda, 2019]. However, the fact that a function can be represented efficiently by a neural network does *not* mean that it can be efficiently learned. Only a subset of such works prove efficient learnability, including Suzuki and Akiyama [2020] and Daniely and Malach [2020]. In our work, we focus on the set of functions that can be *efficiently learned* by neural networks, in particular learned via stochastic gradient descent over the standard training objective starting from random initialization.

**Classification vs. regression.** Many existing works only focus on separating the power of neural networks from that of other learners in a regression setting [Ghorbani et al., 2020, Suzuki and Akiyama, 2020, Allen-Zhu and Li, 2020, 2019, Li et al., 2020a, Wei et al., 2018, Yehudai and Shamir, 2019]. In this case, both the neural network and the other learning methods are required to fit the exact label. Although Daniely and Malach [2020] presents one of the few results in a classification setting, their final separation result is still in terms of hinge loss instead of 0-1 loss. In our result, we focus on the *binary classification setting*, where the neural network and (significantly) the linear learner are only required to fit the sign of the label, instead of the exact labeling function. Our final result is therefore more natural in image classification settings than prior works.

**Learning a hidden subspace.** Many of the existing works showing how neural networks are better than other learners, in particular kernel methods, focus on the case where the concept class is of the form $f(x) = \phi(Wx)$, where $W$ is a rank-deficient matrix [Ghorbani et al., 2020, Wei et al., 2018]. Thus, the learning process can be divided into two phases: (1) Identifying the hidden subspace of $W$; (2) Learning the function $\phi$ over this hidden subspace. We point out that to the best of our knowledge, all the cited works only shows the superior power of neural networks in (1). This means that, if $W$ were known, then a neural network would have the same sample/time complexity as kernels when performing (2). As we will show, in our work, the underlying concept class does not have a fixed subspace, rather the signal can drift between different patches, and the neural network needs to identify the signal patch while ignoring the noise patches. Our explanation of how neural networks outperform kernels is therefore fundamentally different from the cited works.

**Kernel lower bounds.** Most of the existing theoretical works demonstrating the power of neural networks focus on showing how neural networks are better than kernel methods or linear functions over prescribed feature mappings. There are two related lines of work: (1) Showing that neural networks are better than any kernel method or any linear learner [Allen-Zhu and Li, 2020, Daniely and Malach, 2020]; (2) Showing that neural networks are better than a *particular* set of linear learners, most notably, the NTKs or finite-width NTKs of the corresponding neural network [Yehudai and Shamir, 2019, Refinetti et al., 2021, Wei et al., 2018]. Our work belongs to the second line.

Although a result along the first line gives a much stronger separation, we point out that in the *classification* setting, *such a separation is known to be extremely challenging in computational complexity theory* [O'Donnell and Servedio, 2003, Sherstov and Wu, 2019].

**Finite- vs. infinite-width NTK.**  Empirical results in the literature suggest that the infinite-width NTK generally reaches slightly higher final test accuracy than the finite-width NTK [Allen-Zhu and Li, 2020, Lee et al., 2020]. In fact, in certain limited settings, such as low-data regimes, the infinite-width NTK has actually demonstrated performance competitive with that of the original neural network [Arora et al., 2020]. However, under certain conditions, the computational cost of the infinite-width NTK can be significant compared to that of the original neural network and its corresponding finite-width NTK [Allen-Zhu and Li, 2020, Lee et al., 2020]. Therefore, due to these computational and accuracy distinctions between the finite- and infinite-width NTKs, in our experiments we examine both types of NTK. Our theoretical lower bound, however, is a representational lower bound for the finite-width NTK. This is primarily due to theoretical challenges such as those discussed above. Theoretical results such as Allen-Zhu and Li [2020] suggest that it may be possible, in follow-up work, to convert our finite-width NTK representational lower bound into an infinite-width NTK sample complexity lower bound. However, proving a sample complexity gap would likely require tightening our sample complexity upper bound as well. We believe, conceptually (and supported by our experiments), that the LSA phenomenon is at play for infinite-width NTKs as well, and it would thus be interesting to extend our theoretical gap to the infinite-width NTK setting as well.

## 3   Problem setup

We now formally state our data distribution and the learning problem in our paper.

**Relevant problem parameters.**  We consider the input $X = (X_1, \cdots, X_{k+1})$ to have $k + 1$ patches; each patch is associated with a vector $X_i \in \mathbb{R}^d$. We can think of $X$ as either the input image or the output of some intermediate layer of the convolutional neural network. It is convenient to think of $X$ as a matrix with $k + 1$ rows and $d$ columns, i.e., $X \in \mathbb{R}^{(k+1)\times d}$. We treat $k$ as "large" and $d$ as polynomial in $k$, enabling us to simplify certain results by expressing them entirely in terms of $k$. We consider an unknown signal vector $\mathbf{w}_\star \in \mathbb{R}^d$ with $\ell_2$-norm $\|\mathbf{w}_\star\|_2 = 1$, which determines how each (image) $X$ is generated (described below). We also set $\sigma = \log k / \sqrt{k}$, where $\sigma$ determines the intensity of the noise in our problem (described below). In Section 5, we will only prove the kernel lower bound in the case where $\sigma^2 = 1/d$; extending the result to $\sigma^2 = \omega(1/d)$ might require alternative proof techniques.

**Data distribution.**  We sample each (image, label) pair $(X, y) \sim \mathcal{D}$. The marginal distribution $\mathcal{D}_y$ is a uniform distribution over $\{-1, 1\}$, i.e., both classes occur with equal probability. We first sample $y$ from $\mathcal{D}_y$ and then sample $X \sim \mathcal{D}_{X|y}$ as follows: (a) draw $i^\star$ from $\{1, \ldots, k+1\}$ arbitrarily and put $y\mathbf{w}_\star$ in row $i^\star$ (we call this the "signal" row); (b) in each of the $k$ remaining rows (we call these the "noise" rows), put an independently-drawn vector $\epsilon_i \sim \mathcal{N}(0, \sigma^2 I_{d\times d})$. We use $X_i \in \mathbb{R}^d$ to denote row $i$ of $X$, for $i \in \{1, \ldots, k+1\}$. Formally, we therefore have:

$$X_i = \begin{cases} y\mathbf{w}_\star & \text{if } i = i^\star \\ \epsilon_i & \text{otherwise.} \end{cases}$$

Our data distribution shares many similarities with the distribution studied in Yu et al. [2019]. However, perhaps most crucially among the *differences*, the noise magnitude in our setting is *significantly* higher; in Yu et al. [2019], the noise magnitude is low enough to maintain *linear* separability, which is insufficient to show any separation with linear learners.

**Remark.**  We consider the simplest setting where there is only one feature $\mathbf{w}_\star$, though our result trivially extends to the case when there are multiple orthogonal features (as many as $d$). For example, the signal patch can contain features of the form $\sum_i \alpha_i \mathbf{w}_i^\star$, where the label is determined by $\sum_i \alpha_i$, as in Allen-Zhu and Li [2020].

**How to learn this concept class.**  In our concept class, one of the rows $X_i$ is associated with the true signal $y\mathbf{w}_\star$, and all the others are random Gaussian noise. Since this row can appear arbitrarily

in one of the $k+1$ rows, the most naive way to learn this function is to use a convolutional linear classifier: $l(X) = \sum_{i=1}^{k+1} \langle \mathbf{w}_\star, X_i \rangle$. However, we argue that this linear classifier is actually very bad and cannot be used to classify the labels correctly. Note that in our problem setup, the noise rows are sampled according to $\mathcal{N}(0, \sigma^2 I_{d \times d})$, which means that for each noise row $i$, we have $\langle \mathbf{w}_\star, X_i \rangle \sim \mathcal{N}(0, \sigma^2)$. Hence, the *total accumulated noise* over $k$ noise rows would be $\mathcal{N}(0, k\sigma^2)$. By our choice of $\sigma = \log k / \sqrt{k}$, this means that the total noise is much bigger than the signal $|y| = 1$. Hence, the linear classifier fails to classify the labels correctly with at least probability $0.49$.

The above argument suggests that, to learn the concept class, the model cannot naively sum up $\langle \mathbf{w}_\star, X_i \rangle$. Rather, the model has to identify the signal row and ignore the noise rows. In other words, the model needs to distinguish between the case when $\langle \mathbf{w}_\star, X_i \rangle$ is large ($y$) or small ($\mathcal{N}(0, \sigma^2)$). As we will show, this can be *efficiently* learned by a neural network with ReLU activations, trained using standard stochastic gradient descent from random initialization.

**Convolutional neural network (CNN) architecture.**   Our goal is to learn the optimal parameters $\mathbf{w} \in \mathbb{R}^d, b \in \mathbb{R}$ of the following simple CNN:

$$f_{\mathbf{w},b}(x) = \sum_{i=1}^{k+1} \phi_b(\langle \mathbf{w}, X_i \rangle) = \sum_{i=1}^{k+1} \Big[ \mathrm{ReLU}(\langle \mathbf{w}, X_i \rangle + b) - \mathrm{ReLU}(-\langle \mathbf{w}, X_i \rangle + b) \Big].$$

This CNN can be understood from *either* of two equivalent perspectives:

(1) A single convolutional filter $\mathbf{w} \in \mathbb{R}^d$ with stride $d$ is applied to the image. Then the *soft-thresholding* activation function $\phi_b(x) = \mathrm{ReLU}(x + b) - \mathrm{ReLU}(-x + b)$ is applied pointwise over the resulting $(k+1)$-dimensional vector. Finally, the second (non-trainable) layer of the CNN simply sums up the $k+1$ entries (i.e., the second layer is a fully-connected layer mapping from $\mathbb{R}^{k+1}$ to $\mathbb{R}$, with non-trainable weights all equal to 1).

(2) *Two* convolutional filters, each with stride $d$, are applied to the image. One filter is $\mathbf{w} \in \mathbb{R}^d$ and the other is $-\mathbf{w} \in \mathbb{R}^d$. After each filter is applied, the activation function $\varphi_b(x) = \mathrm{ReLU}(x + b)$ is applied pointwise to all $2k+2$ pre-activation values (i.e., a $k+1$ "vector" with *2 channels*). A non-trainable fully-connected layer is then applied, mapping $\mathbb{R}^{2k+2}$ to $\mathbb{R}$; in this layer, each of the $k+1$ weights applied to the the first channel (i.e., from filter $\mathbf{w}$) are all 1, and each of the weights applied to the second channel (i.e., from filter $-\mathbf{w}$) are all $-1$.

The key in both perspectives is how the two ReLU activation functions work together to implement a soft-thresholding function, which enables denoising. Throughout, we refer to our CNN as having one filter, since it only has $d+1$ trainable parameters, regardless of which perspective is taken.

**Training algorithm.**   We initialize $b$ deterministically at 0. We initialize $\mathbf{w}$ randomly by drawing from $\mathcal{N}(0, \sigma_0^2 I_{d \times d})$, where $\sigma_0$ is $1/\mathrm{poly}(k)$. We train the above CNN using mini-batch stochastic gradient descent (SGD) with the logistic loss, where the logistic loss $\ell$ is defined as $\ell(f_{\mathbf{w},b}(X), y) := \log(1 + e^{-y f_{\mathbf{w},b}(X)})$.

Specifically, at each iteration of SGD, we use $\mathrm{poly}(k)$ fresh samples from $\mathcal{D}$. This will allow us to argue that, at each iteration, the empirical gradient is very close to the true population loss gradient.

To simply the proof, we use a slightly smaller learning rate for $b$ (denoted $\eta_b$) than for $\mathbf{w}$ (denoted $\eta_\mathbf{w}$). Specifically, we adopt a $1/\mathrm{poly}(k)$ learning rate for $\mathbf{w}$, and we set $\eta_b/\eta_\mathbf{w} = 1/k$. With a bit more technical care, our proofs can be extended to cover the setting where $\eta_b = \eta_\mathbf{w}$ but we use this simpler version to illustrate the main idea of the learning process, as we will state in the next section.

To avoid any ambiguity, we define this procedure explicitly as Algorithm 1.

---

**Algorithm 1** Mini-batch SGD

---

Initialization and learning rate $b^{(0)} \leftarrow 0;$    $\mathbf{w}^{(0)} \leftarrow \mathcal{N}(0, \sigma_0^2 I_{d \times d}); \quad \eta_b \leftarrow \eta/k; \quad \eta_\mathbf{w} \leftarrow \eta$
**for** $t \leftarrow 1 \ldots T$ **do**
     Sample a mini-batch of examples of size $n = \mathrm{poly}(k)$: $\mathcal{Z} \leftarrow \mathcal{D}^n$
     Compute the stochastic gradient: $g_b = \frac{1}{n} \sum_{z \in \mathcal{Z}} \nabla_b \ell(z)$, $g_\mathbf{w} = \frac{1}{n} \sum_{z \in \mathcal{Z}} \nabla_\mathbf{w} \ell(z)$
     Update using stochastic gradient descent: $b^{(t)} \leftarrow b^{(t-1)} - \eta_b g_b$, $\mathbf{w}^{(t)} \leftarrow \mathbf{w}^{(t-1)} - \eta_\mathbf{w} g_\mathbf{w}$
**return** $b^{(T)}, \mathbf{w}^{(T)}$

---

**Additional notation.** We will occasionally use the shorthand $f_t(\cdot)$ to denote $f_{\mathbf{w}^{(t)}, b^{(t)}}(\cdot)$, the network at iteration $t$. We use the standard big-O notation and its variants: $\mathcal{O}(\cdot), o(\cdot), \Theta(\cdot), \Omega(\cdot), \omega(\cdot)$, where $k$ is the problem parameter that becomes large. Occasionally, we use the symbol $\widetilde{\mathcal{O}}(\cdot)$ (and analogously with the other four variants) to hide $\log k$ factors.

Our results will hold with high probability over the random initialization of $\mathbf{w}$ and the randomness of the mini-batches, where "high probability" here means a failure probability super-polynomially small in $k$. Unless specified otherwise, *w.h.p.* can be taken to mean: with probability at least $1 - e^{-\Omega(\log^2 k)}$.

## 4  Neural network results

We now present and briefly describe our main result on the provably *efficient* learning of the CNN described in Section 3.

**Theorem 1** (Main result). *There exists an absolute constant $k_0$ such that, for every $k \geq k_0$, using poly$(k)$ samples from $\mathcal{D}$, learning rate $\eta = 1/poly(k)$, and $T = poly(k)$ iterations, w.h.p. over the randomness of the initialization and the samples, we have $\Pr_{(X,y)\sim\mathcal{D}}[sign(f_T(X)) \neq y] \leq 0.01$, for the final network $f_T$ returned by Algorithm 1.*

Therefore, in contrast to its corresponding Neural Tangent Kernel (Section 5), our CNN trained via SGD *provably* achieves high accuracy *efficiently*. This stands in contrast to some of the prior works discussed in Section 2, many of which do not prove efficient learnability.

We defer our detailed proofs to Appendix A and summarize the key ideas here.

**Empirical vs. population gradients.** Our general proof technique involves analyzing the gradient of the *population loss* at each iteration (we call this the *true gradient*): $\nabla_{\mathbf{w}} \mathbb{E}_{(X,y)\sim\mathcal{D}}[\ell(f_{\mathbf{w},b}(X), y)]$ and $\nabla_b \mathbb{E}_{(X,y)\sim\mathcal{D}}[\ell(f_{\mathbf{w},b}(X), y)]$. Then, with poly$(k)$ samples per mini-batch, we argue that *w.h.p.* the empirical gradient concentrates around the true gradient, and over $T = poly(k)$ steps, the accumulated error is fairly small. This argument is made rigorous in Appendix A. Therefore, in the remainder of this section, to illustrate the key idea of the proof, we limit our discussion to the *true gradient* as just defined (and use the term *gradient* or *derivative* without further qualification).

**Learning the signal direction.** As training progresses, $\mathbf{w} \cdot \mathbf{w}_\star$ grows from its small-magnitude initialization to a relatively large, but still $\mathcal{O}(1)$, *positive* value. Specifically, at each iteration, the gradient with respect to $\mathbf{w}$ has a component parallel to $\mathbf{w}_\star$, with two *opposing* forces determining the sign/magnitude of this component: in expectation over $(X, y) \sim \mathcal{D}$,

1. the $k$ "noise" rows of $X$ push $\mathbf{w} \cdot \mathbf{w}_\star$ to *shrink* and
2. the "signal" row pushes $\mathbf{w} \cdot \mathbf{w}_\star$ to *grow*.

The "signal" row's contribution has a magnitude of $\Theta(1)$, and the $k$ "noise" rows have a total contribution of magnitude at most $\mathcal{O}\left(|\mathbf{w} \cdot \mathbf{w}_\star| \cdot k \log k \cdot \sigma^2\right) = \mathcal{O}\left(|\mathbf{w} \cdot \mathbf{w}_\star| \cdot \log^3 k\right)$. Therefore, as long as $|\mathbf{w} \cdot \mathbf{w}_\star| = o(1/\log^3 k)$, the "signal" row's contribution overpowers the "noise" rows' contribution, causing $\mathbf{w} \cdot \mathbf{w}_\star$ to grow. This means that, within $\Theta((\eta_{\mathbf{w}} \log^4 k)^{-1}) = poly(k)$ iterations, $\mathbf{w} \cdot \mathbf{w}_\star$ rises from its small-magnitude initialization to $\Theta(1/\log^4 k)$. After $\mathbf{w} \cdot \mathbf{w}_\star$ rises to $\Theta(1/\log^4 k)$, we no longer track its dynamics explicitly, as it must stay somewhere between $\Omega(1/\log^4 k)$ and $\mathcal{O}(1)$ throughout the rest of training, and this turns out to be sufficient for the remainder of the argument. The main lemma we prove in Appendix A regarding the growth of $\mathbf{w} \cdot \mathbf{w}_\star$ is a slightly more formal version of the following:

**Lemma 1** (Sketched). *For any $t \leq T$, if $|\mathbf{w}^{(t)} \cdot \mathbf{w}_\star| = o\left(\frac{1}{\log^3 k}\right)$, then $\nabla_{\mathbf{w}} \mathbb{E}[\ell(f_t(X), y)] \cdot \mathbf{w}_\star = -\Theta(1)$.*

**Bounded growth along non-signal directions.** As training progresses, we also track $\|\mathbf{w} - (\mathbf{w} \cdot \mathbf{w}_\star)\mathbf{w}_\star\|_2$, the magnitude of $\mathbf{w}$'s projection onto the orthogonal complement of $\text{span}\{\mathbf{w}_\star\}$. Although the projection's direction can change, we show that its norm remains very small, allowing $\mathbf{w} \cdot \mathbf{w}_\star$ to dominate.

Specifically, letting $\mathbf{w}_\perp$ denote $\mathbf{w} - (\mathbf{w} \cdot \mathbf{w}_\star)\mathbf{w}_\star$ at the start of iteration $t$, we show that $\nabla_\mathbf{w}\mathbb{E}[\ell(f_t(X), y)] \in \mathrm{span}\{\mathbf{w}_\star, \mathbf{w}_\perp\}$. Further, $\nabla_\mathbf{w}\mathbb{E}[\ell(f_t(X), y)] \cdot \mathbf{w}_\perp/\|\mathbf{w}_\perp\|_2$ mirrors the "noise" contribution above: it has magnitude at most $\mathcal{O}\left(\|\mathbf{w}_\perp\|_2 \cdot k \log k \cdot \sigma^2\right) = \mathcal{O}\left(\|\mathbf{w}_\perp\|_2 \cdot \log^3 k\right)$ and pushes $\|\mathbf{w}_\perp\|_2$ to *shrink*. Thus, if we were actually using the *true gradient*, $\mathbf{w}_\perp$ would maintain its direction, and its norm would shrink. The effect of the *stochastic* gradient is that $\mathbf{w}_\perp$ *can* change direction slightly, and its norm can grow a bounded amount per iteration; however, as long as the mini-batch size is large enough (poly($k$) suffices), its norm stays small enough throughout training. The main lemma we prove in Appendix A regarding $\|\mathbf{w}_\perp\|_2$ is therefore a slightly more formal version of the following:

**Lemma 2** (Sketched). *For any $t \leq T$, let $g_\perp^{(t)}$ denote $\nabla_\mathbf{w}\mathbb{E}[\ell(f_t(X), y)] \cdot \mathbf{w}_\perp^{(t)}/\|\mathbf{w}_\perp^{(t)}\|_2$. Then $g_\perp^{(t)} \geq 0$ and $g_\perp^{(t)} = \mathcal{O}\left(\|\mathbf{w}_\perp^{(t)}\|_2 \cdot \log^3 k\right)$.*

**Learning to threshold out the noise.** The derivative with respect to $b$ similarly has two *opposing* forces determining its sign/magnitude: in expectation over $(X, y) \sim \mathcal{D}$,

1. the $k$ "noise" rows of $X$ push $b$ to decrease and
2. the "signal" row pushes $b$ to increase.

The "signal" row's contribution has a magnitude of $\Theta(1)$. Thus, we can only guarantee that $b$ is decreasing if the "noise" rows' total contribution is $\omega(1)$. Roughly, the "noise" rows' contribution is determined by (i) the scalar projection of each noise vector $\epsilon_i$ onto $\mathbf{w}$ and (ii) (when $b < 0$) how much of these scalar projections survive "thresholding". As $\|\mathbf{w}\|_2$ grows throughout training (primarily due to the growth of $\mathbf{w} \cdot \mathbf{w}_\star$, discussed above), (i) becomes larger. However, as $b$ decreases (i.e, $|b|$ grows, for $b < 0$), (ii) becomes smaller (i.e., much of the noise does not survive "thresholding"). Therefore, the crux of the proof is to show that, as long as the probability of misclassification is still $> 0.01$, despite $b$ already "thresholding out" a fair amount of the noise, the "noise" rows' total contribution will remain $\omega(1)$ and thus cause $b$ to further decrease. The main lemma we prove in Appendix A regarding $b$ is therefore a slightly more formal version of the following:

**Lemma 3** (Sketched). *For any $t \leq T$, if $\mathbf{w}^{(t)} \cdot \mathbf{w}_\star = \omega(1/k^{1/8})$, then $\nabla_b\mathbb{E}[\ell(f_t(X), y)] = \Omega(1)$.*

We note that the requirement $\mathbf{w}^{(t)} \cdot \mathbf{w}_\star = \omega(1/k^{1/8})$ is satisfied by the $\Omega(1/\log^4 k)$ lower bound presented above, thus connecting the two phases of training.

**Comment on actual dynamics.** The elegance of this approach is that it largely allows us to ignore $b$'s behavior prior to $\mathbf{w} \cdot \mathbf{w}_\star$ reaching the $\Omega(1/\log^4 k)$ regime. In reality, there is a short phase at the beginning of training during which $b$ *grows* (i.e., becomes increasingly *positive*); this occurs because, even though none of the noise is being "thresholded out" at this point, $\|\mathbf{w}\|_2$ is not yet large enough for the "noise" rows' contribution to dominate the "signal" row's $\Theta(1)$ contribution (which itself does *not* scale with $\|\mathbf{w}\|_2$). Then, at some point *prior* to $\mathbf{w} \cdot \mathbf{w}_\star$ reaching $\Omega(1/\log^4 k)$, $b$ begins to decrease rather than increase - and, as we prove, eventually continues to decrease throughout the rest of training. However, for our performance guarantee, it does not matter exactly *when* this transition from increasing to decreasing actually occurs. The $\eta_b/\eta_\mathbf{w} = 1/k$ ratio ensures that $|b|/|\mathbf{w} \cdot \mathbf{w}_\star|$ never exceeds $\Theta(1/k)$ while $b > 0$, which means that $b = \mathcal{O}(1/k)$ throughout its *positive* phase.

**Provable efficiency.** We argue that if $\Pr_{(X,y)\sim\mathcal{D}}[\mathrm{sign}(f_T(X)) \neq y] \geq 0.01$, then we must have $|\mathbf{w} \cdot \mathbf{w}_\star| = \mathcal{O}(1)$. This, along with the rest of the argument, is made fully rigorous in Appendix A. We sum up the total number of iterations as follows. First, there are $\mathcal{O}((\eta_\mathbf{w} \log^4 k)^{-1})$ iterations before $b$ begins to decrease. Then, because $|\mathbf{w} \cdot \mathbf{w}_\star| = \mathcal{O}(1)$, $\nabla_b\mathbb{E}[\ell(f_t(X), y)] = \Omega(1)$, and the classification accuracy is controlled by the ratio $|b|/|\mathbf{w} \cdot \mathbf{w}_\star|$, we conclude that there are poly($k$) iterations before $|b|$ becomes large enough to yield $\Pr_{(X,y)\sim\mathcal{D}}[\mathrm{sign}(f_T(X)) \neq y] \leq 0.01$. Thus, $T = \mathrm{poly}(k)$, and with poly($k$) samples per mini-batch, we have total sample complexity poly($k$).

**Illustration of training dynamics with synthetic data.** Figure 3 illustrates these training dynamics on synthetic data (with $k = 1000$, $d = 10$, $\sigma = 1$, $\eta_w = 0.1$, $\eta_b = 0.1/1000$). As can be seen in the figure, as training progresses, $\mathbf{w} \cdot \mathbf{w}_\star$ increases and $b$ decreases (i.e., $b < 0$, with increasing magnitude). Furthermore, as the ratio $|b|/|\mathbf{w} \cdot \mathbf{w}_\star|$ grows (and thus more and more noise is "thresholded out"), the test accuracy increases as well, tightly aligning with the theory.

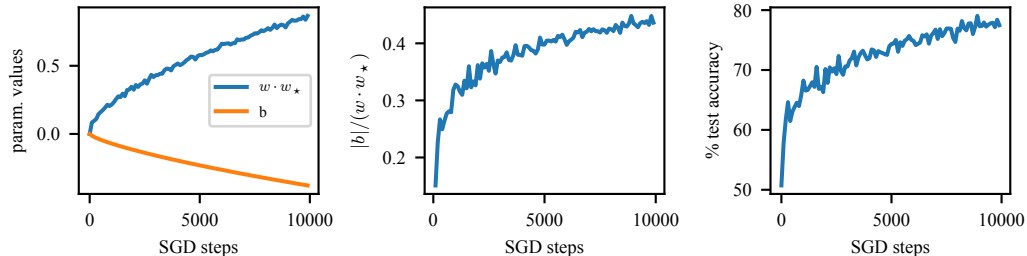

Figure 3: Training our model on synthetic data. Left: Values of $b$ and $\mathbf{w} \cdot \mathbf{w}_\star$. Center: Ratio $|b|/(\mathbf{w} \cdot \mathbf{w}_\star)$. Right: Test accuracy.

## 5 Kernel results

In this section, we compare the function learned by our simple convolution neural network to its corresponding finite-width CNTK [Allen-Zhu et al., 2019]. We define the finite-width CNTK as:

$$k_{\mathbf{w}}(X) = \sum_{i \in [k]} \sum_{j \in [m]} \langle \mathbf{w}_j, X_i \rangle 1_{|\langle \mathbf{w}_j^{(0)}, X_i \rangle| + b_j \geq 0},$$

which is a linear function over the gradient of $f_{\mathbf{w},b}(X)$ at initialization. We show that this NTK is unable to classify the target function correctly, unless the total number of channels $m$ is large.

We prove this for the case where $\sigma^2 = 1/d$. Extending the result to $\sigma^2 = \omega(1/d)$ might require alternative proof techniques.

**Theorem 2.** *As long as $m = O(1)$, w.p. at least $0.999$ over the random initialization $\{\mathbf{w}_j^{(0)}\}_{j \in [m]}$ where each $\mathbf{w}_j^{(0)}$ i.i.d. $\sim \mathcal{N}(0, \sigma_0^2 I)$, we have that for every set of weights $\mathbf{w}$ and for every set of biases $\{b_j\}_{j \in [m]}$,*

$$\Pr_{X,y\sim\mathcal{D}}[sign(k_{\mathbf{w}}(X)) \neq y] \geq 0.1.$$

Compared to our convolutional neural network, which only requires $m = 1$ neurons and can learn the target function correctly, the corresponding finite-width CNTK needs $\omega(1)$ neurons in order to even represent the target function. This is due to the fact that the features $\mathbf{w}_j^{(0)}$ *are prescribed at random initialization*, instead of trained. Thus, even with arbitrary tuned bias $b_j$, the neural tangent kernel still fails to adapt to the local signal and perform denoising.

## 6 Experiments

We now provide concrete empirical evidence that the LSA phenomenon does explain the performance gap between CNNs and kernels in real-world datasets. We first construct several variants of CIFAR-10 [Krizhevsky, 2009] with various forms of structured noise and compare how the scale of the noise affects CNNs and their corresponding finite-width NTKs. Next, we perform similar experiments using a smaller CNN on MNIST [LeCun et al., 2010]; in this setting we can efficiently compute the infinite-width NTK, as well as vary the width of the finite-width NTK, and observe whether the performance gap persists. CNN architecture and training details are given in Appendix C.[1]

**CIFAR with structured noise.** We construct several CIFAR-10 variants with structured noise, which capture a key property of real-world image classification tasks, that the signal is localized to patch amidst a large amount of background noise. Although real-world images also exhibit this property (as in Figure 1), we wanted the ability to easily vary the intensity of the background noise, while leaving the signal intact. On each dataset, we compare the performance of a 10-layer WideResNet (with widening factor of 10) to that of its finite-width NTK [Zagoruyko and Komodakis,

---

[1]Code for experiments is available at `https://github.com/skarp/local-signal-adaptivity`.

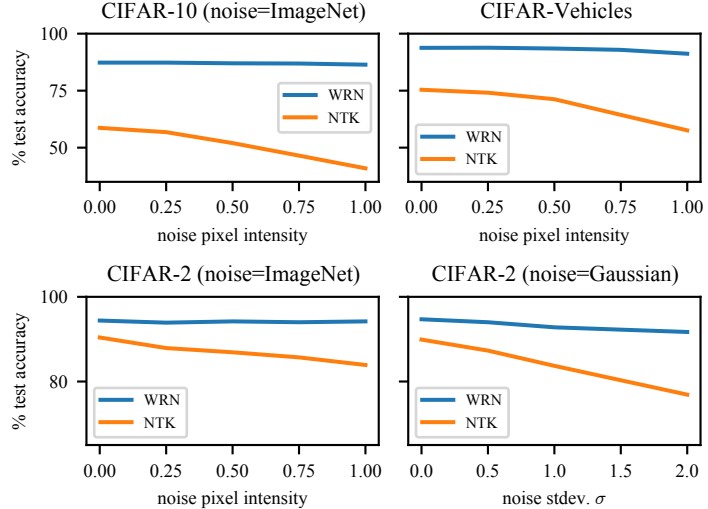
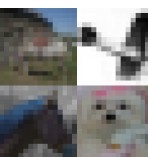
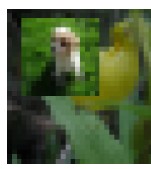

Figure 5: Examples from CIFAR-VEHICLES (top) and CIFAR-10 with IMAGENET noise (bottom), both for background noise pixel intensity scaled to 0.75.

Figure 4: WRN and NTK test accuracy over a range of noise levels.

2016]. We vary the intensity of the noise and observe the resulting degradation in WideResNet (WRN) and NTK performance.

Each dataset is constructed by scaling a CIFAR image to 16x16 pixels and placing it onto a 32x32 noise background. In some instances described below, we used IMAGENET backgrounds [Deng et al., 2009]. Here we highlight experiments on four such datasets:

- **CIFAR-10, IMAGENET noise.** CIFAR-10 images are placed at a random location onto random background image chosen from the IMAGENET Plants synset. The Plants synset was chosen for its visual similarity to backgrounds in real images. In our experiments, we scale the background pixel intensity in a range between zero (black background) and one (original IMAGENET background). An example is shown in Figure 5 (bottom).

- **CIFAR-2, IMAGENET noise.** Same as above, except the CIFAR-10 classes are grouped into animals and vehicles and the classification task is now binary. In contrast to CIFAR-10, on CIFAR-2 the NTK performs nearly as well as the CNN in the zero-noise setting.

- **CIFAR-2, Gaussian noise.** Also using the CIFAR-2 task, but with standard Gaussian noise in the background, for a range of standard deviations $\sigma$.

- **CIFAR-VEHICLES.** The task is to classify between the four vehicle classes from CIFAR-10. The vehicle appears in a random corner of the image, and the other three corners are filled with random images from the four CIFAR-10 animal classes. See Figure 5 (top). Unlike the relatively uniform IMAGENET plants backgrounds, this dataset is designed to capture a common type of background noise which consists of "distractor" signals which are not predictive of the true image class. For example, irrelevant people, bicycles, and birds could all occur in the background of a real-world vehicle-classification task.

For a more natural setting, we conduct an additional experiment on CIFAR-2 with IMAGENET noise, but maintaining the full pixel intensity and varying the size of the background noise. This is detailed in Appendix D, along with additional experiment variants such as different image placements, all of which display similar behavior to the experiments described here. While these datasets capture key attributes of real images, their synthetic construction does limit how realistic the images can be. We construct the datasets in this way in order to allow tunable noise levels, thus providing the $x$-axis of the plots in Figure 4. One could imagine alternative, more realistic ways of creating tunable noise levels, such as increasing the diversity of distractor images; these are viable avenues for future experimentation.

**Observations.** As seen in Figure 4, as the scale of the noise increases, NTK performance decreases significantly while WRN performance is relatively unaffected. Table 1 gives the percent of test

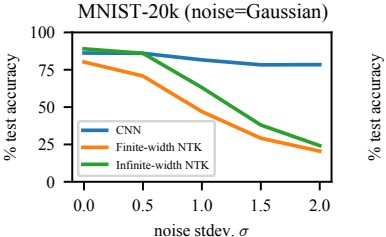
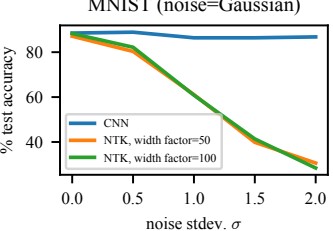

Figure 6: Comparison of CNN, finite-width, and infinite-width NTKs on a 20k MNIST sample (left), and various finite-width NTKs on full MNIST (right).

accuracy retained by each model as noise intensity increases from 0 to 1 (or from $\sigma = 0$ to $\sigma = 2$ in the Gaussian-noise case); that is, the percentage $100 \cdot \frac{\text{test acc. at max noise level}}{\text{test acc. at zero noise}}$. The WRN always retains over $96\%$ of its zero-noise performance, while the NTK in one case degrades below $70\%$.

Table 1: Percentage of zero-noise test accuracy retained at maximum noise level.

| | **CIFAR-10**, ImNet noise | **CIFAR-Vehicles** | **CIFAR-2**, ImNet noise | **CIFAR-2**, Gauss. noise |
|---|---|---|---|---|
| **WRN** | 98.97 | 97.25 | 99.79 | 96.83 |
| **NTK** | 69.75 | 76.37 | 92.81 | 85.54 |

**Effect of NTK width (MNIST).**  We place MNIST images of size 28x28 onto 42x42 Gaussian-noise backgrounds. We use a small CNN with two convolutions with 8 and 16 channels, and compare to finite-width NTKs where the width has been increased by a factor of between 50 and 100. On a subset of 20k MNIST images, we also compare the same CNN and finite-width NTK (width factor=1) to the infinite-width NTK.

The results (Figure 6) indicate that the degradation of NTK performance persists for both infinite- and finite-width NTKs (regardless of finite NTK width). Results for intermediate NTK widths are shown in Appendix D. Another desirable feature which we observe in the MNIST setting is that the NTK and CNN perform on-par in the presence of zero noise.

## 7  Conclusion

We have considered a simple, noisy data distribution that captures some of the key structure seen in natural images. We have (1) provably shown that a particular two-layer CNN trained via SGD can *efficiently* (in time and sample complexity) achieve high accuracy and (2) that its corresponding linear model (the finite-width NTK) requires a larger network size (i.e., more features). Overall, our results shed light on a new mechanism through which neural networks are provably better than their corresponding kernels: in particular, when there is a signal hidden within background noise, a neural network is able to simultaneously adapt to the local signal and perform "denoising". We provide empirical justification showing that our theoretical framework does coincide with the superior power of neural networks over linear learners in practice.

One avenue for future work involves extending the noise distribution: increasing $\sigma$ beyond $\log k/\sqrt{k}$, which can provide an even stronger separation with linear learners, and extending to other noise models beyond Gaussian. Another possible extension is to drop the restriction that the same filter is used in both parts of the activation function and study whether this soft-thresholding behavior is recovered automatically upon training the final-layer weights. We could also consider more general models in which the signal "patch" and the CNN filter are not perfectly aligned. Another possibility is to develop a limitation result for all kernel methods instead of the CNTK corresponding to our CNN; however, as discussed in Section 2, this would likely be a regression result instead of a classification result, which is weaker in other ways. Finally, it would be interesting to extend this theoretical analysis to deeper networks and thus more practical CNNs (perhaps "hierarchical denoising").

*[Ethics statement]* Our work is primarily theoretical in nature and analyzes existing methods; thus, to the best of our knowledge, it does not have any negative societal impact.

## Acknowledgments & Funding Disclosure

The authors thank Zico Kolter for conversations about the experiments and Jaehoon Lee and Ben Adlam for their help in navigating the neural-tangents API. The authors also thank the anonymous NeurIPS reviewers for their thoughtful, helpful feedback. YL acknowledges support from NSF RI 2007517.

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
