# A Full proofs for Section 4

## A.1 Simplifying calculations and notation

We introduce the following calculations and notation to simplify the subsequent proofs.

Let $I = [k+1] \setminus \{i^\star\}$.

For any given $t$, we decompose $\mathbf{w}^{(t)} = (\mathbf{w}^{(t)} \cdot \mathbf{w}_\star)\mathbf{w}_\star + \mathbf{w}_\perp^{(t)}$.

We let $c_t := \mathbf{w}^{(t)} \cdot \mathbf{w}_\star$ and $v_t := \|\mathbf{w}_\perp^{(t)}\|_2$. For $i \in I$, we define $A_i := X_i \cdot \mathbf{w}_\star$ and $B_i := X_i \cdot \mathbf{w}_\perp^{(t)}/\|\mathbf{w}_\perp^{(t)}\|_2$. Since $X_i$ is a spherically symmetric Gaussian vector and $\mathbf{w}_\star$ and $\mathbf{w}_\perp^{(t)}/\|\mathbf{w}_\perp^{(t)}\|_2$ define two fixed (per $t$) orthogonal directions, we have

$$A_i, B_i \overset{i.i.d.}{\sim} \mathcal{N}(0, \sigma^2).$$

Finally, we define $Z_i := c_t A_i + v_t B_i \sim \mathcal{N}(0, (c_t^2 + v_t^2)\sigma^2)$ and $s_t^2 := (c_t^2 + v_t^2)\sigma^2$. With this notation, we obtain the following simplification of $f_t(X)$, used extensively throughout the proofs:

$$f_t(X) = \sum_{i=1}^{k+1} \left[ \mathrm{ReLU}(\langle \mathbf{w}_t, X_i \rangle + b_t) - \mathrm{ReLU}(-\langle \mathbf{w}_t, X_i \rangle + b_t) \right]$$

$$= \sum_{i \in I} \left[ \mathrm{ReLU}(Z_i + b_t) - \mathrm{ReLU}(-Z_i + b_t) \right] + \mathrm{ReLU}(c_t Y + b_t) - \mathrm{ReLU}(-c_t Y + b_t).$$

The population gradients then simplify as follows:

$$\nabla_{\mathbf{w}} \mathbb{E}[\ell(f_t(X), y)] = \mathbb{E}\left[ -Y\sigma(-Yf_t(X)) \sum_{i \in I} A_i \left[ \mathbb{I}\{Z_i \geq -b_t\} + \mathbb{I}\{Z_i \leq b_t\} \right] \right] \mathbf{w}_\star$$

$$+ \mathbb{E}\left[ -\sigma(-Yf_t(X)) \left[ \mathbb{I}\{c_t Y \geq -b_t\} + \mathbb{I}\{c_t Y \leq b_t\} \right] \right] \mathbf{w}_\star$$

$$+ \mathbb{E}\left[ -Y\sigma(-Yf_t(X)) \sum_{i \in I} B_i \left[ \mathbb{I}\{Z_i \geq -b_t\} + \mathbb{I}\{Z_i \leq b_t\} \right] \right] \mathbf{w}_\perp/\|\mathbf{w}_\perp\|_2.$$

$$\nabla_b \mathbb{E}[\ell(f_t(X), y)] = \mathbb{E}\left[ -Y\sigma(-Yf_t(X)) \sum_{i \in I} \left[ \mathbb{I}\{Z_i \geq -b_t\} - \mathbb{I}\{Z_i \leq b_t\} \right] \right]$$

$$+ \mathbb{E}\left[ -Y\sigma(-Yf_t(X)) \left[ \mathbb{I}\{c_t Y \geq -b_t\} - \mathbb{I}\{c_t Y \leq b_t\} \right] \right].$$

We define the following variants of the sum over $I$:

$$S := \sum_{i \in I} \left[ \mathrm{ReLU}(Z_i + b_t) - \mathrm{ReLU}(-Z_i + b_t) \right]$$

$$S_i := \mathrm{ReLU}(Z_i + b_t) - \mathrm{ReLU}(-Z_i + b_t)$$

$$S_{-i} := \sum_{j \in I \setminus \{i\}} \left[ \mathrm{ReLU}(Z_j + b_t) - \mathrm{ReLU}(-Z_j + b_t) \right]$$

Thus, for any $i \in I$, $S = S_{-i} + S_i$.

We further define:

$$f_{-i}(X) := f_t(X) - S_i = S_{-i} + \mathrm{ReLU}(c_t Y + b_t) - \mathrm{ReLU}(-c_t Y + b_t).$$

We define $\alpha_t := |b_t|/s_t$.

## A.2 Frequently-used Gaussian facts

Following common notation, we let $\phi(x)$ denote the standard normal PDF, $\Phi(x)$ denote the standard normal CDF, and $\Phi^c(x)$ denote the standard normal complementary CDF.

Using the notation defined in Section A.1, we have:

$$\mathbb{E}[Z_i \mid Z_i \geq -b_t] = s_t \frac{\phi(\alpha_t)}{\Phi^c(\alpha_t)} = \sqrt{c_t^2 + v_t^2}\, \sigma \frac{\phi(\alpha_t)}{\Phi^c(\alpha_t)}. \tag{1}$$

$$\mathrm{Var}[Z_i \mid Z_i \geq -b_t] = s_t^2 \left[ 1 + \alpha_t \frac{\phi(\alpha_t)}{\Phi^c(\alpha_t)} - \left( \frac{\phi(\alpha_t)}{\Phi^c(\alpha_t)} \right)^2 \right]. \tag{2}$$

$$\mathbb{E}[Z_i^2 \mid Z_i \geq -b_t] = s_t^2 \left[ 1 + \alpha_t \frac{\phi(\alpha_t)}{\Phi^c(\alpha_t)} \right] = (c_t^2 + v_t^2)\sigma^2 \left[ 1 + \alpha_t \frac{\phi(\alpha_t)}{\Phi^c(\alpha_t)} \right]. \tag{3}$$

$$\mathbb{E}[c_t A_i \mid Z_i] = Z_i \frac{c_t^2 \sigma^2}{c_t^2 \sigma^2 + v_t^2 \sigma^2} = Z_i \frac{c_t^2}{c_t^2 + v_t^2}. \tag{4}$$

$$\mathbb{E}[v_t B_i \mid Z_i] = Z_i \frac{v_t^2 \sigma^2}{v_t^2 \sigma^2 + c_t^2 \sigma^2} = Z_i \frac{v_t^2}{c_t^2 + v_t^2}. \tag{5}$$

We use the following (relatively tight) bounds on Mills ratio $\frac{\Phi^c(x)}{\phi(x)}$, for $x \geq 0$:

$$\frac{2}{\sqrt{x^2 + 4} + x} \leq \frac{\Phi^c(x)}{\phi(x)} \leq \frac{2}{\sqrt{x^2 + 2} + x}. \tag{6}$$

**Lemma 4.** *Let $X_1, \ldots, X_n$ be i.i.d. $\mathcal{N}(0, \sigma^2)$ RVs. Then $\mathbb{P}\left[ \max_{j \leq n} X_j \geq 2\sigma\sqrt{\log n} \right] \leq 1/n$.*

*Proof.* For all $t > 0$, $\mathbb{P}[\max_j X_j \geq t] \leq \sum_j \mathbb{P}[X_j \geq t] \leq n \cdot \exp\left( -\frac{t^2}{2\sigma^2} \right)$. Now let $t = 2\sigma\sqrt{\log n}$.

Then $\mathbb{P}[\max_j X_j \geq t] \leq n \exp\left( -\frac{4\sigma^2 \log n}{2\sigma^2} \right) = n \exp(-2\log n) = 1/n$. $\qquad\square$

**Lemma 5.** *Let $X_1, \ldots, X_n$ be i.i.d. $\mathcal{N}(0, \sigma^2)$ RVs. Then $\mathbb{P}\left[ \max_{j \leq n} |X_j| \geq 2\sigma\sqrt{\log n} \right] \leq 2/n$.*

*Proof.* For all $t > 0$, $\mathbb{P}[\max_j |X_j| \geq t] \leq \sum_j \mathbb{P}[|X_j| \geq t] \leq n \cdot 2\exp\left( -\frac{t^2}{2\sigma^2} \right)$. Now let $t = 2\sigma\sqrt{\log n}$.

Then $\mathbb{P}[\max_j |X_j| \geq t] \leq 2n \exp\left( -\frac{4\sigma^2 \log n}{2\sigma^2} \right) = 2n \exp(-2\log n) = 2/n$. $\qquad\square$

### A.3 Neural network upper bound proofs

#### A.3.1 Training invariants

**Lemma 6** (Initialization). *Initialize* $\mathbf{w}^{(0)} \sim \mathcal{N}\left(0, \frac{\delta^2}{kd} I_{d \times d}\right)$. *Then with probability at least* $1 - e^{-\Omega(k)}$, *we have* $|c^{(0)}|, v^{(0)} \le \delta$.

*Proof.* Let $E$ denote the event that, $\forall i \in [d]$, $w_i^{(0)} \le \frac{\delta}{\sqrt{d}}$. Then via a Gaussian tail bound and union bound, we have

$$\mathbb{P}[\neg E] \le \sum_{i \in [d]} \mathbb{P}\left[|w_i^{(0)}| \ge \frac{\delta}{\sqrt{d}}\right] \le d \cdot 2e^{-k/2}.$$

$E$ implies $\|w^{(0)}\|_2 \le \delta$, so $|c^{(0)}|, v^{(0)} \le \delta$. $\qquad\qquad\square$

**Remark 1.** *We choose* $\delta = 1/k^{100}$. *Thus,* $\mathbf{w}^{(0)} \sim \mathcal{N}\left(0, \sigma_0^2 I_{d \times d}\right)$, *where* $\sigma_0$ *is* $1/poly(k)$. *We choose* $\eta$ *so that* $\eta/\delta = \Omega(k)$.

**Lemma 7** (Stochastic gradients). *Using* $n = poly(k)$ *samples per mini-batch, at any given time step* $t \le T$, *with probability at least* $1 - e^{-\widetilde{\Omega}(k)}$, *we have:*

$$\left\| \frac{1}{n} \sum_{(X,y) \in \mathcal{Z}} \nabla_{\mathbf{w}} \ell(f_t(X), y) - \mathbb{E}\left[\nabla_{\mathbf{w}} \ell(f_t(X), y)\right] \right\|_2 \le \frac{\delta}{\eta poly(k)}.$$

$$\left| \frac{1}{n} \sum_{(X,y) \in \mathcal{Z}} \nabla_b \ell(f_t(X), y) - \mathbb{E}\left[\nabla_b \ell(f_t(X), y)\right] \right| \le \frac{1}{k}.$$

*Proof.* Consider a mini-batch of $n$ examples $\mathcal{Z} \sim \mathcal{D}^n$, where $n = poly(k)$.

For some $(X, y)$, $j \in [d]$, let $E_j$ be the event that $|X_{i,j}| \le 1$ for all $i \in I$ and $g_j := (\nabla_{\mathbf{w}} \ell(f(X), y))_j$.

$$\mathbb{E}[g_j] = \mathbb{E}[g_j \mid E_j]\mathbb{P}[E_j] + \mathbb{E}[g_j \mid \neg E_j]\mathbb{P}[\neg E_j]$$
$$= \mathbb{E}[g_j \mid E_j]\left(1 - e^{-\widetilde{\Omega}(k)}\right) + \mathbb{E}[g_j \mid \neg E_j]e^{-\widetilde{\Omega}(k)}$$
$$\mathbb{E}[g_j] - \mathbb{E}[g_j \mid E_j] = e^{-\widetilde{\Omega}(k)}\left(\mathbb{E}[g_j \mid \neg E_j] - \mathbb{E}[g_j \mid E_j]\right)$$
$$|\mathbb{E}[g_j] - \mathbb{E}[g_j \mid E_j]| \le e^{-\widetilde{\Omega}(k)}\mathcal{O}(k)$$
$$|\mathbb{E}[g_j] - \mathbb{E}[g_j \mid E_j]| \le e^{-\widetilde{\Omega}(k)}.$$

Let $\widehat{E}_j$ be the event that $|X_{i,j}| \le 1$ for all $i \in I$, for all $(X, y) \in \mathcal{Z}$. For any $j \in [d]$, $t > 0$, we have:

$$\mathbb{P}\left[\left| \frac{1}{n} \sum_{(X,y) \in \mathcal{Z}} (\nabla_{\mathbf{w}} \ell(f(X), y))_j - \mathbb{E}[(\nabla_{\mathbf{w}} \ell(f(X), y))_j] \right| \ge t\right]$$

$$= \mathbb{P}\left[\left| \frac{1}{n} \sum_{(X,y) \in \mathcal{Z}} (\nabla_{\mathbf{w}} \ell(f(X), y))_j - \mathbb{E}[(\nabla_{\mathbf{w}} \ell(f(X), y))_j] \right| \ge t \,\middle|\, \widehat{E}_j\right] \mathbb{P}[\widehat{E}_j]$$

$$+ \mathbb{P}\left[\left| \frac{1}{n} \sum_{(X,y) \in \mathcal{Z}} (\nabla_{\mathbf{w}} \ell(f(X), y))_j - \mathbb{E}[(\nabla_{\mathbf{w}} \ell(f(X), y))_j] \right| \ge t \,\middle|\, \neg\widehat{E}_j\right] \mathbb{P}[\neg\widehat{E}_j]$$

$$\le \mathbb{P}\left[\left| \frac{1}{n} \sum_{(X,y) \in \mathcal{Z}} (\nabla_{\mathbf{w}} \ell(f(X), y))_j - \mathbb{E}[(\nabla_{\mathbf{w}} \ell(f(X), y))_j] \right| \ge t \,\middle|\, \widehat{E}_j\right] + e^{-\widetilde{\Omega}(k)}$$

$$\leq \mathbb{P}\left[\left.\left|\frac{1}{n}\sum_{(X,y)\in\mathcal{Z}}(\nabla_{\mathbf{w}}\ell(f(X),y))_j - \mathbb{E}[g_j\mid E_j]\right| + \left|\mathbb{E}[g_j\mid E_j] - \mathbb{E}[g_j]\right| \geq t \;\right| \widehat{E}_j\right] + e^{-\widetilde{\Omega}(k)}$$

$$\leq \mathbb{P}\left[\left.\left|\frac{1}{n}\sum_{(X,y)\in\mathcal{Z}}(\nabla_{\mathbf{w}}\ell(f(X),y))_j - \mathbb{E}[g_j\mid E_j]\right| \geq t - \left|\mathbb{E}[g_j\mid E_j] - \mathbb{E}[g_j]\right| \;\right| \widehat{E}_j\right] + e^{-\widetilde{\Omega}(k)}$$

We can now apply a Hoeffding bound with $t = \frac{\delta}{\eta\sqrt{d}\mathrm{poly}(k)}$. For sufficiently large $n = \mathrm{poly}(k)$, we obtain:

$$\mathbb{P}\left[\left|\frac{1}{n}\sum_{(X,y)\in\mathcal{Z}}(\nabla_{\mathbf{w}}\ell(f(X),y))_j - \mathbb{E}[(\nabla_{\mathbf{w}}\ell(f(X),y))_j]\right| \geq t\right] \leq e^{-\widetilde{\Omega}(k)}.$$

Finally, with probability at least $1 - e^{-\widetilde{\Omega}(k)}$, we have:

$$\left\|\frac{1}{n}\sum_{(X,y)\in\mathcal{Z}}\nabla_{\mathbf{w}}\ell(f(X),y) - \mathbb{E}\left[\nabla_{\mathbf{w}}\ell(f(X),y)\right]\right\|_2 \leq \frac{\delta}{\eta\mathrm{poly}(k)}.$$

For all $(X,y) \in \mathbb{R}^{(k+1)\times d} \times \{-1,1\}$, we have $|\nabla_b\ell(f(X),y)| \leq \mathrm{poly}(k)$ (deterministically).

Then, by a Hoeffding bound for sufficiently large $n = \mathrm{poly}(k)$, with probability at least $1 - e^{-\widetilde{\Omega}(k)}$, we have:

$$\left|\frac{1}{n}\sum_{(X,y)\in\mathcal{Z}}\nabla_b\ell(f(X),y) - \mathbb{E}\left[\nabla_b\ell(f(X),y)\right]\right| \leq \frac{1}{k}.$$

$\square$

**Lemma 8.** *There exists an absolute constant $k_0$ such that, for every $k \geq k_0$, we have $\alpha_t \leq 2\sqrt{\log k}$ for all $t$ such that $\mathbb{P}_{(X,y)\sim\mathcal{D}}[\mathrm{sign}(f_t(X)) \neq y] \geq 0.01$.*

*Proof.* Recall that we have defined $\alpha_t := |b_t|/s_t$ in Section A.1.

At the start of training, when the derivatives of $b$ and $c$ are each coming from the signal patch and thus both $b$ and $c$ are increasing, the ratio $b/c$ is determined by $\eta_b/\eta_{\mathbf{w}} = 1/k$.

Once $b$ starts to decrease, we consider $\alpha_t$ for $b_t < 0$ and $c_t > 0$. We prove this case by contradiction. Suppose $\alpha_t > 2\sqrt{\log k}$. Then we have:

$$\mathbb{P}_{(X,y)\sim\mathcal{D}}[\mathrm{sign}(f_t(X)) \neq y] \leq \mathbb{P}[\max_i |Z_i| \geq |b_t|] \leq \sum_{i\in I}\mathbb{P}[|Z_i| \geq |b_t|] \leq k \cdot 2e^{-\alpha_t^2/2} < 2/k.$$

For sufficiently large $k$, we obtain $\mathbb{P}_{(X,y)\sim\mathcal{D}}[\mathrm{sign}(f_t(X)) \neq y] < 0.01$, a contradiction. Thus, we must have $\alpha_t \leq 2\sqrt{\log k}$. $\square$

**Lemma 9.** *There exists a constant $\beta > 0$ and a constant $k_0$ such that, for every $k \geq k_0$, the following holds for all $t \leq T$ such that $\mathbb{P}_{(X,y)\sim\mathcal{D}}[\mathrm{sign}(f_t(X)) \neq y] \geq 0.01$:*

$$\mathrm{Var}[S_i \mid |S_i| \leq s_t\log^2 k] \geq \beta(c_t + b_t)^2/k \;\; \forall i \in I.$$

*Proof.* For any $M > 0$, let $E_M$ be the event that $|S_i| \leq M \;\; \forall i \in I$. For any $t > 0$, we then have:

$$\mathbb{P}\left[\sum_{i\in I} S_i \geq t\right] = \mathbb{P}\left[\left.\sum_{i\in I} S_i \geq t\;\right| E_M\right]\mathbb{P}[E_M] + \mathbb{P}\left[\left.\sum_{i\in I} S_i \geq t\;\right| \neg E_M\right]\mathbb{P}[\neg E_M]$$

$$\leq \mathbb{P}\left[\sum_{i \in I} S_i \geq t \;\middle|\; E_M\right] + \mathbb{P}[\neg E_M]$$

$$\leq \mathbb{P}\left[\sum_{i \in I} S_i \geq t \;\middle|\; E_M\right] + k\mathbb{P}[|S_1| > M]$$

$$\leq \exp\left(-\frac{t^2}{2k\operatorname{Var}[S_1 \mid |S_1| \leq M] + 2Mt/3}\right) + k\mathbb{P}[|Z_1| > M - b]$$

$$\leq \exp\left(-\frac{t^2}{2k\operatorname{Var}[S_1 \mid |S_1| \leq M] + 2Mt/3}\right) + 2k\exp\left(-\frac{(M-b)^2}{2s_t^2}\right),$$

by applying a Bernstein inequality.

Choosing $M = s_t \log^2 k$ and $t = c_t + b_t$ and defining $u_t^2 := \operatorname{Var}[S_1 \mid |S_1| \leq M]$, we obtain:

$$\mathbb{P}\left[\sum_{i \in I} S_i \geq c_t + b_t\right] \leq \exp\left(-\frac{(c_t + b_t)^2}{2ku_t^2 + 2s_t \log^2 k(c_t + b_t)/3}\right) + 2k\exp\left(-\frac{(s_t \log^2 k - b_t)^2}{2s_t^2}\right).$$

$2k\exp\left(-\frac{(s_t \log^2 k - b_t)^2}{2s_t^2}\right) = e^{-\Omega(\log^4 k)}$, and $2s_t \log^2 k(c_t + b_t)/3 = \mathcal{O}(c_t^2 \log^3 k/\sqrt{k})$. Therefore, unless $u_t^2 \geq \beta(c_t + b_t)^2/k$ for some constant $\beta > 0$, we will have $\mathbb{P}\left[\sum_{i \in I} S_i \geq c_t + b_t\right] < 0.01$ for sufficiently large $k$.

$$\square$$

**Corollary 1.** *There exists a constant $\beta > 0$ and a constant $k_0$ such that, for every $k \geq k_0$, the following holds for all $t \leq T$ such that $\mathbb{P}_{(X,y)\sim\mathcal{D}}[\operatorname{sign}(f_t(X)) \neq y] \geq 0.01$ and all $i \in I$:*

$$\operatorname{Var}[S_i] \geq \beta\frac{(c_t + b_t)^2}{k}.$$

### A.3.2 Function value stays $\mathcal{O}(1)$ with $\Omega(1)$ probability

**Lemma 10.** *There exists a constant $C > 0$ and constant $k_0$ such that, for all $k \geq k_0$, the following holds for all $t \leq T$:*

$$c_t \leq C.$$

*Proof.* Since we are doing stochastic gradient descent with stochastic gradients very close to the population gradients, we can easily conclude that the loss cannot grow throughout training. $\mathbb{E}[\ell(f_0(X), y)] \leq 1$ at the start of training, so we must have $\mathbb{E}[\ell(f_t(X), y)] \leq 1$ for all $t \leq T$.

Let $E$ denote the event that $|S_i| \leq s_t \log^2 k \ \ \forall i \in I$.

$$
\begin{aligned}
1 &\geq \mathbb{E}[\ell(f_t(X), y)] \\
&= \mathbb{E}[\log(1 + \exp(-Y f_t(X)))] \\
&= \frac{1}{2}\mathbb{E}[\log(1 + \exp(-f_t(X))) \mid Y = 1] + \frac{1}{2}\mathbb{E}[\log(1 + \exp(f_t(X))) \mid Y = -1] \\
&\geq \frac{1}{2}\mathbb{E}[\mathrm{ReLU}(-f_t(X)) \mid Y = 1] + \frac{1}{2}\mathbb{E}[\mathrm{ReLU}(f_t(X)) \mid Y = -1] \\
&= \mathbb{E}[\mathrm{ReLU}(-f_t(X)) \mid Y = 1] \\
&= \mathbb{E}\left[\mathrm{ReLU}\left(-(c_t + b_t) - \sum_{i \in I} S_i\right)\right] \\
&\geq \mathbb{E}\left[\mathrm{ReLU}\left(-(c_t + b_t) - \sum_{i \in I} S_i\right) \ \Bigg| \ \sum_{i \in I} S_i \leq -2(c_t + b_t)\right] \mathbb{P}\left[\sum_{i \in I} S_i \leq -2(c_t + b_t)\right] \\
&\geq (c_t + b_t) \mathbb{P}\left[\sum_{i \in I} S_i \leq -2(c_t + b_t)\right] \\
&= (c_t + b_t)\left(\mathbb{P}\left[\sum_{i \in I} S_i \leq -2(c_t + b_t) \mid E\right] \mathbb{P}[E] + \mathbb{P}\left[\sum_{i \in I} S_i \leq -2(c_t + b_t) \mid \neg E\right] \mathbb{P}[\neg E]\right) \\
&\geq (c_t + b_t) \mathbb{P}\left[\sum_{i \in I} S_i \leq -2(c_t + b_t) \mid E\right] \mathbb{P}[E].
\end{aligned}
$$

Following the notation in the proof of Lemma 9, we let $u_t^2 := \mathrm{Var}[S_i \mid |S_i| \leq s_t \log^2 k]$, for any $i \in I$. Since $\mathbb{E}[S_i \mid |S_i| \leq s_t \log^2 k] = 0$, we also have $\mathbb{E}[S_i^2 \mid |S_i| \leq s_t \log^2 k] = u_t^2$.

We then use Lemma 9 and Berry-Esseen to lower bound $\mathbb{P}\left[\sum_{i \in I} S_i \leq -2(c_t + b_t) \mid E\right]$ as follows, where $C_1$ is some positive constant.

$$
\begin{aligned}
\sup_{x \in \mathbb{R}} \left| \mathbb{P}\left(\frac{\sum_{i \in I} S_i}{u_t \sqrt{k}} \leq x \ \Big| \ E\right) - \Phi(x) \right| &\leq \frac{C_1}{\sqrt{k}} \cdot \frac{\mathbb{E}[|S_i|^3 \mid |S_i| \leq s_t \log^2 k]}{u^3} \\
&\leq \frac{C_1}{\sqrt{k}} \cdot \frac{s_t \log^2 k \cdot \mathbb{E}[|S_i|^2 \mid |S_i| \leq s_t \log^2 k]}{u_t^3} \\
&= \frac{C_1}{\sqrt{k}} \cdot \frac{s_t \log^2 k}{u_t} \\
&\leq \frac{C_1 s_t \log^2 k}{\sqrt{\beta}(c_t + b_t)} \\
&\leq \frac{C_1}{2\sqrt{\beta}} \frac{\log^3 k}{\sqrt{k}},
\end{aligned}
$$

by Lemma 9, which says that $u_t \geq \sqrt{\beta} \frac{c_t + b_t}{\sqrt{k}}$ for some constant $\beta > 0$. Based on Lemma 9, we choose $x$ such that $x < 0$ and $|x| \leq 2/\sqrt{\beta}$. We obtain

$$
\mathbb{P}\left[\sum_{i \in I} S_i \leq -2(c_t + b_t) \mid E\right] \geq \Phi(x) - \frac{C_1}{2\sqrt{\beta}} \frac{\log^3 k}{\sqrt{k}} \geq \frac{1}{2}\Phi(x)
$$

for sufficiently large $k$.

By Lemma 9's proof, we know that $\mathbb{P}[E] = 1 - e^{-\Omega(\log^4 k)}$.

Putting these pieces together, we obtain $1 \geq \mathbb{E}[\ell(f_t(X), y)] = \frac{\Phi(x)}{4}(c_t + b_t)$, and by Lemma 8, we conclude that there must exist a constant $C > 0$ such that $c_t \leq C$.

$\square$

**Lemma 11.** *For all $t \leq T$, $\mathbb{P}[|S| < c_t + b_t + 10] \geq 4/5$.*

*Proof.* We closely follow the proof of Lemma 10. Since we are doing stochastic gradient descent with stochastic gradients very close to the population gradients, we can easily conclude that the loss cannot grow throughout training. $\mathbb{E}[\ell(f_0(X), y)] \leq 1$ at the start of training, so we must have $\mathbb{E}[\ell(f_t(X), y)] \leq 1$ for all $t \leq T$.

$$
\begin{aligned}
1 &\geq \mathbb{E}[\ell(f_t(X), y)] \\
&= \mathbb{E}[\log(1 + \exp(-Y f_t(X)))] \\
&= \frac{1}{2}\mathbb{E}[\log(1 + \exp(-f_t(X))) \mid Y = 1] + \frac{1}{2}\mathbb{E}[\log(1 + \exp(f_t(X))) \mid Y = -1] \\
&\geq \frac{1}{2}\mathbb{E}[\text{ReLU}(-f_t(X)) \mid Y = 1] + \frac{1}{2}\mathbb{E}[\text{ReLU}(f_t(X)) \mid Y = -1] \\
&= \mathbb{E}[\text{ReLU}(-f_t(X)) \mid Y = 1] \\
&= \mathbb{E}\left[\text{ReLU}\left(-(c_t + b_t) - \sum_{i \in I} S_i\right)\right] \\
&\geq \mathbb{E}\left[\text{ReLU}\left(-(c_t + b_t) - \sum_{i \in I} S_i\right) \,\middle|\, \sum_{i \in I} S_i \leq -(c_t + b_t) - 10\right] \mathbb{P}\left[\sum_{i \in I} S_i \leq -(c_t + b_t) - 10\right] \\
&\geq 10 \cdot \mathbb{P}\left[\sum_{i \in I} S_i \leq -(c_t + b_t) - 10\right] \\
&= 5 \cdot \mathbb{P}\left[\left|\sum_{i \in I} S_i\right| \geq (c_t + b_t) + 10\right] \\
&= 5 \cdot \left(1 - \mathbb{P}\left[\left|\sum_{i \in I} S_i\right| < (c_t + b_t) + 10\right]\right),
\end{aligned}
$$

finally yielding

$$
\mathbb{P}\left[\left|\sum_{i \in I} S_i\right| < (c_t + b_t) + 10\right] \geq 4/5.
$$

$\square$

### A.3.3   Initial growth of $c$

**Lemma 1** (Full). *For any constant $H > 0$, there exist constants $A > 0$ and $k_0$ such that, for all $k \geq k_0$, for any $t \leq T$ such that $|c_t| \leq H/\log^4 k$, we have the following with probability at least $1 - e^{-\widetilde{\Omega}(k)}$ over a mini-batch $\mathcal{Z} \sim \mathcal{D}^n$ of $n = poly(k)$ samples:*

$$\frac{1}{n} \sum_{(X,y)\in\mathcal{Z}} \nabla_{\mathbf{w}}\ell(f_t(X), y) \cdot \mathbf{w}_\star \leq -A.$$

*Proof.* We recall the following:

$$\nabla_{\mathbf{w}}\mathbb{E}[\ell(f_t(X), y)] \cdot \mathbf{w}^\star = \underbrace{\mathbb{E}\left[-Y\sigma(-Yf_t(X))\sum_{i\in I} A_i\Big[\mathbb{I}\{Z_i + b_t \geq 0\} + \mathbb{I}\{-Z_i + b_t \geq 0\}\Big]\right]}_{(1)} +$$

$$\underbrace{\mathbb{E}\left[-\sigma(-Yf(X))\Big[\mathbb{I}\{c_t Y + b_t \geq 0\} + \mathbb{I}\{-c_t Y + b_t \geq 0\}\Big]\right]}_{(2)}.$$

Let $g(Y) := \text{ReLU}\Big(c_t Y + b\Big) - \text{ReLU}\Big(-c_t Y + b\Big)$. Then $g(1) = -g(-1)$ and $f(X) = S + g(Y)$.

We first simplify $(1)$ as follows:

$$(1) = \mathbb{E}\left[-Y\sigma(-Yf_t(X))\sum_{i\in I} A_i\Big[\mathbb{I}\{Z_i \geq -b_t\} + \mathbb{I}\{Z_i \leq b_t\}\Big]\right]$$

$$= \mathbb{E}\left[-\sigma\Big(-S - g(1)\Big)\sum_{i\in I} A_i\Big[\mathbb{I}\{Z_i \geq -b_t\} + \mathbb{I}\{Z_i \leq b_t\}\Big]\right]\mathbb{P}[Y = +1]$$

$$+ \mathbb{E}\left[\sigma\Big(S + g(-1)\Big)\sum_{i\in I} A_i\Big[\mathbb{I}\{Z_i \geq -b_t\} + \mathbb{I}\{Z_i \leq b_t\}\Big]\right]\mathbb{P}[Y = -1]$$

$$= \frac{1}{2}\sum_{i\in I}\mathbb{E}\left[-\sigma\Big(-g(1) - (S_{-i} + S_i)\Big)A_i\mathbb{I}\{Z_i \geq -b_t\}\right]$$

$$+ \frac{1}{2}\sum_{i\in I}\mathbb{E}\left[-\sigma\Big(-g(1) - (S_{-i} + S_i)\Big)A_i\mathbb{I}\{Z_i \leq b_t\}\right]$$

$$+ \frac{1}{2}\sum_{i\in I}\mathbb{E}\left[\sigma\Big(-g(1) + (S_{-i} + S_i)\Big)A_i\mathbb{I}\{Z_i \geq -b_t\}\right]$$

$$+ \frac{1}{2}\sum_{i\in I}\mathbb{E}\left[\sigma\Big(-g(1) + (S_{-i} + S_i)\Big)A_i\mathbb{I}\{Z_i \leq b_t\}\right] \qquad (g(-1) = -g(1))$$

$$= \frac{1}{2}\sum_{i\in I}\mathbb{E}\left[-\sigma\Big(-g(1) + S_{-i} - S_i\Big)A_i\mathbb{I}\{Z_i \geq -b_t\}\right]$$

$$+ \frac{1}{2}\sum_{i\in I}\mathbb{E}\left[-\sigma\Big(-g(1) + S_{-i} - S_i\Big)A_i\mathbb{I}\{Z_i \leq b_t\}\right]$$

$$+ \frac{1}{2}\sum_{i\in I}\mathbb{E}\left[\sigma\Big(-g(1) + S_{-i} + S_i\Big)A_i\mathbb{I}\{Z_i \geq -b_t\}\right]$$

$$+ \frac{1}{2}\sum_{i\in I}\mathbb{E}\left[\sigma\Big(-g(1) + S_{-i} + S_i\Big)A_i\mathbb{I}\{Z_i \leq b_t\}\right] \qquad (S_{-i} \overset{d}{=} -S_{-i})$$

$$= \frac{1}{2}\sum_{i\in I}\mathbb{E}\left[-\sigma\Big(-g(1) + S_{-i} - S_i\Big)A_i\mathbb{I}\{Z_i \geq -b_t\}\right]$$

$$+ \frac{1}{2} \sum_{i \in I} \mathbb{E}\left[\sigma\left(-g(1) + S_{-i} + S_i\right) A_i \mathbb{I}\{Z_i \geq -b_t\}\right]$$

$$+ \frac{1}{2} \sum_{i \in I} \mathbb{E}\left[\sigma\left(-g(1) + S_{-i} + S_i\right) A_i \mathbb{I}\{Z_i \geq -b_t\}\right]$$

$$+ \frac{1}{2} \sum_{i \in I} \mathbb{E}\left[-\sigma\left(-g(1) + S_{-i} - S_i\right) A_i \mathbb{I}\{Z_i \geq -b_t\}\right] \qquad (Z_i \overset{d}{=} -Z_i).$$

We then collapse the four summands above. Letting $a_2 := -g(1) + S_{-i} + S_i$, $a_1 := -g(1) + S_{-i} - S_i$, and $\Delta := a_2 - a_1 = 2S_i$, we have:

$$(1) = \sum_{i \in I} \mathbb{E}\left[\left(\sigma(a_2) - \sigma(a_1)\right) A_i \mathbb{I}\{Z_i \geq -b_t\}\right]$$

$$= \sum_{i \in I} \mathbb{E}\left[\left(\sigma(a_2) - \sigma(a_1)\right) A_i \mid Z_i \geq -b_t\right] \mathbb{P}[Z_i \geq -b_t]$$

$$= \sum_{i \in I} \mathbb{E}\left[\mathbb{E}\left[\left(\sigma(a_2) - \sigma(a_1)\right) A_i \mid Z_j \ \forall j \in I, Z_i \geq -b_t\right] \mid Z_i \geq -b_t\right] \mathbb{P}[Z_i \geq -b_t]$$

$$= \sum_{i \in I} \mathbb{E}\left[\left(\sigma(a_2) - \sigma(a_1)\right) \frac{1}{c_t} Z_i \frac{c_t^2}{c_t^2 + v_t^2} \mid Z_i \geq -b_t\right] \mathbb{P}[Z_i \geq -b_t]$$

$$= \frac{c_t}{c_t^2 + v_t^2} \sum_{i \in I} \mathbb{E}\left[\left(\sigma(a_2) - \sigma(a_1)\right) Z_i \mid Z_i \geq -b_t\right] \mathbb{P}[Z_i \geq -b_t].$$

We first note that $(\sigma(a_2) - \sigma(a_1))Z_i \geq 0$ for all instantiations of the random variables $Z_i, i \in I$. This is because $\sigma$ is monotonically increasing and $\text{sign}(S_i) = \text{sign}(Z_i)$.

Then, using the fact that $\sigma$ is $1/4$-Lipschitz and $|S_i| \leq \max\{2|Z_i|, |Z_i| + |b_t|\} \leq 2|Z_i| + |b_t|$, we upper bound $|(1)|$ as follows:

$$|(1)| = \frac{|c_t|}{c_t^2 + v_t^2} \sum_{i \in I} \mathbb{E}\left[\left(\sigma(a_2) - \sigma(a_1)\right) Z_i \mid Z_i \geq -b_t\right] \mathbb{P}[Z_i \geq -b_t]$$

$$\leq \frac{|c_t|}{c_t^2 + v_t^2} \sum_{i \in I} \mathbb{E}\left[\frac{1}{2} S_i Z_i \mid Z_i \geq -b_t\right] \mathbb{P}[Z_i \geq -b_t]$$

$$\leq \frac{|c_t|}{c_t^2 + v_t^2} \sum_{i \in I} \mathbb{E}\left[\frac{1}{2}(2Z_i^2 + |b_t||Z_i|) \mid Z_i \geq -b_t\right] \mathbb{P}[Z_i \geq -b_t]$$

$$= \frac{|c_t|}{c_t^2 + v_t^2} \sum_{i \in I} \left(\mathbb{E}[Z_i^2 \mid Z_i \geq -b_t] + \frac{1}{2}|b_t|\mathbb{E}[|Z_i| \mid Z_i \geq -b_t]\right) \mathbb{P}[Z_i \geq -b_t]$$

$$\leq \frac{|c_t|}{c_t^2 + v_t^2} \sum_{i \in I} \left(\mathbb{E}[Z_i^2 \mid Z_i \geq |b_t|] + \frac{1}{2}|b_t|\mathbb{E}[|Z_i| \mid Z_i \geq |b_t|]\right) \mathbb{P}[Z_i \geq -b_t]$$

$$= \frac{|c_t|}{c_t^2 + v_t^2} \sum_{i \in I} \left((c_t^2 + v_t^2)\sigma^2 \left[1 + \alpha_t \frac{\phi(\alpha_t)}{\Phi^c(\alpha_t)}\right] + \frac{1}{2}|b_t|\sqrt{c_t^2 + v_t^2}\sigma \frac{\phi(\alpha_t)}{\Phi^c(\alpha_t)}\right) \mathbb{P}[Z_i \geq -b_t]$$

$$= |c_t| \sum_{i \in I} \left(\sigma^2 \left[1 + \alpha_t \frac{\phi(\alpha_t)}{\Phi^c(\alpha_t)}\right] + \frac{1}{2}|b_t| \frac{1}{\sqrt{c_t^2 + v_t^2}} \frac{\sigma^2}{\sigma} \frac{\phi(\alpha_t)}{\Phi^c(\alpha_t)}\right) \mathbb{P}[Z_i \geq -b_t]$$

$$= |c_t| \sum_{i \in I} \left(\sigma^2 \left[1 + \alpha_t \frac{\phi(\alpha_t)}{\Phi^c(\alpha_t)}\right] + \frac{1}{2}\sigma^2 \alpha_t \frac{\phi(\alpha_t)}{\Phi^c(\alpha_t)}\right) \mathbb{P}[Z_i \geq -b_t]$$

$$= |c_t| \sum_{i \in I} \left(\sigma^2 + \frac{3}{2}\sigma^2 \alpha_t \frac{\phi(\alpha_t)}{\Phi^c(\alpha_t)}\right) \mathbb{P}[Z_i \geq -b_t].$$

Since $\alpha_t \leq 2\sqrt{\log k}$ by Lemma 8, we have $|(1)| \leq A_1 |c_t| \log^3 k$ for some constant $A_1 > 0$.

By Lemmas 10 and 11, there exists a constant $A_2 > 0$ such that $(2) \leq -A_2$.

Thus, if $|c_t| \leq H \frac{1}{\log^4 k}$ for any constant $H > 0$, then there exists a constant $A_3 > 0$ such that $(1) + (2) \leq -A_3$ for any sufficiently large $k$.

Finally, by Lemma 7, we have with probability at least $1 - e^{-\widetilde{\Omega}(k)}$,

$$\left\| \frac{1}{n} \sum_{(X,y) \in \mathcal{Z}} \nabla_{\mathbf{w}} \ell(f_t(X), y) - \mathbb{E}\left[ \nabla_{\mathbf{w}} \ell(f_t(X), y) \right] \right\|_2 \leq \frac{\delta}{\eta \mathrm{poly}(k)},$$

and therefore with probability at least $1 - e^{-\widetilde{\Omega}(k)}$,

$$\left| \frac{1}{n} \sum_{(X,y) \in \mathcal{Z}} \nabla_{\mathbf{w}} \ell(f_t(X), y) \cdot \mathbf{w}_\star - \mathbb{E}\left[ \nabla_{\mathbf{w}} \ell(f_t(X), y) \right] \cdot \mathbf{w}_\star \right| \leq \frac{\delta}{\eta \mathrm{poly}(k)}.$$

We conclude that, with probability at least $1 - e^{-\widetilde{\Omega}(k)}$, we have

$$\frac{1}{n} \sum_{(X,y) \in \mathcal{Z}} \nabla_{\mathbf{w}} \ell(f_t(X), y) \cdot \mathbf{w}_\star \leq -A_3 + \frac{\delta}{\eta \mathrm{poly}(k)} \leq -A$$

for some constant $A > 0$, for any sufficiently large $k$.

$\square$

### A.3.4 Bounded growth of $\|\mathbf{w}_\perp^{(t)}\|_2$

**Lemma 2** (Full). *For any $t \leq T$, we have the following with probability at least $1 - e^{-\widetilde{\Omega}(k)}$ over a mini-batch $\mathcal{Z} \sim \mathcal{D}^n$ of $n = poly(k)$ samples:*

$$\|\mathbf{w}_\perp^{(t)}\|_2 - \|\mathbf{w}_\perp^{(t-1)}\|_2 \leq \frac{\delta}{poly(k)}.$$

*Proof.*

$$\nabla_{\mathbf{w}}\mathbb{E}[\ell(f_t(X), y)] \cdot \mathbf{w}_\perp^{(t)}/\|\mathbf{w}_\perp^{(t)}\|_2 = \underbrace{\mathbb{E}\left[-Y\sigma(-Yf_t(X))\sum_{i \in I} B_i\Big[\mathbb{I}\{Z_i \geq -b_t\} + \mathbb{I}\{Z_i \leq b_t\}\Big]\right]}_{(1)}.$$

Let $g(Y) := \mathrm{ReLU}\Big(c_t Y + b_t\Big) - \mathrm{ReLU}\Big(-c_t Y + b_t\Big)$. Then $g(1) = -g(-1)$ and $f_t(X) = S + g(Y)$.

We simplify (1) as follows:

$$\begin{aligned}
(1) &= \mathbb{E}\left[-Y\sigma(-Yf_t(X))\sum_{i \in I} B_i\Big[\mathbb{I}\{Z_i \geq -b_t\} + \mathbb{I}\{Z_i \leq b_t\}\Big]\right] \\
&= \mathbb{E}\left[-\sigma\Big(-S - g(1)\Big)\sum_{i \in I} B_i\Big[\mathbb{I}\{Z_i \geq -b_t\} + \mathbb{I}\{Z_i \leq b_t\}\Big]\right]\mathbb{P}[Y = +1] \\
&\quad + \mathbb{E}\left[\sigma\Big(S + g(-1)\Big)\sum_{i \in I} B_i\Big[\mathbb{I}\{Z_i \geq -b_t\} + \mathbb{I}\{Z_i \leq b_t\}\Big]\right]\mathbb{P}[Y = -1] \\
&= \frac{1}{2}\sum_{i \in I}\mathbb{E}\left[-\sigma\Big(-g(1) - (S_{-i} + S_i)\Big)B_i\mathbb{I}\{Z_i \geq -b_t\}\right] \\
&\quad + \frac{1}{2}\sum_{i \in I}\mathbb{E}\left[-\sigma\Big(-g(1) - (S_{-i} + S_i)\Big)B_i\mathbb{I}\{Z_i \leq b_t\}\right] \\
&\quad + \frac{1}{2}\sum_{i \in I}\mathbb{E}\left[\sigma\Big(-g(1) + (S_{-i} + S_i)\Big)B_i\mathbb{I}\{Z_i \geq -b_t\}\right] \\
&\quad + \frac{1}{2}\sum_{i \in I}\mathbb{E}\left[\sigma\Big(-g(1) + (S_{-i} + S_i)\Big)B_i\mathbb{I}\{Z_i \leq b_t\}\right] \qquad (g(-1) = -g(1)) \\
&= \frac{1}{2}\sum_{i \in I}\mathbb{E}\left[-\sigma\Big(-g(1) + S_{-i} - S_i\Big)B_i\mathbb{I}\{Z_i \geq -b_t\}\right] \\
&\quad + \frac{1}{2}\sum_{i \in I}\mathbb{E}\left[-\sigma\Big(-g(1) + S_{-i} - S_i\Big)B_i\mathbb{I}\{Z_i \leq b_t\}\right] \\
&\quad + \frac{1}{2}\sum_{i \in I}\mathbb{E}\left[\sigma\Big(-g(1) + S_{-i} + S_i\Big)B_i\mathbb{I}\{Z_i \geq -b_t\}\right] \\
&\quad + \frac{1}{2}\sum_{i \in I}\mathbb{E}\left[\sigma\Big(-g(1) + S_{-i} + S_i\Big)B_i\mathbb{I}\{Z_i \leq b_t\}\right] \qquad (S_{-i} \overset{d}{=} -S_{-i}) \\
&= \frac{1}{2}\sum_{i \in I}\mathbb{E}\left[-\sigma\Big(-g(1) + S_{-i} - S_i\Big)B_i\mathbb{I}\{Z_i \geq -b_t\}\right] \\
&\quad + \frac{1}{2}\sum_{i \in I}\mathbb{E}\left[\sigma\Big(-g(1) + S_{-i} + S_i\Big)B_i\mathbb{I}\{Z_i \geq -b_t\}\right] \\
&\quad + \frac{1}{2}\sum_{i \in I}\mathbb{E}\left[\sigma\Big(-g(1) + S_{-i} + S_i\Big)B_i\mathbb{I}\{Z_i \geq -b_t\}\right]
\end{aligned}$$

$$+ \frac{1}{2} \sum_{i \in I} \mathbb{E}\left[ -\sigma\left( -g(1) + S_{-i} - S_i \right) B_i \mathbb{I}\{Z_i \geq -b_t\} \right] \qquad\qquad (Z_i \stackrel{d}{=} -Z_i).$$

We then collapse the four summands above. Letting $a_2 := -g(1) + S_{-i} + S_i$, $a_1 := -g(1) + S_{-i} - S_i$, and $\Delta := a_2 - a_1 = 2S_i$, we have:

$$\begin{aligned}
(1) &= \sum_{i \in I} \mathbb{E}\left[ \left( \sigma(a_2) - \sigma(a_1) \right) B_i \mathbb{I}\{Z_i \geq -b_t\} \right] \\
&= \sum_{i \in I} \mathbb{E}\left[ \left( \sigma(a_2) - \sigma(a_1) \right) B_i \mid Z_i \geq -b_t \right] \mathbb{P}[Z_i \geq -b_t] \\
&= \sum_{i \in I} \mathbb{E}\left[ \mathbb{E}\left[ \left( \sigma(a_2) - \sigma(a_1) \right) B_i \mid Z_j \; \forall j \in I, Z_i \geq -b_t \right] \mid Z_i \geq -b_t \right] \mathbb{P}[Z_i \geq -b_t] \\
&= \sum_{i \in I} \mathbb{E}\left[ \left( \sigma(a_2) - \sigma(a_1) \right) \frac{1}{v_t} Z_i \frac{v_t^2}{c_t^2 + v_t^2} \mid Z_i \geq -b_t \right] \mathbb{P}[Z_i \geq -b_t] \\
&= \frac{v_t}{c_t^2 + v_t^2} \sum_{i \in I} \mathbb{E}\left[ \left( \sigma(a_2) - \sigma(a_1) \right) Z_i \mid Z_i \geq -b_t \right] \mathbb{P}[Z_i \geq -b_t].
\end{aligned}$$

We first note that $(\sigma(a_2) - \sigma(a_1)) Z_i \geq 0$ for all instantiations of the random variables $Z_i, i \in I$. This is because $\sigma$ is monotonically increasing and $\text{sign}(S_i) = \text{sign}(Z_i)$. Then, using the fact that $\sigma$ is $1/4$-Lipschitz and $|S_i| \leq \max\{2|Z_i|, |Z_i| + |b_t|\} \leq 2|Z_i| + |b_t|$, we upper bound (1) as follows:

$$\begin{aligned}
(1) &= \frac{v_t}{c_t^2 + v_t^2} \sum_{i \in I} \mathbb{E}\left[ \left( \sigma(a_2) - \sigma(a_1) \right) Z_i \mid Z_i \geq -b_t \right] \mathbb{P}[Z_i \geq -b_t] \\
&\leq \frac{v_t}{c_t^2 + v_t^2} \sum_{i \in I} \mathbb{E}\left[ \frac{1}{2} S_i Z_i \mid Z_i \geq -b_t \right] \mathbb{P}[Z_i \geq -b_t] \\
&\leq \frac{v_t}{c_t^2 + v_t^2} \sum_{i \in I} \mathbb{E}\left[ \frac{1}{2} (2 Z_i^2 + |b_t||Z_i|) \mid Z_i \geq -b_t \right] \mathbb{P}[Z_i \geq -b_t] \\
&= \frac{v_t}{c_t^2 + v_t^2} \sum_{i \in I} \left( \mathbb{E}[Z_i^2 \mid Z_i \geq -b_t] + \frac{1}{2}|b_t|\mathbb{E}[|Z_i| \mid Z_i \geq -b_t] \right) \mathbb{P}[Z_i \geq -b_t] \\
&\leq \frac{v_t}{c_t^2 + v_t^2} \sum_{i \in I} \left( \mathbb{E}[Z_i^2 \mid Z_i \geq |b_t|] + \frac{1}{2}|b_t|\mathbb{E}[|Z_i| \mid Z_i \geq |b_t|] \right) \mathbb{P}[Z_i \geq -b_t] \\
&= \frac{v_t}{c_t^2 + v_t^2} \sum_{i \in I} \left( (c_t^2 + v_t^2)\sigma^2 \left[ 1 + \alpha_t \frac{\phi(\alpha_t)}{\Phi^c(\alpha_t)} \right] + \frac{1}{2}|b_t|\sqrt{c_t^2 + v_t^2}\sigma \frac{\phi(\alpha_t)}{\Phi^c(\alpha_t)} \right) \mathbb{P}[Z_i \geq -b_t] \\
&= v_t \sum_{i \in I} \left( \sigma^2 \left[ 1 + \alpha_t \frac{\phi(\alpha_t)}{\Phi^c(\alpha_t)} \right] + \frac{1}{2}|b_t| \frac{1}{\sqrt{c_t^2 + v_t^2}} \frac{\sigma^2}{\sigma} \frac{\phi(\alpha_t)}{\Phi^c(\alpha_t)} \right) \mathbb{P}[Z_i \geq -b_t] \\
&= v_t \sum_{i \in I} \left( \sigma^2 \left[ 1 + \alpha_t \frac{\phi(\alpha_t)}{\Phi^c(\alpha_t)} \right] + \frac{1}{2}\sigma^2 \alpha_t \frac{\phi(\alpha_t)}{\Phi^c(\alpha_t)} \right) \mathbb{P}[Z_i \geq -b_t] \\
&= v_t \sum_{i \in I} \left( \sigma^2 + \frac{3}{2}\sigma^2 \alpha_t \frac{\phi(\alpha_t)}{\Phi^c(\alpha_t)} \right) \mathbb{P}[Z_i \geq -b_t].
\end{aligned}$$

Since $\alpha_t \leq 2\sqrt{\log k}$ by Lemma 8, we have $(1) \leq A v_t \log^3 k$ for some constant $A > 0$. On its own, this would cause $v_t$ to decrease toward 0. Finally, by Lemma 7, with probability at least $1 - e^{-\widetilde{\Omega}(k)}$:

$$\left\| \frac{1}{n} \sum_{(X,y) \in \mathcal{Z}} \nabla_{\mathbf{w}} \ell(f(X), y) - \mathbb{E}\left[ \nabla_{\mathbf{w}} \ell(f(X), y) \right] \right\|_2 \leq \frac{\delta}{\eta \text{poly}(k)}.$$

Thus, for any $t$, $\|\mathbf{w}_\perp^{(t+1)}\|_2 - \|\mathbf{w}_\perp^{(t)}\|_2 \leq \frac{\delta}{\text{poly}(k)}$.

$\square$

### A.3.5  $b$ decreasing

**Lemma 12.** *There exists a constant $C_1 > 0$ and a constant $k_0$ such that, for all $k \geq k_0$, we have the following for all $t \leq T$ for which $b_t < 0$ and for all $i \in I$:*

$$\mathbb{E}[|S_i|] \geq C_1 \sqrt{\frac{\mathrm{Var}[S_i]}{k^{3/4}}}.$$

*Proof.* Consider an arbitrary $t \leq T$.

The probability of misclassification is $\mathbb{P}[S > c_t + b_t]$.

Let $B$ denote the event that $\forall i \in I$, $Z_i \leq 2s_t \sqrt{\log k}$. By Lemma 4, $\mathbb{P}[\neg B] \leq 1/k$.

Let $M = \sum_{i \in I} \mathbb{I}\{Z_i \geq -b_t\}$.

We first note that, for sufficiently large $k$, we have $\mathbb{P}[S > c_t + b_t \mid B, M < k^{1/4}] = 0$. This is because, conditioned on both $B$ and $M < k^{1/4}$, we have:

$$S < k^{1/4} \cdot 2s_t \sqrt{\log k}$$
$$= \frac{2k^{1/4}\sqrt{c_t^2 + v_t^2}\log^{3/2} k}{\sqrt{k}}$$
$$= \frac{2\sqrt{c_t^2 + v_t^2}\log^{3/2} k}{k^{1/4}}$$
$$< \frac{1}{2}c_t$$
$$< c_t + b_t,$$

by Lemmas 2 and 8, for sufficiently large $k$.

Let $P = \mathbb{P}[Z_i \geq -b_t]$ and let $P' = \mathbb{P}[Z_i \geq -b_t \mid B]$, for any $i \in I$. We note that $\frac{1}{2}\mathbb{P}[S_i \neq 0] = P \geq P'$, and therefore $\mathbb{P}[S_i \neq 0] \geq 2P'$.

We then upper bound $\mathbb{P}[S > c_t + b_t]$ as follows:

$$\begin{aligned}
0.01 \leq \mathbb{P}[S > c_t + b_t] &= \mathbb{P}[S > c_t + b_t \mid B]\mathbb{P}[B] + \mathbb{P}[S > c_t + b_t \mid \neg B]\mathbb{P}[\neg B] \\
&\leq \mathbb{P}[S > c_t + b_t \mid B] + \mathbb{P}[\neg B] \\
&= \mathbb{P}[S > c_t + b_t \mid B, M < k^{1/4}]\mathbb{P}[M < k^{1/4} \mid B] \\
&\quad + \mathbb{P}[S > c_t + b_t \mid B, M \geq k^{1/4}]\mathbb{P}[M \geq k^{1/4} \mid B] + \mathbb{P}[\neg B] \\
&= \mathbb{P}[S > c_t + b_t \mid B, M \geq k^{1/4}]\mathbb{P}[M \geq k^{1/4} \mid B] + \mathbb{P}[\neg B] \\
&\leq \mathbb{P}[M \geq k^{1/4} \mid B] + \mathbb{P}[\neg B] \\
&\leq \mathbb{P}[M \geq k^{1/4} \mid B] + 1/k.
\end{aligned}$$

We therefore have $\mathbb{P}[M \geq k^{1/4} \mid B] \geq 0.01 - 1/k \geq 0.005$ for sufficiently large $k$. So, via Chernoff, we conclude that there exists a constant $C_6 > 0$ such that $P' \geq C_6 \frac{1}{k^{3/4}}$ and thus there exists a constant $C_5 > 0$ such that $\mathbb{P}[S_i \neq 0] \geq C_5 \frac{1}{k^{3/4}}$.

Via Equations 1 and 2, we have

$$\frac{\mathbb{E}[|S_i|]}{\sqrt{\mathrm{Var}[S_i]}} \geq C_4'\sqrt{\Phi^c(\alpha_t)} = C_4\sqrt{\mathbb{P}[S_i \neq 0]} \geq C_4 C_5 \sqrt{\frac{1}{k^{3/4}}},$$

completing the proof.

$\square$

**Lemma 3** (Full). *There exist constants $C, C_2 > 0$ and a constant $k_0$ such that, for all $k \geq k_0$, for any $t \leq T$ such that $c_t \geq C_2/\log^4 k$, we have the following with probability at least $1 - e^{-\widetilde{\Omega}(k)}$ over a mini-batch $\mathcal{Z} \sim \mathcal{D}^n$ of $n = \mathrm{poly}(k)$ samples:*

$$\frac{1}{n}\sum_{(X,y) \in \mathcal{Z}} \nabla_b \ell(f_t(X), y) \geq C.$$

*Proof.* Consider an arbitrary $t \leq T$ such that $c_t \geq C_2/\log^4 k$.

$$\nabla_b \mathbb{E}[\ell(f_t(X), y)] = \sum_{i \in I} \underbrace{\mathbb{E}\left[\sigma'(\overline{-Yf_{-i}(X)}) \left(\text{ReLU}(Z_i + b_t) + \text{ReLU}(-Z_i + b_t)\right)\right]}_{(1)}$$
$$- \underbrace{\mathbb{E}\left[Y\sigma(-Yf_t(X))\left(\mathbb{I}\{c_t Y \geq -b_t\} - \mathbb{I}\{c_t Y \leq b_t\}\right)\right]}_{(2)},$$

where (1) comes from a first-order Taylor expansion of the sigmoid function $\sigma$ around $-Yf_{-i}(X)$, $\sigma'$ is the first derivative of $\sigma$, and $\overline{-Yf_{-i}(X)}$ is some value between $-Yf_{-i}(X)$ and $-Yf_t(X)$ so that $\sigma(-Yf_t(X)) = \sigma(-Yf_{-i}(X)) + \sigma'(\overline{-Yf_{-i}(X)})(-Yf_t(X) + Yf_{-i}(X))$ holds with equality.

By Corollary 1, $\text{Var}[S_i] \geq \beta(c_t + b_t)^2/k$.

By Lemma 12, when $b_t < 0$, we have

$$\mathbb{E}[|S_i|] \geq C_1 \sqrt{\frac{\text{Var}[S_i]}{k^{3/4}}} \geq C_1 \sqrt{\beta} \frac{c_t + b_t}{k^{7/8}}.$$

When $b_t \geq 0$ instead of $< 0$, this only increases $\mathbb{E}[|S_i|]$. Combined with Lemma 8, we conclude that, for any $b_t$, we have:

$$\sum_{i \in I} \mathbb{E}[|S_i|] \geq C_1' c_t k^{1/8},$$

for some constant $C_1' > 0$.

Let $E_1$ be the event that $f_{-i}(X) \leq C + 10$. Let $E_2$ be the event that $f_i(X) \leq 1$. Let $E = E_1 \cap E_2$. Under $E$ and Lemma 10, $\sigma'(\overline{-Yf_{-i}(X)})$ is at least some constant $D > 0$.

$$(1) = \mathbb{E}\left[\sigma'(\overline{-Yf_{-i}(X)})\left(\text{ReLU}(Z_i + b) + \text{ReLU}(-Z_i + b)\right) \mid E_1, E_2\right] \mathbb{P}[E_1]\mathbb{P}[E_2]$$
$$+ \mathbb{E}\left[\sigma'(\overline{-Yf_{-i}(X)})\left(\text{ReLU}(Z_i + b) + \text{ReLU}(-Z_i + b)\right) \mid \neg E\right] \mathbb{P}[\neg E]$$
$$\geq D' \cdot \mathbb{E}[\text{ReLU}(Z_i + b) + \text{ReLU}(-Z_i + b) \mid E_2],$$

for some constant $D' = D \cdot \mathbb{P}[E]$.

Finally, we related the conditioned expectation to the unconditioned expectation as follows:

$$\mathbb{E}[\text{ReLU}(Z_i + b) + \text{ReLU}(-Z_i + b)]$$
$$= \mathbb{E}[\text{ReLU}(Z_i + b) + \text{ReLU}(-Z_i + b) \mid E_2]\mathbb{P}[E_2]$$
$$+ \mathbb{E}[\text{ReLU}(Z_i + b) + \text{ReLU}(-Z_i + b) \mid \neg E_2]\mathbb{P}[\neg E_2]$$
$$= \mathbb{E}[\text{ReLU}(Z_i + b) + \text{ReLU}(-Z_i + b) \mid E_2](1 - e^{-\widetilde{\Omega}(k)}) + e^{-\widetilde{\Omega}(k)},$$

so we have

$$\mathbb{E}[\text{ReLU}(Z_i + b) + \text{ReLU}(-Z_i + b) \mid E_2] = \frac{\mathbb{E}[\text{ReLU}(Z_i + b) + \text{ReLU}(-Z_i + b)] - e^{-\widetilde{\Omega}(k)}}{1 - e^{-\widetilde{\Omega}(k)}}.$$

Thus, $(1) \geq A \cdot \mathbb{E}[\text{ReLU}(Z_i + b) + \text{ReLU}(-Z_i + b)]$, for some constant $A > 0$.

We have a trivial upper bound on (2) of 1 (because $\sigma$ and the indicators are bounded).

Therefore, for $c_t \geq C_2/\log^4 k$, as specified, we have $\nabla_b \mathbb{E}[\ell(f_t(X), y)] = k \cdot (1) - (2) \geq 2C$ for sufficiently large $k$.

Finally, by Lemma 7, we have the following with probability at least $1 - e^{-\widetilde{\Omega}(k)}$, over a mini-batch $\mathcal{Z} \sim \mathcal{D}^n$ of $n = \text{poly}(k)$ samples:

$$\frac{1}{n} \sum_{(X,y) \in \mathcal{Z}} \nabla_b \ell(f_t(X), y) \geq C.$$

$\square$

### A.3.6 Conclusion: efficiency in time and sample complexity

**Theorem 1.** *There exists an absolute constant $k_0$ such that, for every $k \geq k_0$, using $poly(k)$ samples from $\mathcal{D}$, learning rate $\eta = 1/poly(k)$, and $T = poly(k)$ iterations, w.h.p. over the randomness of the initialization and the samples, we have $\Pr_{(X,y) \sim \mathcal{D}}[sign(f_T(X)) \neq y] \leq 0.01$, for the final network $f_T$ returned by Algorithm 1.*

*Proof.* Throughout, consider the largest $k_0$ such that all intermediate lemmas hold for all $k \geq k_0$. Then, for all $k \geq k_0$, we have the following.

By Lemma 3, there exist constants $C, C_2 > 0$ such that, for all $k \geq k_0$, for any $t \leq T$ such that $c_t \geq C_2/\log^4 k$, the following holds with probability at least $1 - e^{-\widetilde{\Omega}(k)}$ over a mini-batch $\mathcal{Z} \sim \mathcal{D}^n$ of $n = poly(k)$ samples:

$$\frac{1}{n} \sum_{(X,y) \in \mathcal{Z}} \nabla_b \ell(f_t(X), y) \geq C.$$

By Lemma 1, plugging in $H = 2C_2$, there exists a constant $A > 0$ such that, for all $k \geq k_0$, for any $t \leq T$ such that $|c_t| \leq 2C_2/\log^4 k$, we have the following with probability at least $1 - e^{-\widetilde{\Omega}(k)}$ over a mini-batch $\mathcal{Z} \sim \mathcal{D}^n$ of $n = poly(k)$ samples:

$$\frac{1}{n} \sum_{(X,y) \in \mathcal{Z}} \nabla_{\mathbf{w}} \ell(f_t(X), y) \cdot \mathbf{w}_\star \leq -A.$$

Therefore, within $T_1 = (\frac{2C_2}{\log^4 k} + \delta)\frac{1}{\eta A} = poly(k)$ iterations, $c_t$ will rise to $2C_2/\log^4 k$. Since we only have $poly(k)$ iterations, the overall failure probability for this first phase is still $e^{-\widetilde{\Omega}(k)}$. Although $c_t$ is not guaranteed to continue *increasing*, Lemma 1 guarantees that $c_t$ will not drop below $C_2/\log^4 k$ after it reaches $2C_2/\log^4 k$.

Once $c_t$ has reached $C_2/\log^4 k$, Lemma 3 says that, with probability at least $1 - e^{-\widetilde{\Omega}(k)}$ per iteration, $\frac{1}{n} \sum_{(X,y) \in \mathcal{Z}} \nabla_b \ell(f_t(X), y) \geq C$. Thus, $b_t$ will decrease at a rate of $\eta_b C$ per iteration.

By Lemma 2, while $t \leq poly(k)$, we have $v_t \leq 2\delta$. Since $c_t \geq C_2/\log^4 k$ and $v_t \leq 2\delta$, we have (loosely) $s_t \leq 2c_t\sigma$, and so $\frac{|b_t|}{s_t} \geq \frac{|b_t|}{2c_t\sigma}$.

By Lemma 8, once we have $\alpha_t = |b_t|/s_t > 2\sqrt{\log k}$, we will have $\mathbb{P}_{(X,y) \sim \mathcal{D}}[sign(f_t(X)) \neq y] < 0.01$. By Lemma 10, there exists a constant $C_3 > 0$ such that $c_t \leq C_3$. Thus, once $\frac{|b_t|}{2C_3\sigma} > 2\sqrt{\log k}$, or equivalently $|b_t| > 4C_3 \frac{\log^{3/2} k}{\sqrt{k}}$, we will have $\mathbb{P}_{(X,y) \sim \mathcal{D}}[sign(f_t(X)) \neq y] < 0.01$.

Since $b$ is decreasing at a rate of at least $\eta_b C$ per iteration, this occurs within $T_2 = (4C_3 \frac{\log^{3/2} k}{\sqrt{k}} + \frac{C_2}{\log^4 k} \frac{\eta_b}{\eta})\frac{1}{\eta_b C}$ iterations, where $\frac{C_2}{\log^4 k} \frac{\eta_b}{\eta}$ accounts for $b$'s initial growth at the beginning of training.

Since $T_1 + T_2 = poly(k)$, we reach final classification error $\leq 0.01$ within $poly(k)$ iterations (assuming the $poly(k)$ in Lemma 2 is at least $T_1 + T_2$).

Since a mini-batch of size $n = poly(k)$ is used per iteration, the final sample complexity over $poly(k)$ total iterations is also $poly(k)$.

Since the failure probability per iteration is $e^{-\widetilde{\Omega}(k)}$, the total failure probability over initialization and $poly(k)$ iterations is also $e^{-\widetilde{\Omega}(k)}$. This completes the proof.

$\square$

# B  Full proofs for Section 5

**Lemma 13** (Small ball probability, [Tao, 2010]). *Consider $n$ independent Rademacher random variables $\xi_i$ for $i \in [n]$ and constants $a_i$ for $i \in [n]$ such that $|a_i| \geq 1$. Then, for any length-$2\Delta$ interval $B$, for $\Delta > 0$, we have $\mathbb{P}\left[\sum_{i \in [n]} \xi_i a_i \in B\right] \leq \frac{s\sqrt{\frac{2}{\pi}} + o(1)}{\sqrt{n}}$ whenever $s \leq n$ and $s - 1 \leq \Delta < s$ for some natural number $s$.*

## B.1  Warm up: one filter

We first present a proof when there is only one filter, to help elucidate the proof skeleton. Then, in Section B.2, we provide the full proof of Theorem 2. Readers can ignore this subsection and skip directly to Section B.2 if they like; this subsection is merely provided to help make some of the themes in Section B.2 a bit clearer.

**Theorem 3** ($m = 1$). *There exists $k_1 \in \mathbb{N}$ such that, for all $k \geq k_1$, with probability at least $0.999$ over the random initialization $\mathbf{w}^{(0)} \sim \mathcal{N}(0, \sigma_0^2 I_d)$, the following holds for all $\mathbf{w} \in \mathbb{R}^d, b \in \mathbb{R}$:*

$$\mathbb{P}_{(X,y) \sim \mathcal{D}}[\operatorname{sign}(k_{\mathbf{w}}(X)) \neq y] \geq 0.1.$$

*Proof.* We recall that the finite-width CNTK is defined as:

$$k_{\mathbf{w}}(X) = \sum_{i \in [k+1]} \langle \mathbf{w}, X_i \rangle 1_{|\langle \mathbf{w}^{(0)}, X_i \rangle| + b \geq 0}.$$

This comes from the gradient of the CNN function $f_{\mathbf{w},b}(X)$ with respect to $\mathbf{w}$:

$$\nabla_{\mathbf{w}} f_{\mathbf{w},b}(X) = \sum_{i=1}^{k+1} X_i \left[ \mathbb{I}\{\langle \mathbf{w}, X_i \rangle + b > 0\} + \mathbb{I}\{-\langle \mathbf{w}, X_i \rangle + b > 0\} \right].$$

For convenience, for a single row of the input image $X$, denoted $x \in \mathbb{R}^d$, we define

$$g(x) := \langle \mathbf{w}, x \rangle 1_{|\langle \mathbf{w}^{(0)}, x \rangle| + b \geq 0},$$

which represents the total contribution to $k_{\mathbf{w}}(X)$ coming from $x$. We thus have

$$k_{\mathbf{w}}(X) = \sum_{i \in [k+1]} g(X_i) = g(y\mathbf{w}_\star) + \sum_{i \in I} g(\epsilon_i).$$

If $y(g(y\mathbf{w}_\star)) \leq 0$, then by symmetry of $g(\epsilon_i)$, we have $\mathbb{P}_{X,y \sim \mathcal{D}}\left[y \sum_{i \in I} g(\epsilon_i) \leq 0\right] = 0.5$, so

$$\mathbb{P}_{(X,y) \sim \mathcal{D}}[\operatorname{sign}(k_{\mathbf{w}}(X)) \neq y] \geq 0.5 > 0.1.$$

It thus remains to consider the more complex case $y(g(y\mathbf{w}_\star)) > 0$.

There exists some constant $D_{0.999} > 0$ such that

$$\mathbb{P}_{\mathbf{w}^{(0)} \sim \mathcal{N}(0, \sigma_0^2 I)}\left[\underbrace{\left|\left\langle \frac{\mathbf{w}^{(0)}}{\|\mathbf{w}^{(0)}\|}, \mathbf{w}_\star \right\rangle\right| \leq \frac{D_{0.999}}{\sqrt{d}}}_{\text{Init Event}}\right] \geq 0.999.$$

We assume $\mathbf{w}^{(0)}$ satisfies the Init Event. Then, by Lemma 14, for any $\mathbf{w}, b$, there exist constants $C > 0, p_C \in (0, 1]$ such that

$$p := \mathbb{P}_{\epsilon_i \sim \mathcal{N}(0, \sigma^2 I_d)}\left[\underbrace{|g(\epsilon_i)| \geq C\sigma|g(\mathbf{w}_\star)|}_{\text{Event } A}\right] \geq p_C.$$

Since each noise row has the same distribution, the same $C, p, p_C$ hold for all $i \in I$.

Let $I^+ := \{i \in I : |g(\epsilon_i)| \geq C\sigma|g(\mathbf{w}_\star)|\}$ and let $I^- := I \setminus I^+$.

By Chernoff, $\mathbb{P}[|I^+| \leq 0.5kp] \leq e^{-0.5^2 kp/2}$.

Applying Lemma 13, we obtain:

$$\mathbb{P}\left[\sum_{i \in I^+} g(\epsilon_i) \in \left[-|g(\mathbf{w}_\star)|, |g(\mathbf{w}_\star)|\right]\right] = \mathbb{P}\left[\sum_{i \in I^+} \frac{g(\epsilon_i)}{C\sigma|g(\mathbf{w}_\star)|} \in \left[-\frac{|g(\mathbf{w}_\star)|}{C\sigma|g(\mathbf{w}_\star)|}, \frac{|g(\mathbf{w}_\star)|}{C\sigma|g(\mathbf{w}_\star)|}\right]\right]$$

$$= \mathbb{P}\left[\sum_{i \in I^+} \frac{g(\epsilon_i)}{C\sigma|g(\mathbf{w}_\star)|} \in \left[-\frac{1}{C\sigma}, \frac{1}{C\sigma}\right]\right]$$

$$= \mathcal{O}\left(\frac{1}{\sigma\sqrt{|I^+|}}\right).$$

Thus, the **probability of misclassification** $\mathbb{P}_{(X,y)\sim\mathcal{D}}[\text{sign}(k_\mathbf{w}(X)) \neq y]$ is at least:
(We have $\text{sign}(0) := 0$, so $k_\mathbf{w}(X) = 0$ is necessarily a misclassification event.)

$$\mathbb{P}_{(X,y)\sim\mathcal{D}}[\text{sign}(k_\mathbf{w}(X)) \neq y] = \mathbb{P}_{(X,y)\sim\mathcal{D}}[yk_\mathbf{w}(X) \leq 0]$$

$$= \mathbb{P}_{(X,y)\sim\mathcal{D}}\left[y\sum_{i \in I} g(\epsilon_i) \leq -yg(\mathbf{w}_\star)\right]$$

$$\geq \mathbb{P}_{(X,y)\sim\mathcal{D}}\left[\sum_{i \in I} g(\epsilon_i) \geq |g(\mathbf{w}_\star)|\right]$$

$$\geq \mathbb{P}_{(X,y)\sim\mathcal{D}}\left[\sum_{i \in I^+} g(\epsilon_i) \geq |g(\mathbf{w}_\star)|, \sum_{i \in I^-} g(\epsilon_i) \geq 0\right]$$

$$= \mathbb{P}_{(X,y)\sim\mathcal{D}}\left[\sum_{i \in I^+} g(\epsilon_i) \geq |g(\mathbf{w}_\star)|\right] \cdot \mathbb{P}_{(X,y)\sim\mathcal{D}}\left[\sum_{i \in I^-} g(\epsilon_i) \geq 0\right]$$

$$= \frac{1}{2}\mathbb{P}_{(X,y)\sim\mathcal{D}}\left[\sum_{i \in I^+} g(\epsilon_i) \geq |g(\mathbf{w}_\star)|\right]$$

$$= \frac{1}{4}\mathbb{P}_{(X,y)\sim\mathcal{D}}\left[\sum_{i \in I^+} g(\epsilon_i) \notin (-|g(\mathbf{w}_\star)|, |g(\mathbf{w}_\star)|)\right]$$

$$= \frac{1}{4}\left(1 - \mathbb{P}_{(X,y)\sim\mathcal{D}}\left[\sum_{i \in I^+} g(\epsilon_i) \in (-|g(\mathbf{w}_\star)|, |g(\mathbf{w}_\star)|)\right]\right)$$

$$\mathbb{P}_{(X,y)\sim\mathcal{D}}\left[\sum_{i \in I^+} g(\epsilon_i) \in (-|g(\mathbf{w}_\star)|, |g(\mathbf{w}_\star)|)\right]$$

$$= \underbrace{\mathbb{P}_{(X,y)\sim\mathcal{D}}\left[\sum_{i \in I^+} g(\epsilon_i) \in (-|g(\mathbf{w}_\star)|, |g(\mathbf{w}_\star)|) \mid |I^+| \leq 0.5kp\right]}_{\leq 1} \mathbb{P}[|I^+| \leq 0.5kp]$$

$$+ \mathbb{P}_{(X,y)\sim\mathcal{D}}\left[\sum_{i \in I^+} g(\epsilon_i) \in (-|g(\mathbf{w}_\star)|, |g(\mathbf{w}_\star)|) \mid |I^+| > 0.5kp\right] \underbrace{\mathbb{P}[|I^+| > 0.5kp]}_{\leq 1}$$

$$\leq e^{-0.5^2 kp/2} + \mathcal{O}\left(\frac{1}{\sigma\sqrt{kp}}\right)$$

$$\leq \mathcal{O}\left(\frac{1}{(\log k)\sqrt{p_C}}\right)$$

$$= \mathcal{O}\left(\frac{1}{\log k}\right).$$

Putting this together, we get

$$\mathbb{P}_{(X,y)\sim\mathcal{D}}[\text{sign}(k_{\mathbf{w}}(X)) \neq y] \geq \frac{1}{4}\left(1 - \mathcal{O}(1/\log k)\right).$$

Thus, there exists $k_1 \in \mathbb{N}$ such that, for all $k \geq k_1$, with probability at least $0.999$ over the random initialization $\mathbf{w}^{(0)} \sim \mathcal{N}(0, \sigma_0^2 I_d)$, the following holds for all $\mathbf{w} \in \mathbb{R}^d, b \in \mathbb{R}$:

$$\mathbb{P}_{(X,y)\sim\mathcal{D}}[\text{sign}(k_{\mathbf{w}}(X)) \neq y] \geq 0.1.$$

$\square$

**Lemma 14** ($m = 1$). *Assume $\mathbf{w}^{(0)}$ satisfies the Init Event. Then, for any $\mathbf{w}, b$, there exist constants $C > 0, p_C \in (0, 1]$ such that*

$$\mathbb{P}_{\epsilon\sim\mathcal{N}(0,\sigma^2 I_d)}\Big[\underbrace{|g(\epsilon)| \geq C\sigma|g(\mathbf{w}_\star)|}_{Event\ A}\Big] \geq p_C.$$

*Proof.* As a reminder, in this lemma, we are looking exclusively at the $m = 1$ case. When $m = 1$, we have:

$$g(\mathbf{w}_\star) = \langle \mathbf{w}, \mathbf{w}_\star\rangle 1_{|\langle\mathbf{w}^{(0)}, \mathbf{w}_\star\rangle|+b\geq 0}.$$

We now consider different possible settings of $\mathbf{w}^{(0)}, \mathbf{w}, b$, via a case analysis.

*Case 1:* Suppose $|\langle\mathbf{w}^{(0)}, \mathbf{w}_\star\rangle| + b < 0$. Then $g(\mathbf{w}_\star) = 0$. $|g(\epsilon)| \geq 0$, so for all $C > 0$, $p_C = 1$ satisfies the lemma statement.

*Case 2:* Suppose $|\langle\mathbf{w}^{(0)}, \mathbf{w}_\star\rangle| + b \geq 0$. Then $g(\mathbf{w}_\star) = \langle\mathbf{w}, \mathbf{w}_\star\rangle$, so Event $A$ becomes:

$$|\langle\mathbf{w}, \epsilon\rangle 1_{|\langle\mathbf{w}^{(0)}, \epsilon\rangle|+b\geq 0}| \geq C\sigma|\langle\mathbf{w}, \mathbf{w}_\star\rangle|.$$

Let $E$ be the event that $|\langle\mathbf{w}^{(0)}, \epsilon\rangle| \geq |\langle\mathbf{w}^{(0)}, \mathbf{w}_\star\rangle|$. We are introducing $E$ so that we can condition $A$ on $E$ and thus simplify our analysis of $\mathbb{P}[A]$. We can simplify $\mathbb{P}[E]$ as follows, for use later:

$$\mathbb{P}_{\epsilon\sim\mathcal{N}(0,\sigma^2 I_d)}[E] = \mathbb{P}_{x\sim\mathcal{N}(0,\sigma^2)}\left[|x| \geq \left|\left\langle\frac{\mathbf{w}^{(0)}}{\|\mathbf{w}^{(0)}\|}, \mathbf{w}_\star\right\rangle\right|\right].$$

We therefore analyze $\mathbb{P}[A]$ as follows:

$$\begin{aligned}
\mathbb{P}_{\epsilon\sim\mathcal{N}(0,\sigma^2 I_d)}[A] &= \mathbb{P}_{\epsilon\sim\mathcal{N}(0,\sigma^2 I_d)}[A \mid E]\mathbb{P}_{\epsilon\sim\mathcal{N}(0,\sigma^2 I_d)}[E] + \mathbb{P}_{\epsilon\sim\mathcal{N}(0,\sigma^2 I_d)}[A \mid \neg E]\mathbb{P}_{\epsilon\sim\mathcal{N}(0,\sigma^2 I_d)}[\neg E] \\
&\geq \mathbb{P}_{\epsilon\sim\mathcal{N}(0,\sigma^2 I_d)}[A \mid E]\mathbb{P}_{\epsilon\sim\mathcal{N}(0,\sigma^2 I_d)}[E] \\
&= \mathbb{P}_{\epsilon\sim\mathcal{N}(0,\sigma^2 I_d)}[|\langle\mathbf{w}, \epsilon\rangle| \geq C\sigma|\langle\mathbf{w}, \mathbf{w}_\star\rangle| \mid E]\mathbb{P}_{\epsilon\sim\mathcal{N}(0,\sigma^2 I_d)}[E] \\
&\geq \mathbb{P}_{\epsilon\sim\mathcal{N}(0,\sigma^2 I_d)}[|\langle\mathbf{w}, \epsilon\rangle| \geq C\sigma|\langle\mathbf{w}, \mathbf{w}_\star\rangle|]\mathbb{P}_{\epsilon\sim\mathcal{N}(0,\sigma^2 I_d)}[E] \\
&= \mathbb{P}_{\epsilon\sim\mathcal{N}(0,\sigma^2 I_d)}\left[\frac{1}{\sigma}\left|\left\langle\frac{\mathbf{w}}{\|\mathbf{w}\|}, \epsilon\right\rangle\right| \geq C\left|\left\langle\frac{\mathbf{w}}{\|\mathbf{w}\|}, \mathbf{w}_\star\right\rangle\right|\right]\mathbb{P}_{\epsilon\sim\mathcal{N}(0,\sigma^2 I_d)}[E] \\
&= \mathbb{P}_{x\sim\mathcal{N}(0,1)}\left[|x| \geq C\left|\left\langle\frac{\mathbf{w}}{\|\mathbf{w}\|}, \mathbf{w}_\star\right\rangle\right|\right]\mathbb{P}_{\epsilon\sim\mathcal{N}(0,\sigma^2 I_d)}[E] \\
&= \mathbb{P}_{x\sim\mathcal{N}(0,1)}\left[|x| \geq C\left|\left\langle\frac{\mathbf{w}}{\|\mathbf{w}\|}, \mathbf{w}_\star\right\rangle\right|\right]\mathbb{P}_{x\sim\mathcal{N}(0,\sigma^2)}\left[|x| \geq \left|\left\langle\frac{\mathbf{w}^{(0)}}{\|\mathbf{w}^{(0)}\|}, \mathbf{w}_\star\right\rangle\right|\right] \\
&\geq \underbrace{\mathbb{P}_{x\sim\mathcal{N}(0,1)}\left[|x| \geq C\left|\left\langle\frac{\mathbf{w}}{\|\mathbf{w}\|}, \mathbf{w}_\star\right\rangle\right|\right]}_{(1)}\underbrace{\mathbb{P}_{x\sim\mathcal{N}(0,\sigma^2)}\left[|x| \geq \frac{D_{0.999}}{\sqrt{d}}\right]}_{(2)} \\
&= p_C
\end{aligned}$$

for some constant $p_C > 0$, since probability $(1)$ only depends on $C$ and the angle between $\mathbf{w}$ and $\mathbf{w}_\star$, and probability $(2)$ is just $\mathbb{P}_{x\sim\mathcal{N}(0,1)}[|x| \geq D_{0.999}]$ due to the assumption that $\sigma^2 = 1/d$. $\square$

**Remark 2.** *We note that, as long as $\sigma^2 = \Omega(1/d)$, probability (2) above will be $\Omega(1)$, which satisfies the lemma. However, Lemma 15 (the analogous theorem in the case where we have multiple filters) depends more subtly on having $\sigma^2 = \Theta(1/d)$, not just $\sigma^2 = \Omega(1/d)$.*

## B.2  Multiple filters

Throughout, we use the shorthand $\mathbf{w}^{(0)}$ to denote $\{\mathbf{w}_j^{(0)}\}_{j \in [m]}$, $\mathbf{w}$ to denote $\{\mathbf{w}_j\}_{j \in [m]}$, and $b$ to denote $\{b_j\}_{j \in [m]}$.

**Theorem 2.** *For any $C > 0$, if there exists $k_0$ such that $m \leq C$ for all $k \geq k_0$, then there exists $k_1 \geq k_0$ such that, for all $k \geq k_1$, with probability at least $0.999$ over the random initialization $\{\mathbf{w}_j^{(0)}\}_{j \in [m]}$ where each $\mathbf{w}_j^{(0)}$ i.i.d. $\sim \mathcal{N}(0, \sigma_0^2 I)$, the following holds for every set of weights $\mathbf{w} := \{\mathbf{w}_j\}_{j \in [m]}$ and every set of biases $b := \{b_j\}_{j \in [m]}$,*

$$\Pr_{X, y \sim \mathcal{D}}[sign(k_{\mathbf{w}}(X)) \neq y] \geq 0.1,$$

*where $\sigma^2 = 1/d$.*

*Proof.* We recall that the finite-width CNTK is defined as:

$$k_{\mathbf{w}}(X) = \sum_{i \in [k+1]} \sum_{j \in [m]} \langle \mathbf{w}_j, X_i \rangle 1_{|\langle \mathbf{w}_j^{(0)}, X_i \rangle| + b_j \geq 0}.$$

This comes from the gradient of the CNN function $f_{\mathbf{w}, b}(X)$ with respect to $\mathbf{w}$:

$$\nabla_{\mathbf{w}} f_{\mathbf{w}, b}(X) = \sum_{i=1}^{k+1} X_i \left[ \mathbb{I}\{\langle \mathbf{w}, X_i \rangle + b > 0\} + \mathbb{I}\{-\langle \mathbf{w}, X_i \rangle + b > 0\} \right].$$

For $j \in [m]$, let $\hat{v}_j := \dfrac{\mathbf{w}_j^{(0)}}{\|\mathbf{w}_j^{(0)}\|}$, and define the set $\mathcal{J} := \{(j, j') \in [m] \times [m] : j \neq j'\}$.

Then there exist some positive constants $D_{0.999}^{(l)}, D_{0.999}^{(u)}, D_{0.999}^{(p)} > 0$ such that

$$\mathbb{P}_{\mathbf{w}^{(0)}} \left[ \underbrace{\forall j \in [m], \frac{D_{0.999}^{(l)}}{\sqrt{d}} \leq |\langle \hat{v}_j, \mathbf{w}_\star \rangle| \leq \frac{D_{0.999}^{(u)}}{\sqrt{d}} \bigcap \forall (j, j') \in \mathcal{J}, |\langle \hat{v}_j, \hat{v}_{j'} \rangle| \leq \frac{D_{0.999}^{(p)}}{\sqrt{d}}}_{\text{Init Event}} \right] \geq 0.999.$$

(We provide the following two references for completeness: Lalley Lemma 2.7, and Gorban and Tyukin [2018], Proposition 2.1.)

For convenience, for a single row of the input image $X$, denoted $x \in \mathbb{R}^d$, we define

$$g(x) := \sum_{j \in [m]} \langle \mathbf{w}_j, x \rangle 1_{|\langle \mathbf{w}_j^{(0)}, x \rangle| + b_j \geq 0},$$

which represents the total contribution to $k_{\mathbf{w}}(X)$ coming from $x$. We thus have

$$k_{\mathbf{w}}(X) = \sum_{i \in [k+1]} g(X_i) = g(y \mathbf{w}_\star) + \sum_{i \in I} g(\epsilon_i).$$

Throughout the remainder of the proof, we consider arbitrary initialization $\mathbf{w}^{(0)}$ satisfying the above criteria and arbitrary $\mathbf{w}, b$.

If $y(g(y \mathbf{w}_\star)) \leq 0$, then by symmetry of $g(\epsilon_i)$, we have $\mathbb{P}_{X, y \sim \mathcal{D}} \left[ y \sum_{i \in I} g(\epsilon_i) \leq 0 \right] = 0.5$, so

$$\mathbb{P}_{(X, y) \sim \mathcal{D}}[\text{sign}(k_{\mathbf{w}}(X)) \neq y] \geq 0.5 > 0.1.$$

It thus remains to consider the more complex case $y(g(y \mathbf{w}_\star)) > 0$.

By Lemma 15, for any $\mathbf{w}, b$, there exist constants $C > 0, p_C \in (0,1]$ such that

$$p := \mathbb{P}_{\epsilon_i \sim \mathcal{N}(0,\sigma^2 I_d)}\Big[\underbrace{|g(\epsilon_i)| \geq C\sigma|g(\mathbf{w}_\star)|}_{\text{Event } A}\Big] \geq p_C.$$

Since each noise row has the same distribution, the same $C, p, p_C$ hold for all $i \in I$.

Let $I^+ := \{i \in I : |g(\epsilon_i)| \geq C\sigma|g(\mathbf{w}_\star)|\}$ and let $I^- := I \setminus I^+$.

By Chernoff, $\mathbb{P}[|I^+| \leq 0.5kp] \leq e^{-0.5^2 kp/2}$.

Applying Lemma 13, we obtain:

$$\mathbb{P}\left[\sum_{i \in I^+} g(\epsilon_i) \in \left[-|g(\mathbf{w}_\star)|, |g(\mathbf{w}_\star)|\right]\right] = \mathbb{P}\left[\sum_{i \in I^+} \frac{g(\epsilon_i)}{C\sigma|g(\mathbf{w}_\star)|} \in \left[-\frac{|g(\mathbf{w}_\star)|}{C\sigma|g(\mathbf{w}_\star)|}, \frac{|g(\mathbf{w}_\star)|}{C\sigma|g(\mathbf{w}_\star)|}\right]\right]$$

$$= \mathbb{P}\left[\sum_{i \in I^+} \frac{g(\epsilon_i)}{C\sigma|g(\mathbf{w}_\star)|} \in \left[-\frac{1}{C\sigma}, \frac{1}{C\sigma}\right]\right]$$

$$= \mathcal{O}\left(\frac{1}{\sigma\sqrt{|I^+|}}\right).$$

Thus, the **probability of misclassification** $\mathbb{P}_{(X,y)\sim\mathcal{D}}[\text{sign}(k_\mathbf{w}(X)) \neq y]$ is at least:
(We have $\text{sign}(0) := 0$, so $k_\mathbf{w}(X) = 0$ is necessarily a misclassification event.)

$$\mathbb{P}_{(X,y)\sim\mathcal{D}}[\text{sign}(k_\mathbf{w}(X)) \neq y] = \mathbb{P}_{(X,y)\sim\mathcal{D}}[yk_\mathbf{w}(X) \leq 0]$$

$$= \mathbb{P}_{(X,y)\sim\mathcal{D}}\left[y\sum_{i \in I} g(\epsilon_i) \leq -yg(\mathbf{w}_\star)\right]$$

$$\geq \mathbb{P}_{(X,y)\sim\mathcal{D}}\left[\sum_{i \in I} g(\epsilon_i) \geq |g(\mathbf{w}_\star)|\right]$$

$$\geq \mathbb{P}_{(X,y)\sim\mathcal{D}}\left[\sum_{i \in I^+} g(\epsilon_i) \geq |g(\mathbf{w}_\star)|, \sum_{i \in I^-} g(\epsilon_i) \geq 0\right]$$

$$= \mathbb{P}_{(X,y)\sim\mathcal{D}}\left[\sum_{i \in I^+} g(\epsilon_i) \geq |g(\mathbf{w}_\star)|\right] \cdot \mathbb{P}_{(X,y)\sim\mathcal{D}}\left[\sum_{i \in I^-} g(\epsilon_i) \geq 0\right]$$

$$= \frac{1}{2}\mathbb{P}_{(X,y)\sim\mathcal{D}}\left[\sum_{i \in I^+} g(\epsilon_i) \geq |g(\mathbf{w}_\star)|\right]$$

$$= \frac{1}{4}\mathbb{P}_{(X,y)\sim\mathcal{D}}\left[\sum_{i \in I^+} g(\epsilon_i) \notin (-|g(\mathbf{w}_\star)|, |g(\mathbf{w}_\star)|)\right]$$

$$= \frac{1}{4}\left(1 - \mathbb{P}_{(X,y)\sim\mathcal{D}}\left[\sum_{i \in I^+} g(\epsilon_i) \in (-|g(\mathbf{w}_\star)|, |g(\mathbf{w}_\star)|)\right]\right)$$

$$\mathbb{P}_{(X,y)\sim\mathcal{D}}\left[\sum_{i \in I^+} g(\epsilon_i) \in (-|g(\mathbf{w}_\star)|, |g(\mathbf{w}_\star)|)\right]$$

$$= \underbrace{\mathbb{P}_{(X,y)\sim\mathcal{D}}\left[\sum_{i \in I^+} g(\epsilon_i) \in (-|g(\mathbf{w}_\star)|, |g(\mathbf{w}_\star)|) \mid |I^+| \leq 0.5kp\right] \mathbb{P}[|I^+| \leq 0.5kp]}_{\leq 1}$$

$$+ \mathbb{P}_{(X,y)\sim\mathcal{D}}\left[\sum_{i \in I^+} g(\epsilon_i) \in (-|g(\mathbf{w}_\star)|, |g(\mathbf{w}_\star)|) \mid |I^+| > 0.5kp\right] \underbrace{\mathbb{P}[|I^+| > 0.5kp]}_{\leq 1}$$

$$\leq e^{-0.5^2 kp/2} + \mathcal{O}\left(\frac{1}{\sigma\sqrt{kp}}\right)$$

$$\leq \mathcal{O}\left(\frac{1}{(\log k)\sqrt{p_C}}\right)$$

$$= \mathcal{O}\left(\frac{1}{\log k}\right).$$

Putting this together, we get

$$\mathbb{P}_{(X,y)\sim\mathcal{D}}[\text{sign}(k_{\mathbf{w}}(X)) \neq y] \geq \frac{1}{4}\left(1 - \mathcal{O}(1/\log k)\right).$$

Thus, there exists $k_1 \in \mathbb{N}$ such that, for all $k \geq k_1$, with probability at least $0.999$ over the random initialization $\mathbf{w}_j^{(0)} \sim \mathcal{N}(0, \sigma_0^2 I_d)$, the following holds for all $\mathbf{w}, b$:

$$\mathbb{P}_{(X,y)\sim\mathcal{D}}[\text{sign}(k_{\mathbf{w}}(X)) \neq y] \geq 0.1.$$

$\square$

**Lemma 15** (general case: $1 \leq m \leq C$). *Assume $\mathbf{w}^{(0)}$ satisfies the Init Event. Then, for any $\mathbf{w}, b$, there exist constants $C > 0, p_C \in (0, 1]$ such that*

$$\mathbb{P}_{\epsilon\sim\mathcal{N}(0,\sigma^2 I_d)}\Big[\underbrace{|g(\epsilon)| \geq C\sigma|g(\mathbf{w}_\star)|}_{\text{Event A}}\Big] \geq p_C.$$

*Proof.* We first consider the trivial case where $g(\mathbf{w}_\star) = 0$. We necessarily have $|g(\epsilon)| \geq 0$, so for all $C > 0$, $p_C = 1$ satisfies the lemma statement. So throughout the rest of this proof, we focus on the case where $g(\mathbf{w}_\star) \neq 0$.

For any $\mathbf{w} \in \mathbb{R}^d$, define

$$\mathcal{M}(\mathbf{w}) := \{j \in [m] : |\langle \mathbf{w}_j^{(0)}, \mathbf{w}\rangle| + b_j \geq 0\}.$$

Note that, for any $\mathbf{w}$, $\mathcal{M}(\mathbf{w})$ depends on the random initialization $\mathbf{w}^{(0)}$ and the chosen bias $b$.

We use $\mathcal{M}$ to write $g(\mathbf{w}_\star), g(\epsilon)$ as follows:

$$g(\mathbf{w}_\star) = \sum_{j\in[m]} \langle \mathbf{w}_j, \mathbf{w}_\star\rangle 1_{|\langle \mathbf{w}_j^{(0)}, \mathbf{w}_\star\rangle|+b_j\geq0} = \sum_{j\in\mathcal{M}(\mathbf{w}_\star)} \langle \mathbf{w}_j, \mathbf{w}_\star\rangle = \left\langle \sum_{j\in\mathcal{M}(\mathbf{w}_\star)} \mathbf{w}_j, \mathbf{w}_\star\right\rangle$$

$$g(\epsilon) = \sum_{j\in[m]} \langle \mathbf{w}_j, \epsilon\rangle 1_{|\langle \mathbf{w}_j^{(0)}, \epsilon\rangle|+b_j\geq0} = \sum_{j\in\mathcal{M}(\epsilon)} \langle \mathbf{w}_j, \epsilon\rangle = \left\langle \sum_{j\in\mathcal{M}(\epsilon)} \mathbf{w}_j, \epsilon\right\rangle.$$

We first show that $\mathbb{P}[\mathcal{M}(\epsilon) = \mathcal{M}(\mathbf{w}_\star)] \geq q$, for some constant $q > 0$, the proof of which continues until (9).

For $j \in [m]$, let $\hat{v}_j := \frac{\mathbf{w}_j^{(0)}}{\|\mathbf{w}_j^{(0)}\|}$, and let $\{u_j\}_{j\in[m]}$ be the unnormalized output of Gram-Schmidt applied to $\{\hat{v}_j\}_{j\in[m]}$. Formally, Gram-Schmidt will yield:

$$u_1 = \hat{v}_1$$

$$u_j = \hat{v}_j - \sum_{i=1}^{j-1} \frac{\langle \hat{v}_j, u_i\rangle}{\langle u_i, u_i\rangle} u_i.$$

We will show that the following invariant holds for all $j \in [m]$:

$$u_j = \hat{v}_j \pm \mathcal{O}\left(\frac{1}{\sqrt{d}}\right)\hat{v}_1 \pm \cdots \pm \mathcal{O}\left(\frac{1}{\sqrt{d}}\right)\hat{v}_{j-1}. \tag{7}$$

By the triangle inequality, (7) yields $\|u_j\| \le 1 + \mathcal{O}\left(\frac{1}{\sqrt{d}}\right)$ and $\|u_j\| \ge 1 - \mathcal{O}\left(\frac{1}{\sqrt{d}}\right)$, which implies

$$\|u_j\| = \Theta(1). \tag{8}$$

For $j = 1$, (7) holds because $u_j = \hat{v}_1$ by definition.

For any $j > 1$, assuming (7) holds for all $i < j$ (and thus its corollary (8)), we have

$$u_j = \hat{v}_j - \sum_{i=1}^{j-1} \frac{\left\langle \hat{v}_j, \hat{v}_i \pm \mathcal{O}\left(\frac{1}{\sqrt{d}}\right)\hat{v}_1 \pm \cdots \pm \mathcal{O}\left(\frac{1}{\sqrt{d}}\right)\hat{v}_{i-1}\right\rangle}{\|u_i\|^2}\left[\hat{v}_i \pm \mathcal{O}\left(\frac{1}{\sqrt{d}}\right)\hat{v}_1 \pm \cdots \pm \mathcal{O}\left(\frac{1}{\sqrt{d}}\right)\hat{v}_{i-1}\right]$$

$$= \hat{v}_j \pm \mathcal{O}\left(\frac{1}{\sqrt{d}}\right)\hat{v}_1 \pm \cdots \pm \mathcal{O}\left(\frac{1}{\sqrt{d}}\right)\hat{v}_{j-1}.$$

With invariants (7) and (8) holding for all $j \in [m]$, we can now lower bound $\mathbb{P}[\mathcal{M}(\epsilon) = \mathcal{M}(\mathbf{w}_\star)]$.

We first define $E_1$ as the event that, for all $j \in \mathcal{M}(\mathbf{w}_\star)$, $|\langle \hat{v}_j, \epsilon \rangle| \ge |\langle \hat{v}_j, \mathbf{w}_\star \rangle|$.

We define $E_2$ as the event that, for all $j \in [m] \setminus \mathcal{M}(\mathbf{w}_\star)$, $|\langle \hat{v}_j, \epsilon \rangle| \le |\langle \hat{v}_j, \mathbf{w}_\star \rangle|$.

We then have

$$\mathbb{P}[\mathcal{M}(\epsilon) = \mathcal{M}(\mathbf{w}_\star)] \ge \mathbb{P}[E_1 \cap E_2].$$

We will then lower bound $\mathbb{P}[E_1 \cap E_2]$ by considering the probability that each $|\langle \hat{v}_j, \epsilon \rangle|$ belongs to a particular interval.

We define the two intervals

$$I^{(l)} := \left[\frac{D^{(l)}_{0.999}}{2\sqrt{d}}, \frac{D^{(l)}_{0.999}}{\sqrt{d}}\right], \quad I^{(u)} := \left[\frac{D^{(u)}_{0.999}}{\sqrt{d}}, \frac{2D^{(u)}_{0.999}}{\sqrt{d}}\right].$$

Since the *Init Event* holds, we can see that $|\langle \hat{v}_j, \epsilon \rangle| \in I^{(l)} \implies |\langle \hat{v}_j, \epsilon \rangle| \le |\langle \hat{v}_j, \mathbf{w}_\star \rangle|$.

Analogously, $|\langle \hat{v}_j, \epsilon \rangle| \in I^{(u)} \implies |\langle \hat{v}_j, \epsilon \rangle| \ge |\langle \hat{v}_j, \mathbf{w}_\star \rangle|$.

Thus, for all $j \in \mathcal{M}(\mathbf{w}_\star)$, let $I_j$ represent $I^{(u)}$, and for all $j \in [m] \setminus \mathcal{M}(\mathbf{w}_\star)$, let $I_j$ represent $I^{(l)}$.

We then have

$$\mathbb{P}[E_1 \cap E_2] \ge \mathbb{P}[(\forall j \in [m]) \, |\langle \hat{v}_j, \epsilon \rangle| \in I_j]$$

$$= \prod_{j=1}^{m} \mathbb{P}\left[|\langle \hat{v}_j, \epsilon \rangle| \in I_j \,\Big|\, |\langle \hat{v}_1, \epsilon \rangle| \in I_1, \ldots, |\langle \hat{v}_{j-1}, \epsilon \rangle| \in I_{j-1}\right].$$

We consider $\mathbb{P}\left[|\langle \hat{v}_j, \epsilon \rangle| \in I_j \,\Big|\, |\langle \hat{v}_1, \epsilon \rangle| \in I_1, \ldots, |\langle \hat{v}_{j-1}, \epsilon \rangle| \in I_{j-1}\right]$ for an arbitrary $j \in [m]$.

$$\mathbb{P}\left[|\langle \hat{v}_j, \epsilon \rangle| \in I_j \,\Big|\, |\langle \hat{v}_1, \epsilon \rangle| \in I_1, \ldots, |\langle \hat{v}_{j-1}, \epsilon \rangle| \in I_{j-1}\right]$$

$$= \mathbb{P}\left[\left|\left\langle u_j \pm \mathcal{O}\left(\frac{1}{\sqrt{d}}\right)\hat{v}_1 \pm \cdots \pm \mathcal{O}\left(\frac{1}{\sqrt{d}}\right)\hat{v}_{j-1}, \epsilon\right\rangle\right| \in I_j \,\Big|\, |\langle \hat{v}_1, \epsilon \rangle| \in I_1, \ldots, |\langle \hat{v}_{j-1}, \epsilon \rangle| \in I_{j-1}\right]$$

$$= \mathbb{P}\left[\left|\langle u_j, \epsilon \rangle \pm \mathcal{O}\left(\frac{1}{d}\right)\right| \in I_j \,\Big|\, |\langle \hat{v}_1, \epsilon \rangle| \in I_1, \ldots, |\langle \hat{v}_{j-1}, \epsilon \rangle| \in I_{j-1}\right]$$

$$= \mathbb{P}\left[\left|\langle u_j, \epsilon \rangle \pm \mathcal{O}\left(\frac{1}{d}\right)\right| \in I_j\right]$$

$$= \mathbb{P}_{x \sim \mathcal{N}(0, \|u_j\|^2 \sigma^2)}\left[\left|x \pm \mathcal{O}\left(\frac{1}{d}\right)\right| \in I_j\right]$$

$$
= \begin{cases} \mathbb{P}_{x \sim \mathcal{N}(0, \|u_j\|^2 \sigma^2)} \left[ |x| \in \left[ \frac{D_{0.999}^{(l)}}{2\sqrt{d}} + \mathcal{O}\left(\frac{1}{d}\right), \frac{D_{0.999}^{(l)}}{\sqrt{d}} - \mathcal{O}\left(\frac{1}{d}\right) \right] \right] & \text{if } I_j = I^{(l)} \\[2em] \mathbb{P}_{x \sim \mathcal{N}(0, \|u_j\|^2 \sigma^2)} \left[ |x| \in \left[ \frac{D_{0.999}^{(u)}}{\sqrt{d}} + \mathcal{O}\left(\frac{1}{d}\right), \frac{2D_{0.999}^{(u)}}{\sqrt{d}} - \mathcal{O}\left(\frac{1}{d}\right) \right] \right] & \text{if } I_j = I^{(u)} \end{cases}.
$$

Since $\|u_j\| \sigma = \Theta\left(\frac{1}{\sqrt{d}}\right)$, in either case the probability is at least a constant.

Thus, for some constant $q_j > 0$, we have

$$
\mathbb{P}\left[ |\langle \hat{v}_j, \epsilon \rangle| \in I_j \,\middle|\, |\langle \hat{v}_1, \epsilon \rangle| \in I_1, \ldots, |\langle \hat{v}_{j-1}, \epsilon \rangle| \in I_{j-1} \right] = q_j.
$$

So we obtain

$$
\mathbb{P}[\mathcal{M}(\epsilon) = \mathcal{M}(\mathbf{w}_\star)] \geq \mathbb{P}\left[ E_1 \cap E_2 \right] \geq \mathbb{P}[(\forall j \in [m]) \, |\langle \hat{v}_j, \epsilon \rangle| \in I_j] = \prod_{j=1}^{m} q_j =: q, \qquad (9)
$$

for some constant $q > 0$.

Let

$$
\hat{v} := \frac{\sum_{j \in \mathcal{M}(\mathbf{w}_\star)} \mathbf{w}_j}{\|\sum_{j \in \mathcal{M}(\mathbf{w}_\star)} \mathbf{w}_j\|_2}.
$$

Unless otherwise specified, $\mathbb{P}$ means $\mathbb{P}_{\epsilon \sim \mathcal{N}(0, \sigma^2 I_d)}$:

$$
\mathbb{P}\Big[\underbrace{|g(\epsilon)| \geq C\sigma|g(\mathbf{w}_\star)|}_{\text{Event } A}\Big] \geq \mathbb{P}\left[ \mathcal{M}(\epsilon) = \mathcal{M}(\mathbf{w}_\star), \left| \left\langle \sum_{j \in \mathcal{M}(\mathbf{w}_\star)} \mathbf{w}_j, \epsilon \right\rangle \right| \geq C\sigma|g(\mathbf{w}_\star)| \right]
$$

$$
1 - \mathbb{P}\Big[\underbrace{|g(\epsilon)| \geq C\sigma|g(\mathbf{w}_\star)|}_{\text{Event } A}\Big] \leq 1 - \mathbb{P}\left[ \mathcal{M}(\epsilon) = \mathcal{M}(\mathbf{w}_\star), \left| \left\langle \sum_{j \in \mathcal{M}(\mathbf{w}_\star)} \mathbf{w}_j, \epsilon \right\rangle \right| \geq C\sigma|g(\mathbf{w}_\star)| \right]
$$

$$
= \mathbb{P}\left[ \neg \left( \mathcal{M}(\epsilon) = \mathcal{M}(\mathbf{w}_\star), \left| \left\langle \sum_{j \in \mathcal{M}(\mathbf{w}_\star)} \mathbf{w}_j, \epsilon \right\rangle \right| \geq C\sigma|g(\mathbf{w}_\star)| \right) \right]
$$

$$
= \mathbb{P}\left[ \neg \left( \mathcal{M}(\epsilon) = \mathcal{M}(\mathbf{w}_\star) \right) \cup \neg \left( \left| \left\langle \sum_{j \in \mathcal{M}(\mathbf{w}_\star)} \mathbf{w}_j, \epsilon \right\rangle \right| \geq C\sigma|g(\mathbf{w}_\star)| \right) \right]
$$

$$
\leq 1 - \mathbb{P}[\mathcal{M}(\epsilon) = \mathcal{M}(\mathbf{w}_\star)] + \mathbb{P}\left[ \left| \left\langle \sum_{j \in \mathcal{M}(\mathbf{w}_\star)} \mathbf{w}_j, \epsilon \right\rangle \right| < C\sigma|g(\mathbf{w}_\star)| \right]
$$

$$
\leq 1 - q + \mathbb{P}\left[ \frac{1}{\sigma} |\langle \hat{v}, \epsilon \rangle| < C \, |\langle \hat{v}, \mathbf{w}_\star \rangle| \right]
$$

$$
= 1 - q + \mathbb{P}_{x \sim \mathcal{N}(0,1)} \left[ |x| < C \, |\langle \hat{v}, \mathbf{w}_\star \rangle| \right].
$$

There exists a $C > 0$ such that $\mathbb{P}_{x \sim \mathcal{N}(0,1)} \left[ |x| < C \, |\langle \hat{v}, \mathbf{w}_\star \rangle| \right] = q/2$.

We thus have $1 - \mathbb{P}\Big[\underbrace{|g(\epsilon)| \geq C\sigma|g(\mathbf{w}_\star)|}_{\text{Event } A}\Big] \leq 1 - q/2$, so $\mathbb{P}\Big[\underbrace{|g(\epsilon)| \geq C\sigma|g(\mathbf{w}_\star)|}_{\text{Event } A}\Big] \geq q/2$.

$\square$

# C   Experiment details

Details of all experiments are reported here.[2] All experiments were run on a single Nvidia GeForce RTX 2080 Ti GPU and took between 2 and 10 hours per run. Models were implemented using the `jax` python library[Bradbury et al., 2018], with the `neural-tangents` library [Novak et al., 2020] used for the NTK implementation and some code borrowed from the `autol2` library [Lewkowycz and Gur-Ari, 2020]. All three libraries are available under an Apache License, Version 2.0.

**Synthetic data.**   The experiments with our model on synthetic data (Figure 3) were run on data with the following parameters: $k = 1000$, $d = 10$, and $\sigma = 1$. The model was trained for 10,000 steps of SGD with learning rates $\eta_w = 0.1$ and $\eta_b = 0.1/1000$ and minibatches of size 500, with new i.i.d. samples generated for each batch. Batch size and learning rate were chosen as large (resp. small) as possible given computational constraints, to best simulate the population-gradient setting.

**CIFAR variants.**   All experiments on CIFAR variants, including the sparsity experiments in Figure 2, all experiments in Section 6, and the experiments in Appendix D below, use the following:

- **Dataset details**. The IMAGENET Plants synset was constructed using all WordNet IDs which are subclasses of the Plants ID n00017222. The CIFAR-VEHICLES task is to classify between CIFAR-10 classes airplane, automobile, ship and truck, with animal classes as noise (bird, cat, deer, dog, frog, and horse). IMAGENET and CIFAR-10 are both datasets of public-domain images. Our modifications do not add any personally-identifiable information or offensive content. Example images are included in the code repository.

- **Model architectures**.   We use the same architectures as those in Allen-Zhu and Li [2020]. **WRN**: we use a WideResNet WRN10-10, meaning depth=10 and widening factor $k$=10. This corresponds to a total of 3 residual blocks and 7 total convolutional layers. **NTK**:We use the corresponding finite-width (linearized) NTK of the WRN.

- **Training details**.   We again largely follow the protocol of Allen-Zhu and Li [2020]. However, we do not use data augmentation or Cutout since it is unclear how these may interact with the constructed noise images. **WRN**: Momentum optimizer with mass = 0.9, initial lr = 0.1, batch size = 128, weight decay = 0.0005, 200 epoch, and decay lr by 0.2 at epochs 80, 100, and 120 epochs. For the WRN experiments on larger images reported below, we reduced batch size to 50 due to computational constraints. In order to maintain stable training with this smaller batch size, we also had to reduce the initial learning rate to 0.05. **NTK**: Adam optimizer, initial lr = 0.001, batch size = 50, no weight decay, 200 epoch, and decay lr by 0.2 at epochs 140 and 170 epochs.

**MNIST.**   The MNIST experiments involving CNNs and finite-width NTKs are run using the same settings as those on CIFAR. Instead of a WRN, the network architecture consists of two convolutional layers with 8 and 16 channels, 3x3 kernels, stride 2, and ReLU activations, followed by a fully connected layer. The widened finite-width NTKs are obtained by increasing the number of convolutional channels and then taking the corresponding finite-width NTK. For the infinite-width NTK, we use the `nt.predict.gradient_descent_mse_ensemble` method of the `neural-tangents` library. We explored various diagonal regularization values between 1e-1 and 1e-8 (prior to the trace scaling provided by `neural-tangents` library); however, we found this to have little effect on the performance of the infinite-width NTK; the test accuracy stayed within approximately a 2% range per Gaussian noise level. Therefore, for simplicity, we report the values at regularization strength 1e-5.

# D   Additional experiments

**Image size and placement.**   In Section 6 we saw that as the background noise level increases, NTK performance degrades significantly more quickly than WRN performance. Those experiments are done with 16x16 images on a 32x32 noise background, and all but the CIFAR-VEHICLES experiment place the image at a random location on the background. Here we conduct additional experiments to show that the performance gap is not due to varying image location or small image size. Specifically, we replicate the experiment on CIFAR-2 with Gaussian noise, but with two

---

[2]Code for experiments is available at `https://github.com/skarp/local-signal-adaptivity`.

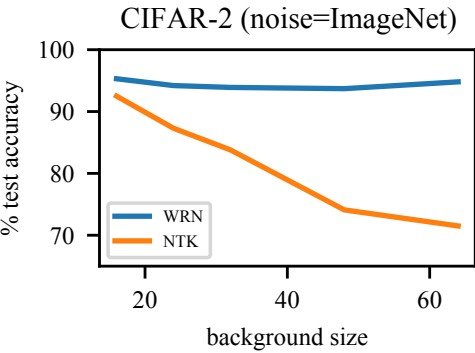

Figure 7: WRN vs NTK on CIFAR-2 with IMAGENET noise, varying size of the background noise image.

additional image-placement methods and at a larger size of 32x32 images on a 64x64 background. The image is either placed at a random location on the background, in a random corner, or in fixed corner.

Table 2: Test accuracy of WRN and NTK on CIFAR-2 with Gaussian noise. The image is placed on the noise background in a random location, random corner, or fixed corner. The image is sized 16x16 on a 32x32 background (first two rows), or 32x32 on a 64x64 background (last two rows).

| | | random location | | | | random corner | | | | fixed corner | | | |
|---|---|---|---|---|---|---|---|---|---|---|---|---|---|---|
| | | noise $\sigma$ | | | | noise $\sigma$ | | | | noise $\sigma$ | | | |
| size | | 0.0 | 0.5 | 1.0 | 2.0 | 0.0 | 0.5 | 1.0 | 2.0 | 0.0 | 0.5 | 1.0 | 2.0 |
| 16x16 | **WRN** | 94.7 | 94.0 | 92.8 | 91.7 | 94.3 | 94.2 | 92.7 | 92.6 | 94.4 | 94.3 | 94.0 | 92.9 |
| on 32x32 | **NTK** | 89.9 | 87.3 | 83.7 | 76.9 | 91.6 | 88.3 | 84.5 | 78.3 | 92.0 | 88.8 | 85.2 | 80.0 |
| 32x32 | **WRN** | 96.1 | 96.2 | 95.8 | 94.9 | 96.5 | 96.5 | 96.5 | 95.9 | 96.9 | 96.6 | 96.0 | 95.6 |
| on 64x64 | **NTK** | 91.2 | 89.4 | 84.6 | 82.2 | 91.8 | 89.1 | 85.3 | 79.1 | 93.1 | 91.0 | 88.3 | 82.9 |

Overall, we find that the gap in WRN vs. NTK performance degradation persists in all cases. With respect to image size, we note that all models perform better on the larger images. However, the WRN appears slightly less impacted by noise on the larger images, while degradation affects the NTK roughly equally on large and small images. With respect to image placement, virtually all models perform best with fixed-corner placement, followed by random-corner, followed by random-location. It is not clear that either the WRN or NTK is more affected by placement than the other.

**Noise background size.** To experiment with tunable structured noise which is somewhat more natural than pixel intensity, we conduct an additional experiment on CIFAR-2 with IMAGENET noise, but maintaining the full pixel intensity and varying the size of the background noise image. The 16x16 signal images are placed at a random location on IMAGENET images which are scaled between 16x16 and 64x64. Note that a background image of size 16x16 means that there is no noise.

The results are shown in Figure 7. We observe the same pattern as that in Section 6: the NTK degrades significantly as the size of the noise background increases, while the WRN performance is unaffected.

**Full MNIST finite-NTK width results.** In Figure 8 we include the full results of the CNN vs. finite-width NTK experiments shown in Figure 6 (right), for the full range of widths of NTK, from 50x to 100x wider than the CNN. We see that the degradation of the NTK performance is consistent across widths.

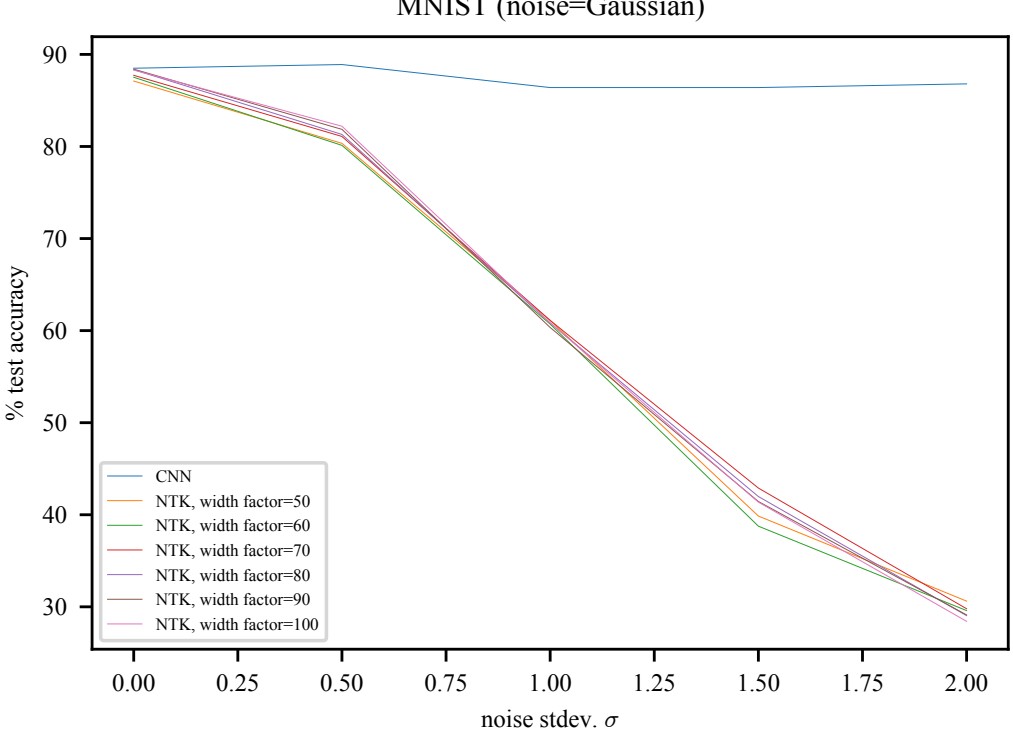

Figure 8: Comparison of CNN with various widths of finite-width NTK on MNIST.

# E  Societal impact

We elaborate on our ethics statement in the main paper as follows.

As stated there: "Our work is primarily theoretical in nature and analyzes existing methods; thus, to the best of our knowledge, it does not have any negative societal impact."

Here, we expand on "to the best of our knowledge". We recognize that there is a delicate feedback cycle created even by the community's selection of paper topics. By writing a paper on the advantages of neural networks over kernel methods, we are perhaps further advancing the perspective that neural networks are, in fact, worthy of "awe". We do not believe this perspective is inherently problematic. However, we do recognize that we have an obligation to prevent over-hype as well. We believe that this theoretical analysis of neural networks is an honest portrayal of one advantage of neural networks but recognize that it might, in fact, further enforce the hype to some degree. Our choice of dataset (CIFAR-10) also reinforces the community's focus on benchmark datasets. We recognize that there is a cost due to the community's focus on a limited set of benchmark datasets, and our paper does not push the community to move beyond such datasets. Thus, although this paper does not introduce any new negative societal impact, we realize that, by participating in various trends in the community, we are perpetuating certain cycles that likely need more careful, large-scale evaluation.