# OpenReview forum: "Local Signal Adaptivity: Provable Feature Learning in Neural Networks Beyond Kernels"
_NeurIPS.cc/2021/Conference — NeurIPS 2021 Poster_

### Official Review · Reviewer_5cbb · 2021-07-13

**Rating:** 7
**Confidence:** 3

**Summary:**

This paper introduces the notion of local signal adaptivity (LSA), which the authors claim is a new perspective on the benefits of feature learning in NNs over kernel methods like the NTK. LSA denotes the ability of non-linear NNs to find a sparse signal in the presence of large noise, and the authors theoretically show that in a particular data distribution setting with a sparse signal, a single filter CNN is able to train with GD to learn the signal. On the other hand, they show that for an underparameterised linear NN (linearised at initialisation in parameter space) the model is unable to learn the signal. Experimental results are provided comparing the degradation of non-linear and linear NNs under increasing noise, demonstrating that non-linear NNs are indeed more robust to noise in line with the theory.

**update after rebuttal** raising my score to 7

**Limitations And Societal Impact:**

- I find theorem 2 unsurprising, as in this setting we just don't have a rich enough hypothesis class (finite-channel hence underparameterised CNN NTK features at init) in order to fit the target class. The vast majority of interest in this area is to do with overparameterised NNs (which is also the settings used empirically in practice) and the benefits of feature learning vs kernel learning , so I don't really see the connection/importance of theorem 2 here. Given that the idea of 'LSA as a new angle on feature learning' is one of the key messages of the paper, I find this quite unsatisfactory. It would be more interesting if you could show e.g. sample complexity of nonlinear NNs is better relative to overparameterised linear NNs, or just develop the LSA idea more (perhaps going into some of the extensions you mention in the conclusion).
- I find the assumption of d changing polynomially with k to be a bit strange. So k is the number of patches, but then the patch sizes is also increasing as the number of patches changes? Is this assumed elsewhere in the literature? Likewise, the assumption that you have enough data such that the empirical gradient and the population loss gradient are close throughout training with mini-batches seems quite strong.
- The empirical results are not very convincing to me. In particular, because the linear WRN is already much worse than the non-linear WRN with 0 noise, it seems like there could be other effects going on to explain the degradation of the linear NN? It would make more sense to me to consider a case where we know that the finite-width linear NN and non-linear NN have similar performance without noise, before adding background noise. Does MNIST work (I know there is not much difference between infinite width NTK and non-linear NN)? Given that this is mainly a theoretical paper, I would find a more thorough smaller scale setting more convincing.

If the authors can address my concerns I would be happy to raise my score.

**Main Review:**

*originality* This paper offers a new take on the benefit of feature learning in NNs using the proposed LSA property, which as far as I am aware hasn't been studied before. Intuitively, sparse signals in the presence of noise are well fit description of image data, hence the relevance for CNNs and the image classification setting.

*quality* I believe the paper is technically sound (although I didn't check all the details). I find some of the results, like the main contribution Theorem 1, to be interesting, and also appreciate the authors' efforts in page 6+7 to clearly describe a sketch of the proof, which seems sensible.

*clarity* The paper is well written and organized, especially section 4 as mentioned.

*signficance* I think this seems like a solid, if unspectacular, contribution. The question of the benefits of feature learning compared to kernel methods in *overparameterised* NNs is an popular area which would be of general interest to the NeurIPS community, though as the authors mention there are many similar papers in this area. I have some gripes as to the relevance of this paper though for that area too (see limitations). I think the idea of LSA is interesting in and of itself, though also have some questions about assumptions which make the data setting seemingly far from standard (see limitations).

**Time Spent Reviewing:**

3

---

> ### Author Response · Authors · 2021-08-10
> **Response to Reviewer 5cbb**
>
> We thank the reviewer for thoughtful comments and an openness to raising the score.
>
> **Re: “It would be more interesting if you could show e.g. sample complexity of nonlinear NNs is better relative to overparameterised linear NNs”**
>
> Like the vast majority of interest in this area, we are also interested in overparameterized neural networks. However, to the best of our knowledge, showing a sample complexity lower bound for overparameterized feature mappings, specifically in the classification setting, is considered hard in complexity theory. We did consider showing such results in the regression setting instead, but we ultimately decided that a regression result would be less interesting/relevant, since we are really trying to study a classification-focused phenomenon. Therefore, to the best of our knowledge, the type of result we have shown (a representational lower bound) is the best one can reasonably hope to show right now in the classification setting. On the empirical side, we have attempted to overcome this theoretical limitation by studying heavily overparameterized finite-width NTKs. Specifically, in our CIFAR-10/WRN experiments, we have 50,000 training examples and 7.4 million parameters, and the networks achieve perfect training accuracy. Finally, we direct attention to Point 1 of our Reviewer cikC author response, which we believe is also relevant.
>
> We are definitely interested in further extensions of the LSA phenomenon in future work, and we are hopeful that this paper sufficiently lays the groundwork for this idea.
>
> **Re: “d changing polynomially with k”**
>
> As mentioned on lines 124-125, we set d=poly(k) so that we only have one parameter to work with; crucially, the proof of Theorem 1 should not have any dependence on what this polynomial actually is (i.e., it could be a constant wrt k). Alternatively, we could have treated k and d separately, but it would have complicated our analysis + lemma/theorem statements a bit.
>
> **Re: “the assumption that you have enough data such that the empirical gradient and the population loss gradient are close throughout training with mini-batches seems quite strong”**
>
> We emphasize that this is enough to show “efficient” learnability theoretically (i.e., polynomially many samples), compared to the representational lower bound. However, perhaps more significantly, the assumption that empirical and population gradients are close is used either explicitly or implicitly in various other theoretical works studying the beyond-NTK regime, such as https://arxiv.org/abs/2007.04596 (COLT 2020). Thus, we believe that this is a standard enough theoretical assumption, although admittedly imperfect.
>
> **Re: “the linear WRN is already much worse than the non-linear WRN with 0 noise”**
>
> We point out that although this is the case for CIFAR-10 and CIFAR-Vehicles, the gap is much smaller in CIFAR-2 (this was actually our primary motivation for studying CIFAR-2, as emphasized on lines 316-317). However, a small gap does persist. To close this gap, we took the reviewer’s recommendation and ran on MNIST as well, searching for a simple CNN architecture with comparable test accuracy between the original CNN and the finite-width NTK. We found that a 2-hidden-layer CNN with 8 and 16 channels (1st and 2nd layers, respectively) is comparable to a finite-width CNTK with 560 and 1120 channels. We randomly placed the 28x28 MNIST images on 42x42 backgrounds, using the same Gaussian noise scheme described in Section 6 of the paper; in the 0-noise setting here, the CNN reached a final test accuracy of 88.5% and the CNTK reached a final test accuracy of 87.8%, which are much closer. We then looked at the full set of standard deviations explored in the paper: 0.0, 0.5, 1.0, 1.5, 2.0. The CNN’s final test accuracy (in the same order) was 88.5%, 88.9%, 86.4%, 86.4%, 86.8%. The CNTK’s final test accuracy (in the same order) was 87.8%, 81.5%, 62.1%, 43.1%, 29.8%, showing significant degradation of the CNTK compared to the CNN. We will add these experiments to our final paper, as we agree that they enhance the empirical robustness of the work.

---

> > ### Comment · Reviewer_5cbb · 2021-08-23
> > **Thank you**
> >
> > Thank you for the reply and sorry for the delay. It is good to see the MNIST results back up your theory, and also thank you for your clarifications. I have a few more points:
> > - What exactly do you mean by 'representational lower bound'?
> > - I would contest the assertion that *We would also like to add that, to the best of our knowledge, there is a general acceptance in the community that, in order to study the gap between neural networks and their corresponding (infinite-width) neural tangent kernels, it is often more reasonable both empirically and theoretically to study the finite-width neural tangent kernels* which you wrote to reviewer cikC. There are phenomena such as double descent that you can observe in finite-width NTKs, or indeed random features models, (e.g. in https://arxiv.org/abs/2011.03321) which don't appear in infinite-width. This is important as your theory for poor performance of finite-width NTKs is in the underparameterised regime? It is quite interesting that you observe a degradation in CNTK still in the very wide very noisy case in your reply to point (3) of cikC, even though this is not capture by your theory (correct me if I'm wrong). As an aside: I don't think parameter counting is necessarily the best measure of 'overparameterisation', perhaps 0 training loss is better?

---

> > > ### Author Response · Authors · 2021-08-26
> > > **Thank you! And responses to additional points.**
> > >
> > > Thank you for your follow-up reply; we really appreciate the time you’re investing in our paper!
> > >
> > > Re: "representational lower bound":
> > > - By 'representational lower bound', we essentially meant 'approximation lower bound'. In other words, there is a lower bound on the number of filters needed in order for the CNTK to successfully represent/approximate the target classification function, regardless of what linear-combination weights on top of the CNTK features are allowed.
> > > - Regarding our motivation for this type of lower bound (instead of, for example, a sample complexity lower bound), we refer to our original response to "It would be more interesting if you could show e.g. sample complexity of nonlinear NNs is better relative to overparameterised linear NNs" (i.e., we do not disagree with your original suggestion but simply wish to emphasize the key challenges discussed in that response - specifically, "to the best of our knowledge, showing a sample complexity lower bound for overparameterized feature mappings, specifically in the classification setting, is considered hard in complexity theory").
> > >
> > > Re: "I would contest the assertion that…":
> > > In response to the comment that Theorem 2 is in the underparameterized regime: In some sense, this goes hand in hand with using a representational lower bound, which we do for the reasons discussed above. Now, taking it as given that we are using a representational lower bound:
> > > 1. First assuming the definition of overparameterization you have suggested ("0 training loss") -- it is unclear what a meaningful training set is for Theorem 2 and, further, what overparameterization with respect to a particular training set would imply, since Theorem 2 is a claim about the true distribution irrespective of training.
> > > 2. Instead, we could perhaps modify our notion of overparameterization and redefine it as "0 test loss", but then, by definition, a representational lower bound such as Theorem 2 implies that the model class is underparameterized.
> > >
> > > Therefore, other than switching to a sample complexity lower bound, is there any modification to Theorem 2 that would address your concerns?
> > >
> > > We also agree that "There are phenomena… that you can observe in finite-width NTKs… which don't appear in infinite-width", and we do not believe (and are not claiming) that all is equal between finite-width and infinite-width NTKs. However, we do believe that:
> > > 1. There are sufficient similarities to make our analysis of the finite-width CNTK interesting.
> > > 2. Our empirical results, including those you’ve mentioned, serve to bridge the gap between our more toy-ish CNTK (studied for theoretical convenience) and the more interesting very wide/increasing-width CNTKs.
> > > 3. On its own, it is sufficiently interesting to study the finite-width CNTK (since it is a more practical model, in many ways, than the infinite-width CNTK - as discussed in response to Reviewer cikC).
> > >
> > > Thank you again!

---

> > > > ### Comment · Reviewer_5cbb · 2021-08-27
> > > > **Thanks**
> > > >
> > > > Thanks for your reply which addresses my concerns. I am raising my score to a 7, 1 for the additional empirical results and 1 for the clarifications/explanations of motivation. It would perhaps be nice to see such explanation for the comparison between infinite and finite width CNTK in the paper as you promised to cikC.

---

> > > > > ### Author Response · Authors · 2021-09-01
> > > > > **Thank you!**
> > > > >
> > > > > Thank you! We will be sure to include this explanation in the final version of our paper. Relatedly, you might also be interested in the empirical infinite-width CNTK results shared in our latest response to Reviewer cikC.

---

### Official Review · Reviewer_cikC · 2021-07-16

**Rating:** 6
**Confidence:** 4

**Summary:**

The paper provides a theoretical analysis comparing the training dynamics and performance of a specific 1 hidden-layer convolutional neural network with those of its corresponding finite width CNTK on a simple data distribution. This analysis is used to support their claim that the gap between neural networks and kernel methods may lie in the ````''Local Signal Adaptivity'', in the sense that neural networks outperform kernel methods in distinguishing localized influential features from noisy background. Furthermore, they empirically identify a performance gap between the CNN and finite width CNTK on noise extended variants of the CIFAR-10 dataset, which emphasizes the influence of the ``''Local Signal Adaptivity''. The authors conclude that this setting degrades the performance of kernels in a more pronounced way than it affects neural networks.


**Ethical Concerns:**

None at this point.

**Limitations And Societal Impact:**

In my opinion, the authors have correctly addressed the potential negative societal impact of their work. However, their work has several limitations that have not been addressed in the paper (explained above along with suggestions for improvement).


**Main Review:**

Positives:

(1) Regarding the originality of the paper, it builds on a simple dataset similar to the one studied on a previous paper, as they explain in line 135. However, to the best of my knowledge, it is the first analysis of the gap between neural networks and kernels focusing on a locally informative image embedded in noise. Related work seems adequately cited.

(2) The main claims are supported by theoretical analysis for a simple network and dataset.

(3) The paper is well organized to my opinion and the theoretical results are well explained.

###################################

Limitations:

(1) Regarding the claims that neural networks have been shown to significantly outperform kernel methods in practice - I'm not sure this is precise.  There is evidence that the infinite width NTK and CNTK can outperform corresponding finite width neural networks and several other standard machine learning methods (e.g. see https://arxiv.org/abs/1910.01663 and https://arxiv.org/abs/2007.15801).  I think the authors should try to clarify their statements in the abstract and introduction to make clear that the gap studied in this work is between the finite width NTK/CNTK and the corresponding finite width network.

(2) I also have a few concerns about the conclusion drawn from the experimental results.  The authors claim that the experiments provide concrete empirical evidence that local signal adaptivity explains the performance gap between the finite width NTK and training finite width networks.  This experiment is performed using a WideResNet for 4 different settings.  Since the model used is widely different from the one considered in theoretical analysis, I feel the authors need to demonstrate that additional simpler models (maybe 1 to 3 hidden layer convolutional nets) also show a similar gap.

(3) Regarding the conclusions drawn about this gap between the finite width NTK and neural networks, I'm also a bit unsure whether this gap would disappear as width increases.   The authors already claim that the NTK requires many more features in order to identify the signal and threshold noise.  The main benefit of the NTK is that a closed form is readily computed for infinite width.  It seems entirely plausible that this performance gap decreases when increasing width.  Would the authors be able to run experiments to demonstrate that this is not the case?

(4) Lastly, I think the authors should also present the training accuracy for the finite width NTK and the finite width neural network to ensure that these are are trained to 100% training accuracy (and similar training losses).  This is important since the results could be a side-effect of underparameterization for the finite width NTK.  Please let me know if this is not the case.

I am happy to consider raising the score provided that some of these limitations are addressed.



**Time Spent Reviewing:**

4 hours

---

> ### Author Response · Authors · 2021-08-10
> **Response to Reviewer cikC**
>
> We thank the reviewer for helpful comments and an openness to raising the score.
>
> **Re: “(1) I think the authors should try to clarify their statements in the abstract and introduction to make clear that the gap studied in this work is between the finite width NTK/CNTK and the corresponding finite width network.”**
>
> Thanks for raising this point! We will make it clearer in the abstract and introduction that our theory and experiments use the finite-width CNTK vs. the infinite-width CNTK. We will also add a more nuanced discussion of the related empirical work on the finite- and infinite-width NTKs, to help make our empirical performance claims more precise.
>
> We would also like to add that, to the best of our knowledge, there is a general acceptance in the community that, in order to study the gap between neural networks and their corresponding (infinite-width) neural tangent kernels, it is often more reasonable both empirically and theoretically to study the finite-width neural tangent kernels. On the empirical side, infinite-width NTK training on a dataset like CIFAR-10, with a reasonable convolutional architecture, can take between 300-1000 GPU hours (https://arxiv.org/abs/2003.02237, https://arxiv.org/abs/2007.15801, https://github.com/google/neural-tangents - searching for “hours” in each of these should surface the relevant information). Training a finite-width NTK, on the other hand, is often much more computationally feasible. On the theoretical side, proving a classification lower bound for kernels/infinite-width NTKs (instead of a linear function over feature mappings) is considered extremely hard in complexity theory, and to our knowledge no such results exist in the literature. Therefore, although our work does not flesh out the link between finite- and infinite-width NTKs, we believe that there is a general acceptance that such results are nevertheless a very reasonable approach to this problem, given the limitations discussed above. If this additional context seems useful, we can add a discussion along these lines to the final version of our paper as well.
>
> **Re: “(2) I feel the authors need to demonstrate that additional simpler models (maybe 1 to 3 hidden layer convolutional nets) also show a similar gap.”**
>
> To address this point, we ran MNIST experiments with a 2-hidden-layer CNN. We chose a CNN with 8 and 16 channels (1st and 2nd layers, respectively) and a corresponding wide finite-width CNTK with 560 and 1120 channels. After some searching, we chose these particular widths to make the 0-noise performance of the 2 models as close as possible. We randomly placed the 28x28 MNIST images on 42x42 backgrounds, using the same Gaussian noise scheme described in Section 6 of the paper. We looked at the same standard deviations explored in the paper: 0.0, 0.5, 1.0, 1.5, 2.0. The CNN’s final test accuracy (in the same order) was 88.5%, 88.9%, 86.4%, 86.4%, 86.8%. The CNTK’s final test accuracy (in the same order) was 87.8%, 81.5%, 62.1%, 43.1%, 29.8%, showing significant degradation of the CNTK compared to the CNN. We will add these experiments to our final paper and thank the reviewer for the suggestion.
>
> **Re: “(3) It seems entirely plausible that this performance gap decreases when increasing width. Would the authors be able to run experiments to demonstrate that this is not the case?”**
>
> To explore this question, we have run experiments on MNIST on a 2-hidden-layer CNN. Very-high-width experiments on our original WideResNet were not computationally feasible, so we are hopeful that - per comment (2) above - the reviewer finds our simpler experiments sufficiently compelling (as this very-high-width finite-width NTK training already requires several GPU hours per run in this simplified setup). We examined MNIST (28x28) randomly located within a noise-free background (42x42) and MNIST (28x28) randomly located within a noisy background (42x42) with Gaussian noise of std. dev. $\sigma =2.0$. As above, our 2-hidden-layer CNN had 8 and 16 channels (1st and 2nd layers, respectively). Our finite-width CNTKs had between 400 and 720 channels in the 1st-layer and 800 to 1440 channels in the 2nd-layer. In the noise-free setting, as the width of the CNTK increased, we saw convergence to test accuracy 88%, approximately matching the original CNN’s test accuracy (89%). In the noise-2.0 setting, as the width increased, we saw CNTK test accuracy of 30% $\pm$ 0.5% for all widths, while the original CNN maintained a test accuracy of 87%. We believe that these noise scale endpoints (0 and 2.0) provide strong evidence that increasing width does not eliminate the performance gap (or even mitigate it), and we will flesh out these experiments (at intermediate noise levels between 0 and 2.0) and add such graphs to our final paper. If the reviewer has further related concerns that can be addressed by MNIST-scale width-increasing experiments, we are happy to try to run those during the discussion period.
>
> **Re: “(4) Lastly, I think the authors should also present the training accuracy for the finite width NTK and the finite width neural network to ensure that these are trained to 100% training accuracy (and similar training losses).”**
>
> We thank the reviewer for raising this point. We have confirmed that the finite-width NTKs reach around  99.9%-100% training accuracy by the end of training (200 epochs). This holds for the neural network (the WideResNet) too. Furthermore, the number of training examples (50,000) is much smaller than the number of parameters (7.4 million). Since the neural network and its finite-width NTK have the same number of parameters, this means that the neural network and finite-width NTK both have 150x more parameters than training examples. We believe that this satisfies a very reasonable definition of over-parameterization.

---

> > ### Comment · Reviewer_cikC · 2021-08-27
> > **Follow up**
> >
> > Thank you for the detailed response! I apologize for the delay in my response.  I just have a few questions/suggestions below.
> >
> > “We would also like to add that, to the best of our knowledge, there is a general acceptance in the community that, in order to study the gap between neural networks and their corresponding (infinite-width) neural tangent kernels, it is often more reasonable both empirically and theoretically to study the finite-width neural tangent kernels. […]”
> >
> > - I found this point a bit confusing since especially empirically it is not too difficult to run some simple experiments with the CNTK.  For example, the 300-1000 GPU hours reported with Google’s Neural Tangents library is using networks that involve Global Average Pooling (which if I understood correctly, is not crucial to the experiments in this paper).   I’ve definitely done some experiments without Global Average Pooling with this library on all of CIFAR10 and MNIST and it didn’t take too long (around 10 minutes if sufficiently many layers involve strided convolution).  It would really be helpful if even on a subset of MNIST, the authors could try again a 1 or 2 hidden layer infinite width CNTK to see how the performance compares.
> >
> > “To address this point, we ran MNIST experiments with a 2-hidden-layer CNN. We chose a CNN with 8 and 16 channels (1st and 2nd layers, respectively) and a corresponding wide finite-width CNTK with 560 and 1120 channels. After some searching, we chose these particular widths to make the 0-noise performance of the 2 models as close as possible. We randomly placed the 28x28 MNIST images on 42x42 backgrounds, using the same Gaussian noise scheme described in Section 6 of the paper. We looked at the same standard deviations explored in the paper: 0.0, 0.5, 1.0, 1.5, 2.0. The CNN’s final test accuracy (in the same order) was 88.5%, 88.9%, 86.4%, 86.4%, 86.8%. The CNTK’s final test accuracy (in the same order) was 87.8%, 81.5%, 62.1%, 43.1%, 29.8%, showing significant degradation of the CNTK compared to the CNN. We will add these experiments to our final paper and thank the reviewer for the suggestion.”
> >
> > - I thank the authors for running this new experiment.  I think it is definitely a clearer demonstration of the result for finite width CNTKs.  I’m not sure how difficult these were to run but the infinite width CNTK experiment could be easier to run in this setting provided that your architecture is not using Global Average Pooling.  The main reason I am interested in seeing the performance of the infinite width CNTK is that in theorem 2, having more hidden units is necessary for the finite width CNTK to even represent the target function.  I’m just unsure if there is a stronger claim to be made that the function learned by the infinite width CNTK is strictly worse for this setting with background noise even though it could potentially represent this function. Let me know if I am mistaken or confused about a point here though.
> >
> > “To explore this question, we have run experiments on MNIST on a 2-hidden-layer CNN. Very-high-width experiments on our original WideResNet were not computationally feasible, so we are hopeful that - per comment (2) above - the reviewer finds our simpler experiments sufficiently compelling (as this very-high-width finite-width NTK training already requires several GPU hours per run in this simplified setup). We examined MNIST (28x28) randomly located within a noise-free background (42x42) and MNIST (28x28) randomly located within a noisy background (42x42) with Gaussian noise of std. dev. . As above, our 2-hidden-layer CNN had 8 and 16 channels (1st and 2nd layers, respectively). Our finite-width CNTKs had between 400 and 720 channels in the 1st-layer and 800 to 1440 channels in the 2nd-layer. In the noise-free setting, as the width of the CNTK increased, we saw convergence to test accuracy 88%, approximately matching the original CNN’s test accuracy (89%). In the noise-2.0 setting, as the width increased, we saw CNTK test accuracy of 30% 0.5% for all widths, while the original CNN maintained a test accuracy of 87%. We believe that these noise scale endpoints (0 and 2.0) provide strong evidence that increasing width does not eliminate the performance gap (or even mitigate it), and we will flesh out these experiments (at intermediate noise levels between 0 and 2.0) and add such graphs to our final paper. If the reviewer has further related concerns that can be addressed by MNIST-scale width-increasing experiments, we are happy to try to run those during the discussion period.”
> >
> > - I completely agree that it is unreasonable to ask for high width experiments on WideResNet.  I definitely appreciate the simpler MNIST experiments.  My question based on this analysis is again whether infinite width would mitigate this performance drop in any way.  It seems to be not the case, but since solving kernel regression should be much easier with the infinite width CNTK, I would be interested in seeing if this is actually the case.
> >
> > Overall Response:
> >
> > In light of the authors’ new experiments, I have raised the score to a 6.  I know that I didn’t give too much time to run the experiment for the infinite width case, but I would still be interested in seeing this for even a small subset of MNIST.  The experiments for infinite width would be a bit involved since early stopping needs to be considered since this is also happening for the finite width models (i.e. the models are all trained to 100% accuracy not 0 loss if I understood correctly).  This can be done with EigenPro or with diagonal regularization (https://arxiv.org/abs/2007.15801).  In my opinion, I feel that this experiment would help to further bolster the claims made in this paper.

---

> > > ### Author Response · Authors · 2021-09-01
> > > **Infinite-width CNTK results**
> > >
> > > Thank you for the suggestion and for the helpful pointers!
> > >
> > > Following your advice, we were successfully able to run infinite-width experiments on a subset of MNIST (20K training examples), using the infinite-width CNTK corresponding to the simple CNN in our earlier reply.
> > >
> > > To account for early stopping, we explored various diagonal regularization values between 1e-1 and 1e-8 (prior to the trace scaling provided by Google's Neural Tangents library) and found that the test accuracy stayed within approximately a 2% range per Gaussian noise level. Therefore, for simplicity, we are reporting the values at regularization strength 1e-5 (at noise levels 0.0, 0.5, 1.0, 1.5, 2.0): 89.0%, 86.0%, 63.1%, 38.0%, 24.2%.
> > >
> > > For completeness, we also reran our CNN with a size-20K training sample (instead of 60K). We saw test accuracies: 86.2%, 86.0%, 81.6%, 78.3%, 78.4%.
> > > With a size-20K training sample, our finite-width CNTK had final test accuracies: 80.2%, 70.8%, 47.0%, 29.2%, 20.5%.
> > >
> > > These numbers show considerable degradation of the infinite-width CNTK compared to its corresponding CNN (roughly on par with the finite-width CNTK degradation).

---

### Official Review · Reviewer_X9Jj · 2021-07-16

**Rating:** 8
**Confidence:** 4

**Summary:**

This paper tackles the question of whether neural networks can be provable better than their induce neural tangent kernel for some problem setting. This question is answered in the context of a binary classification problem, in which the signal is shift invariant and embedded in a background of noise. The authors show that for this problem, a two layer neural network with relu activations and logistic loss efficiently learns (in sample and time complexity) this problem class with constant number of neurons, whereas its neural tangent kernel counterpart requires at least order d - the dimension of the (local) inputs. The authors then illustrate this basic problem in image classification settings where the problem setting is conceptually present, with results that align with their results in the toy-like distribution.


**Limitations And Societal Impact:**

Adequately addressed.

**Main Review:**

This paper was a pleasure to read. To the best of my knowledge, the results presented here are significant, as they illustrate a simple and clear setting in which the ability to learn the filters in a neural network model enables the problem to be learned efficiently, whereas building a kernel with their not-trained version does not.

I have only a few minor questions and comments:

1. The degree of difficulty in the problem is given, in a nutshell, buy the signal to noise ratio. The authors address the simple case where the classifier vector appears only once in the input matrix, but presumably the SNR could be improved by repeating the true vector in several patches. Is this correct?

2. It is clear that the data distribution is a strong simplification of the distribution of images for image classification. One difference between them is that, in real settings, the background patches are not really independent of the image class (determined by the foreground object). Examples of this are sky backgrounds for planes, water background for boats, etc. Could the authors comment on whether such correlations could be taken into account in their model? Presumably, one could decompose the "background" patches into a 'signal' component (e.g. only weakly correlated with the signal) and a random component?

3. Could the authors comment briefly on multi-class extensions of their analysis? Would it be the case that gradients will decompose into a 'true class' versus 'all other classes' components?

4. Finally, I believe that a more natural way to evaluate this problem in practice would have been to implant cifar10 images in backgrounds of different sizes, as opposed to controlling the energy of the background. This is also mentioned by authors, and unfortunately left as future work.

**Time Spent Reviewing:**

2.5

---

> ### Author Response · Authors · 2021-08-10
> **Response to Reviewer X9Jj**
>
> We are very glad that the reviewer enjoyed the paper and appreciate all of the feedback!
>
> **Re: (1)**&nbsp;&nbsp;&nbsp;That understanding is correct. For a constant number of signal patches, the extension should be simple. To change the SNR in a meaningful way, we could allow the number of signal patches to vary with k.
>
> **Re: (2) and (3)**&nbsp;&nbsp;&nbsp;We thank the reviewer for suggesting these possible extensions! We agree with the intuition that our analysis should support these extensions, primarily through a decomposition into a signal component and a remaining component. In the multi-class setting, extending to one filter per class seems like it could be relatively straightforward. We will look into these extensions in more detail and see if we can add them to the appendix or whether they have unexpected subtleties making them better suited for follow-up work.
>
> **Re: (4)**&nbsp;&nbsp;&nbsp;We agree that varying background size is a more natural evaluation; to this end we have replicated our CIFAR-2-with-ImageNet-noise experiment (Figure 4, bottom left), but with fixed pixel intensity of 1 (i.e., original ImageNet images), and varying the size of the background from no background to 64x64 background (image size 16x16). We found the same phenomenon as in the prior experiments: on noise backgrounds of size 24x24, 32x32, 48x48, and 64x64, the WRN retained 99% of its no-noise test accuracy while the NTK degraded to 94%, 91%, 80%, and 78% of its no-noise test accuracy. We will include the full experiment in the final version of the paper.

---

### Official Review · Reviewer_MLtr · 2021-07-21

**Rating:** 6
**Confidence:** 4

**Summary:**

The authors study a toy binary classification problem where a certain CNN
architecture is trained to detect one of two templates embedded in gaussian
noise. The motivation is twofold: the problem is argued to capture relevant
structures present in natural image classification tasks where CNNs succeed in
practice (i.e., the detection of an object/motif in the image in the presence
of a nuisance background), and the theoretical analysis establishes an
architectural separation via the rates one gets by analyzing the problem with
the neural tangent kernel approach and via the authors' analysis. Concretely,
the data model consists of a fixed unknown template embedded in one of $k+1$
consecutive disjoint length-$d$ blocks of a length $d(k+1)$ 1D signal, with the
other $k$ blocks containing i.i.d. gaussian noise of a certain variance (the
blocks are imagined to be disjoint "image patches", and the template a motif of
interest) -- samples from this data model have the patch where the template is
contained uniformly randomized and contain independent noise, and the template
is given either the positive or negative sign (representing the two classes in
the problem). The noise variance is set to be large enough that it is not
possible to perform naive matched filtering to recover the template.
Accordingly, the CNN that is trained to solve this task contains a custom
denoising nonlinearity, the standard soft thresholding function -- this
corresponds to a one-layer, two-neuron ReLU network that computes two
$d$-strided convolutions with specific weight sharing and output averaging (the
shared biases end up corresponding to the soft thresholding function's
parameter) -- which is trained using finite samples from the data generating
distribution by mini-batch SGD on the logistic loss with gaussian random
initialization. The authors show that this procedure yields a classifier for
the data whp given polynomial samples and suitable step sizes; in contrast,
they show that the network's convolutional NTK is unable to solve the task whp
as long as the number of filters satisfies $o(d)$ (the neural network studied
corresponds to a two filter network). Experiments involving embedding CIFAR-10
images into random backgrounds (either from ImageNet or having gaussian noise)
with varying intensities and comparing neural network performance to their
NTK's performance computationally on the classification tasks thus generated
verifies the predicted performance gap.



**Limitations And Societal Impact:**

No issues. In this connection, I would like to thank the authors for their
honest remarks in section D.


**Main Review:**

I thank the authors for their stimulating and interesting submission. I would
like to highlight what I perceive as a few strengths of the paper:
- The theoretical task, while fundamentally simple and idealized (as the
  authors acknowledge throughout the paper), has a strong motivation in the
  understanding of aspects of neural network training problems that match the
  settings where they are successful in practice, here specifically in the
  context of image classification. This has the potential to spur further
  impactful research.
- In a similar vein, one major implication of the results is that kernel
  approaches to analyzing the neural network training problem cannot capture
  the same guarantees as in the authors' analysis without using additional
  network resources. A number of works of this type have appeared in the
  literature; the authors' has the advantage of pertaining to an algorithmic
  learning scenario with a practically-motivated data model, which sets it
  apart and will hopefully influence future works in this line.
- In addition to being well-motivated, the paper is well-written, specifically
  in its technical aspects. The sketch of the argument in section 4 is
  particularly lucid.

I will discuss what I perceive as some weaknesses/points where clarification
would be helpful below. I do not think these negatively outweigh the paper's
strengths. I hope the authors will correct any misconceptions I make in my
discussion in the rebuttal.


### Toy-ness of the task

- I am onboard with the idea that the task is simple in order to act as a minimal
  demonstration of a separation between kernel tools and direct methods for
  analyzing neural network training that may have broader implications in the
  future, and that it is therein essential to consider a CNN training scenario
  with the given data model. However, it seems notable that the classification
  task studied is algorithmically trivial: given that the class of CNN
  architectures we search over with gradient descent is essentially
  hand-designed to perform the denoising we need in the task, it is not
  surprising that a simple algorithm solves this problem (it seems: given one
  labeled sample from $\mathcal{D}$, use this as a template to perform
  filtering with new samples: cross-correlate over blocks on new samples and soft
  threshold the output (if the noise distribution is not known, the threshold
  level could be learned from a training set using classical detection
  methods), then use the label from before to determine the class of the
  result). A dual view here is that a "data driven initialization" approach
  trivializes the problem. As before, I understand that the purpose of the
  paper is to use this simple task as a way to study some interesting
  properties of CNN training, but the task's algorithmic simplicity seems to
  point to a need to include more challenging aspects in the problem
  (overlapping blocks; templates obscured in noise; non-antipodal classes;
  correlated models for the background noise) in order to obtain an analysis
  that can truly offer insights into the features relevant to practical image
  classification tasks.
- In a similar vein, the CNN architecture used here feels ad-hoc designed to
  perform well on the task, and some possibly unrealistic prior knowledge is
  assumed in the non-overlapping blocks and the incorporation of $d$ as the
  stride (along with all samples being "canonized" in the same block
  alignment). This is reminiscent of other separations in the literature
  between neural networks and kernel methods, which are often established on
  tasks where the target function is per force well-specified by some neural
  network architecture by construction of the task (notwithstanding the good
  practical motivation here). Specifically in the context of some of the other
  motivation in the "Learning a hidden subspace" paragraph, it seems like the
  current task, rather than being one of learning a target $f(x) = \phi(Wx)$ as
  in related works, is one of learning a target $f(x) = \phi(W_i x)$, where the
  $W_i$s vary randomly over some finite set and one obtains different
  observations of this form. In this connection, I would find it interesting to
  know if there is any connection to be made with the analysis of [1] here.

### Connections to signal processing literature?

- Given that the task studied here is essentially a simple denoising task in
  the language/formalism of CNN training, it would seem appropriate to mention
  any relevant prior art in theory for neural networks and denoising (e.g. [2]
  and much related work). I understand that the specific motivations and
  focuses of the current work may be different from those in this literature
  (here, the simple task is a vehicle to illustrate a point about neural
  network training; there the tasks are more general and theoretical results
  may be directly relevant to applications) and that this may make these
  works not directly relevant; however, the focus on denoising and image data
  makes it seem likely to me that there should be nontrivial overlap.
- One area of this literature I do feel may likely be relevant in terms of
  technical prior art is the literature on theory for sparse blind
  deconvolution (e.g. [3-6] below), given that the authors' analysis here is an
  analysis of the gradient descent trajectory. Natural optimization
  formulations for these tasks collapse into the minimization of a loss like
  the authors' $\ell( f_{w, b}(X), y)$ in the non-overlapping-data setting the
  authors study here, but the loss will be different from the logistic loss
  (e.g. something like the Huber penalty function, which involves
  soft-thresholding of large magnitude components at a scale commensurate with
  the regularization threshold, which corresponds to $b < 0$ here). In the
  multi-template classification setting with multiple observations, as
  considered here, the areas of convolutional dictionary learning [7] and
  multi-channel sparse blind deconvolution [8], respectively, may have the
  relevant connections (given the functional similarities) -- although in the
  context of the authors' simple data model the additional generality inherent
  in these frameworks may not add anything relative to the authors' CNN
  training framework. I would be interested to hear if the authors detect any
  technical overlap between their work and this area of the literature -- I
  suppose works on noisy SBD will be most relevant to the authors' work, and of
  course the results and models considered in these works apply to more general
  problem scenarios than the present setting where the motif locations do not
  overlap. I also believe there may be some interesting messages with regards
  to this literature from the authors' work here -- the fact that the parameter
  $b$ is learned in the current work may be of interest in SBD settings where
  one would like to adaptively set the regularization parameter, and I am not
  aware of theoretical analyses that proceed in this way.

### References mentioned above

[1] http://arxiv.org/abs/2102.13219

[2] https://ieeexplore.ieee.org/document/8398588/

[3] https://arxiv.org/abs/1806.00338

[4] https://arxiv.org/abs/1901.00256v2

[5] https://arxiv.org/abs/2007.06753

[6] https://arxiv.org/abs/2106.07053

[7] https://arxiv.org/abs/1709.02893

[8] https://arxiv.org/abs/1908.10776


**Time Spent Reviewing:**

5

---

> ### Author Response · Authors · 2021-08-10
> **Response to Reviewer MLtr**
>
> We thank the reviewer for very clearly-articulated comments.
>
> Regarding the toyness of the task, we wish to emphasize a few points that might help further clarify our goals:
> 1. We agree that there are various aspects of practical image classification tasks not captured by this model. However, in this work, we are studying what we believe to be one key aspect of practical image classification problems that can drive a separation between convolutional neural networks and their corresponding neural tangent kernels. Therefore, rather than enhancing our distribution with additional features that appear in practical image classification tasks, which could lead to questions about which features are actually driving the separation, we have chosen to focus on this single feature.
> 2. Although other algorithms can solve this problem, we do not see this as a weakness, as the existence of alternative algorithms does not explain why gradient-descent-trained neural networks themselves succeed. From a theoretical perspective, understanding the behavior of gradient-descent-trained neural networks on very simple tasks, even on linearly separable data, is considered highly non-trivial (e.g., https://arxiv.org/abs/1710.10345). To the best of our knowledge, almost all existing works that study efficient learning of neural networks using gradient descent are for learning concept classes that are learnable by other simple algorithms.
>
> Therefore, with these two points in mind, we hope that the “toyness” of the data distribution - as a means of getting to the heart of the phenomenon we are studying - can also be viewed as a key strength.
>
> To the best of our knowledge, [1] studies the benefit of invariant architectures over non-invariant architectures (e.g., convolutional architectures vs. non-convolutional architectures), whereas our work shows the benefit of an invariant neural network over its corresponding invariant neural tangent kernel. So, in that sense, [1] seems related but distinct from our work. It is certainly possible, though, that more detailed insights from [1] can help to further contextualize our results, and we will take a closer look at it and consider adding it to our Related work section.
>
> We thank the reviewer for various references to the signal processing literature, especially on sparse blind deconvolution. We will continue to look through these references and incorporate them into the Related work section of our final version.

---

> > ### Comment · Reviewer_MLtr · 2021-09-02
> > **response**
> >
> > Dear authors,
> >
> > Thank you for the response to my review. I take your points regarding the task, but also tend to maintain my original rating. At the same time, I agree that your contribution here is quite interesting, and should spur interesting future deep learning theory research.

---

### Decision · Program_Chairs · 2021-09-27

**Decision:**

Accept (Poster)

**Comment:**

The paper is motivated by the need for mathematical understanding of mechanisms of feature learning in neural networks, and for understanding the limitations of kernel models (i.e., neural tangent kernel NTK) for learning in randomly initialized networks. The paper studies a very simple model problem, in which the goal is to determine whether a given input contains a noisy copy of a certain motif, or simply noise. In positive samples, the location of the motif is chosen uniformly at random.

This simple model problem provides a separation between kernel methods, and methods which learn features: the paper considers single-layer nonlinear networks, and shows that (i) this problem can be solved by a network which only two neurons [which essentially implements a soft thresholding denoiser], which can be learned by randomly initialized  stochastic gradient descent, and (ii) the corresponding neural tangent kernel is only capable of achieving nontrivial performance when the number of neurons exceeds the length, d, of the motif.

The paper complements these observations with experiments on both Gaussian backgrounds and “visual clutter”, involving more complex architectures, and showing a performance gap between the trained network and the corresponding kernel model, when the background is prominent.

Initial reviews appreciated the paper’s clarity, and its contribution to the discussion on gaps between kernels and feature learning. Reviewers also noted several limitations — the “handcrafted” nature of the neural network and the simplicity of this model problem — and raised questions about comparisons to finite vs infinity width neural tangent kernels.

After discussion of the above points, the reviewers converged to a decision to accept the paper. The AC concurs with this evaluation. There are some obvious limitations to the analysis. At the same time, the strength of the paper is that it provides rigorous results on feature learning and separations vis kernel methods in the practically relevant setting of detecting a motif (feature) in background clutter. The experiments suggest that these phenomena also obtain for more realistic networks and backgrounds. The paper is clearly written, and seems likely to stimulate future work.